# Catoni Contextual Bandits are Robust to Heavy-tailed Rewards

**Chenlu Ye** [1]   **Yujia Jin** [2]   **Alekh Agarwal** [3]   **Tong Zhang** [4]

## Abstract

Typical contextual bandit algorithms assume that the rewards at each round lie in some fixed range $[0, R]$, and their regret scales polynomially with this reward range $R$. However, many practical scenarios naturally involve heavy-tailed rewards or rewards where the worst-case range can be substantially larger than the variance. In this paper, we develop an algorithmic approach building on Catoni's estimator from robust statistics, and apply it to contextual bandits with general function approximation. When the variance of the reward at each round is known, we use a variance-weighted regression approach and establish a regret bound that depends only on the cumulative reward variance and logarithmically on the reward range $R$ as well as the number of rounds $T$. For the unknown-variance case, we further propose a careful peeling-based algorithm and remove the need for cumbersome variance estimation. With additional dependence on the fourth moment, our algorithm also enjoys a variance-based bound with logarithmic reward-range dependence. Moreover, we demonstrate the optimality of the leading-order term in our regret bound through a matching lower bound.

## 1. Introduction

Minimax optimal regret bounds in the worst-case over problem instances for contextual bandit learning are relatively well-understood in the literature, both using policy-based approaches in the agnostic case, and regression-based approaches in the realizable case. A variety of algorithms attain these bounds in both settings, and the minimax optimality implies that the bounds are unimprovable in general. When the expected reward of each action is realizable using

some function class $\mathcal{F}$ available to the learner, this optimal regret scales as $O(R\sqrt{Td_{\mathcal{F}} \log N_{\mathcal{F}}})$, where $R$ is the range of the rewards, $T$ is the number of rounds, $d_{\mathcal{F}}$ is a complexity notion for $\mathcal{F}$, such as the eluder dimension (Russo & Van Roy, 2013), and $N_{\mathcal{F}}$ is the covering number of $\mathcal{F}$.

However, this worst-case behavior arises only when the rewards span their entire range $[0, R]$ with a significant probability, a phenomenon not typical in practice. Even for a common case of binary rewards in $\{0, R\}$ for instance, the expected reward is often relatively close to $0$ in common click/no-click style recommendation settings with low clickthrough rates. Consequently, the expectation, variance and even higher moments of the reward are much smaller than the worst-case range. More generally, rewards with heavier tails naturally arise when considering waiting times in wireless communication networks (Nair et al., 2013), stock prices in financial markets (Cont, 2001; Hull, 2012), or value returns for online advertising (Choi et al., 2020; Jebarajakirthy et al., 2021). In this paper, we study the design of contextual bandit algorithms that can leverage such structures to have regret guarantees dependent polynomially on the reward variance, with only a mild logarithmic scaling with the range parameter $R$.

Such variance-based regret bounds have received significant attention recently, under the topic of *robustness to heavy-tailed rewards*. Huang et al. (2024); Li & Sun (2024) study Huber regression and design variance-weighted regression-based approaches for linear contextual bandits with known variance, and show that their algorithms achieve a variance-based $\widetilde{O}\big(d\sqrt{\sum_{t\in[T]} \sigma_t^2}\big)$ regret bound, where $d$ is the dimension for the linear function, thus avoiding a dependence on range $R$. They also study Markov Decision Processes

---

[*]Equal contribution [1]University of Illinois Urbana-Champaign [2]OpenAI (Work done during an internship at Google) [3]Google Research [4]University of Illinois Urbana-Champaign. Correspondence to: Chenlu Ye <chenluy3@illinois.edu>.

*Proceedings of the 42nd International Conference on Machine Learning*, Vancouver, Canada. PMLR 267, 2025. Copyright 2025 by the author(s).

[1]Huang et al. (2024) consider a more general setting, where the $1 + \epsilon$-th moment of the reward is upper bounded for some $\epsilon \in (0, 1]$, and incur a dependence in terms of this moment along with additional $poly(T)$ terms. Since our work only considers bounded variance, we present their result with (Li & Sun, 2024) together, as the two results are identical for the case of $\epsilon = 1$.

[2]DistUCB relies on a stonger assumption: estimating the full reward distribution rather than just the mean, and hence relies on a stronger realizability on the function class to capture this distribution. In contrast, we only assume realizability of the reward mean $R$. Moreover, $\tilde{d}_{\mathcal{F}}$ measures the complexity of a class of distributions, while $d_{\mathcal{F}}$ measures the complexity of a class of mean functions, thus the two notions are generally incomparable.

*Table 1.* Comparison between different algorithms for stochastic contextual bandits, where $d$ denotes the dimension for linear function approximation, $d_{\mathcal{F}}, \tilde{d}_{\mathcal{F}}$ capture the complexity of the function space $\mathcal{F}$ used for reward estimation, $T$ is the number of rounds, $\sigma_t$ is the variance of the observed reward at round $t$, $\sigma$ is a uniform bound on reward variance ($\sigma_t \leq \sigma$ for all $t \in [T]$), $R$ is the range of rewards, and $N_{\mathcal{F}}$ is the covering number for function class $\mathcal{F}$. $\widetilde{O}$ omits terms logarithmic in $T$ and $R$.

| Algorithm | Function Type | Known Variances | Regret Bound |
|---|---|---|---|
| Weighted OFUL+ (Zhou & Gu, 2022) | Linear | ✓ | $\widetilde{O}\big(d\sqrt{\sum_{t\in[T]}\sigma_t^2} + dR\big)$ |
| Heavy-OFUL (Huang et al., 2024) [1] AdaOFUL (Li & Sun, 2024) | Linear | ✓ | $\widetilde{O}\big(d\sqrt{\sum_{t\in[T]}\sigma_t^2}\big)$ |
| OLS (Pacchiano, 2024) | Non-linear | ✓ | $\widetilde{O}\big(\sigma\sqrt{d_{\mathcal{F}}\log N_{\mathcal{F}}} + Rd_{\mathcal{F}}\log N_{\mathcal{F}}\big)$ |
| Catoni-OFUL (Theorem 3.4) | Non-linear | ✓ | $\widetilde{O}\big(\sqrt{\sum_{t\in[T]}\sigma_t^2 \cdot d_{\mathcal{F}}\log N_{\mathcal{F}}} + d_{\mathcal{F}}\log N_{\mathcal{F}}\big)$ |
| SAVE (Zhao et al., 2023b) | Linear | ✗ | $\widetilde{O}\big(d\sqrt{\sum_{t\in[T]}\sigma_t^2} + dR\big)$ |
| DistUCB (Wang et al., 2024b) [2] | Non-linear | ✗ | $\widetilde{O}\big(\sqrt{\sum_{t\in[T]}\sigma_t^2 \cdot \tilde{d}_{\mathcal{F}}\log N_{\mathcal{F}}} + R\tilde{d}_{\mathcal{F}}\log N_{\mathcal{F}}\big)$ |
| Unknown-Variance OLS (Pacchiano, 2024) | Non-linear | ✗ | $\widetilde{O}\big(d_{\mathcal{F}}\sqrt{\sum_{t\in[T]}\sigma_t^2 \cdot \log N_{\mathcal{F}}} + Rd_{\mathcal{F}}\log N_{\mathcal{F}}\big)$ |
| VACB (Theorem 4.2) | Non-linear | ✗ | $\widetilde{O}\big(d_{\mathcal{F}}\sqrt{\sum_{t\in[T]}\sigma_t^2 \cdot \log N_{\mathcal{F}}} + d_{\mathcal{F}}(\log N_{\mathcal{F}})^{3/4}\big)$ |

(MDPs) with linear function approximation under heavy-tailed rewards with unknown variance, and use the linearity of both expected rewards and variances in linear MDPs to design a weighted regression algorithm relying on variance estimation. To the best of our knowledge, these works heavily rely on the linear function structure and are hard to extend to the non-linear setting. The general question of designing a robust contextual bandit algorithm under the heavy-tailed reward (or a reward with a large range) for general function approximation is still lacking in the literature.

A different line of work called distributional RL estimates the full reward distribution (Wang et al., 2024b;a;c) under the unknown variance case to achieve variance-based regret bounds with general function approximation. However, their focus is on replacing the $T$-based scaling with the cumulative variance and still incurs a polynomial dependence on $R$. Additionally, the distributional approach requires the stronger modeling assumption that the full reward distribution, rather than just the expected reward is realizable.

There are some works considering the unknown variance case for contextual bandits without realizability conditions for the noise (Zhang et al., 2021; Kim et al., 2021; Zhao et al., 2023b; Pacchiano, 2024). Particularly, the most relevant ones to our work for the unknown-variance setting are Zhao et al. (2023b); Pacchiano (2024). Zhao et al. (2023b) develop a peeling approach for the unknown variance case without variance estimation in linear settings, and Pacchiano (2024) extend this technique to general function approximation. Nevertheless, all of these algorithms have an $O(dR)$ or $O(Rd_{\mathcal{F}}\log N_{\mathcal{F}})$ term in the regret bound. We summarize the key results from the prior literature in Table 1 to better contextualize our results, and defer additional related works

to Appendix A.2.

## 1.1. Additional Related Works

There is another line of literature targeting on standard multi-armed bandit (MAB) setting with heavy-tailed rewards (Huang et al., 2022; Chen et al., 2024; Genalti et al., 2024), where they obtain sharper gap-dependent bounds in the MAB setting. In the contextual setting with general function approximation, however, one has lower bounds even in the linear setting with large action spaces (Lattimore, 2017). Consequently, the results are not directly comparable with ours. Regarding algorithms, we adopt the OFUL framework combined with weighted Catoni estimators and peeling techniques. In contrast, Huang et al. (2022) uses a skipping method based on Follow-the-Regularized-Leader, and Genalti et al. (2024); Chen et al. (2024) design adaptive algorithms capable of handling unknown and unknown moment bounds. The remaining related works are deferred to Appendix A.2.

## 1.2. Our contributions

This work considers a different route for robustness to heavy-tailed rewards, building on the well-studied Catoni's mean estimator from the robust statistics literature. We design a contextual bandit (CB) algorithm that uses the Catoni mean as a robust device for constructing a regression error estimator for the **excess loss**, given some function class $\mathcal{F}$ for predicting the expected reward. Using the variance-dependent concentration of the Catoni estimator, we conduct a careful analysis of our algorithm and show that its regret scales as $\widetilde{O}\big(\sqrt{\sum_{t\in[T]}\sigma_t^2 \cdot d_{\mathcal{F}}\log N_{\mathcal{F}}} + d_{\mathcal{F}}\log N_{\mathcal{F}}\big)$, when

the reward variance $\sigma_t$ is known at each round $t$.

Since reward variance information is seldom available in practice, we refine our approach for cases with unknown variances by employing a multi-level uncertainty estimation for the expected rewards of a carefully chosen subset of actions. For this approach, we obtain regret guarantees dependent on the fourth moment of the reward, while still maintaining a logarithmic scaling in $R$. Formally, the regret scales as $\widetilde{O}\big(d_{\mathcal{F}}\sqrt{\sum_{t\in[T]}\sigma_t^2 \cdot \log N_{\mathcal{F}}} + d_{\mathcal{F}}(\log N_{\mathcal{F}})^{3/4}\big)$. Notably, our method does not rely on some other function class to help predict the per-round variance as a function. Instead, we estimate a robust averaged variance quantity, and show that it approximates the averaged true variance up to logarithmic factors in $R$.

Overall, our results significantly improve the state-of-the-art in variance-aware regret guarantees, that are amenable to practical reward structures. We summarize our results relative to the most relevant prior literature in Table 1.

## 2. Preliminary

**Notations.** For any integer $n$, we use the short-hand notation $[n] = \{1, \ldots, n\}$, and define $x_{[n]} = \{x_1, \ldots, x_n\}$. We use $\widetilde{O}$ to omit terms logarithmic in $T$ and $R$. The comprehensive table of notations is provided in Appendix A.1.

We consider a contextual bandit problem over $T$ rounds of interactions between an agent and the environment. At each round $t \in [T]$, the environment generates a decision set $\mathcal{X}_t \in \mathcal{X}$, where each element $x \in \mathcal{X}_t$ is a candidate action for the agent. After observing $\mathcal{X}_t$, the agent plays an action $x_t \in \mathcal{X}_t$ and observes the reward $y_t = f^\star(x_t) + \eta_t$. Particularly, this setting subsumes classic contextual bandit where an action $a_t \in \mathcal{A}$ is chosen upon observing a context $z_t$ at round $t$, since we can always set $\mathcal{X}_t = \{z_t \times \mathcal{A}\}$. We make the standard boundedness assumptions that

$$|\eta_t| \leq R, \ \mathbb{E}\eta_t = 0, \ \mathbb{E}\eta_t^2 \leq \sigma_t^2.$$

We assume access to a function class $\mathcal{F} : \mathcal{X} \to [-R, R]$ such that $f^\star \in \mathcal{F}$. For a function class $\mathcal{F}$, we recall the standard definitions of $\epsilon$-cover and covering number (see e.g., Wainwright (2019); Zhang (2023a)) as follows.

**Definition 2.1** ($\upsilon$-cover and covering number). Given a function class $\mathcal{F}$, for each $\upsilon > 0$, a $\upsilon$-cover of $\mathcal{F}$ with respect to $\|\cdot\|_\infty$, denoted by $\mathcal{C}(\mathcal{F}, \upsilon)$, satisfies that for any $f \in \mathcal{F}$, we can find $f' \in \mathcal{C}(\mathcal{F}, \upsilon)$ such that $\|f - f'\|_\infty \leq \upsilon$. The $\upsilon$-covering number, denoted as $N(\upsilon, \mathcal{F})$, is the smallest cardinality of such a $\mathcal{C}(\mathcal{F}, \upsilon)$.

We assume that the function class $\mathcal{F}$ consists of bounded functions, that is, $|f(x)| \leq L_f$ for all $f \in \mathcal{F}$ and $x \in \mathcal{X}$. The variances $\sigma_t$ at each time step $t$ are not necessarily

known. The (pseudo-) regret is defined as

$$R_T = \mathbb{E}\sum_{t\in[T]}\left[\max_{x\in\mathcal{X}_t}f^\star(x) - f^\star(x_t)\right].$$

To describe the structure of the general function class, we define the following (eluder dimension) quantities (Gentile et al., 2022; Russo & Van Roy, 2013) as

**Definition 2.2** (Eluder dimension (Gentile et al., 2022)). Given a sequence of ordered actions $X = (x_1, x_2, \cdots, x_T) \in \mathcal{X}_1 \times \mathcal{X}_2 \times \cdots \times \mathcal{X}_T$ and a function class $\mathcal{F}$, let the eluder coefficients be

$$D_{\mathcal{F}}^2(x, \bar{\sigma}; x_{[t-1]}, \bar{\sigma}_{[t-1]})$$
$$:= \sup_{f_1, f_2 \in \mathcal{F}} \frac{(f_1(x) - f_2(x))^2/\bar{\sigma}^2}{\sum_{i\in[t-1]}(f_1(x_i) - f_2(x_i))^2/\bar{\sigma}_i^2 + \lambda},$$
$$D_{\mathcal{F}}(x; x_{[t-1]}, \bar{\sigma}_{[t-1]}) := D_{\mathcal{F}}(x, 1; x_{[t-1]}, \bar{\sigma}_{[t-1]}).$$

Then we define the eluder dimension as:

$$\dim(\mathcal{F}, X, \bar{\sigma}_{[T]}) := \sum_{i=1}^{T} \min\left(1, D_{\mathcal{F}}^2(x_i, \bar{\sigma}_i; x_{[i-1]}, \bar{\sigma}_{[i-1]})\right),$$

$$\dim_{\alpha, T}(\mathcal{F}) := \max_{X, \bar{\sigma}_{[T]}: |X|=T, \bar{\sigma}_1, \ldots, \bar{\sigma}_t \geq \alpha} \dim(\mathcal{F}, X, \bar{\sigma}_{[T]}).$$

The weighted eluder coefficient $D_{\mathcal{F}}^2$ describes at each time step $t$, how much the in-sample error can bound the out-of-sample error. We can illustrate the eluder quantities with linear function approximation. If the function class $\mathcal{F}$ is embedded into a linear mapping $\mathcal{F} = \{\theta^\top\phi(\cdot, \cdot) : \theta \in \mathbb{R}^d, \|\theta\|_2 \leq B\}$, and we define the covariance matrix $\Sigma_t = \sum_{i\in[t]} x_i x_i^\top/\bar{\sigma}_i^2$, the weighted eluder coefficient can be simplified as

$$D_{\mathcal{F}}^2(x, \bar{\sigma}; x_{[t-1]}, \bar{\sigma}_{[t-1]})$$
$$= \sup_{\theta_1, \theta_2 \in \mathbb{R}^d} \frac{((\theta_1 - \theta_2)^\top\phi(x)/\bar{\sigma})^2}{\sum_{i\in[t-1]}((\theta_1 - \theta_2)^\top\phi(x_i)/\bar{\sigma}_i)^2} \leq \left\|\frac{\phi(x)}{\bar{\sigma}}\right\|_{\Sigma_{t-1}}^2,$$
$$\tag{1}$$

where the inequality applies Cauchy–Schwarz inequality. Hence, the eluder coefficient reduces to how much a direction is explored in the linear case.

The summation of eluder coefficients over $T$ time steps is the eluder dimension. The (weighted) eluder coefficients and the eluder dimension are broadly used in general function approximation (Zhang, 2023b; Ye et al., 2023; Agarwal et al., 2023; Zhao et al., 2023a). For the linear case in $d$-dimensions, when all the $\bar{\sigma}, \bar{\sigma}_{[t-1]}$ are 1, the $\dim_{1,T}(\mathcal{F})$ can be bounded in terms of $d \log d$ (Zhang, 2023b; Agarwal et al., 2023). When the weights are larger than $\alpha$, we can regard $\phi'(x) = \phi(x)/\bar{\sigma}$ as the new feature representation and bound the $\dim_{\alpha, T}(\mathcal{F})$ via the elliptical potential lemma (Abbasi-Yadkori & Neu, 2014).

## 3. Bandits with Known Variance

In this section, we present upper and lower bounds, when the per-round variance of each action is known to the learner.

## 3.1. Lower Bound

We start with a minimax lower bound for the class of multi-armed bandit problems where the variance of each action's reward is known to the learner.

**Theorem 3.1.** *For any integer $T > 0$, there exists a contextual bandit problem such that any $\pi = \{\pi_t\}_{t=1}^T$ will incur regret at least $\Omega(\sqrt{\mathbb{E}\sum_{t=1}^T \sigma_t^2})$, where $\{\sigma_t = \mathrm{Var}_{x_t \sim \pi_t}[y_t]\}_{t=1}^T$ and the expectation is jointly over any randomness in the environment as well as the algorithm.*

In other words, the theorem states that the regret of any contextual bandit algorithm scales with the square root of the sum of the variances of the rewards for its chosen actions. That is, it rules out a regret bound which scales solely as the variance of the reward of the optimal policy's actions.

The detailed proof is deferred to Appendix B.1. The intuition is to construct two bandit instances, each with two arms $\{x_1, x_2\}$. In the first instance, the arm $x_1$ has a deterministic reward while $x_2$ has a higher expected reward, but with a large variance. In the second instance, $x_2$ has a smaller mean reward. Clearly, the optimal action $x_1$ has a variance of zero in the first instance, but any algorithm needs sufficiently many draws of $x_2$ as well to distinguish between the two instances. In the sequel, we will present a matching upper bound for our robust estimator.

## 3.2. Upper Bound for Known Variance

**Catoni Estimator**   We first introduce Catoni estimator. This is a robust estimator proposed by Audibert & Catoni (2011)(see also (Lugosi & Mendelson, 2019)) to estimate random variables with bounded variance and unbounded range. Following Lugosi & Mendelson (2019, section 2.2), to estimate $t^{-1}\sum_{i\in[t]}\mathbb{E}Z_i$, we first define a function

$$\Psi(x) = \begin{cases} \log(1 + x + x^2/2) & \text{if } x \geq 0, \\ -\log(1 - x + x^2/2) & \text{if } x < 0. \end{cases}$$

Then for some parameter $\theta > 0$, $\mathsf{Catoni}_\theta(\{Z_i\}_{i\in[t]})$ is the unique zero of the antisymmetric increasing function

$$f(x; \{Z_i\}_{i\in[t]}, \theta) := \sum_{i\in[t]} \Psi(\theta(Z_i - x)). \qquad (2)$$

We first provide the following result about the concentration properties of the Catoni estimator, which we use in various places to prove why our design of confidence sets in the sequel algorithms.

**Lemma 3.2** (Informal). *Let $Z_t$ be a random variable adapted to filtration the $\mathcal{H}_t$, with a uniform bound $|Z_t| \leq R$, $\mathbb{E}[Z_i|\mathcal{H}_{i-1}] = \mu_i$, $\sum_{i\in[t]}\mathbb{E}\left[(Z_i - \mu_i)^2 |\mathcal{H}_{i-1}\right] \leq V$ for some fixed $V$. Let $\bar{\mu} := t^{-1}\sum_{i\in[t]}\mu_i$. Let $\theta \in [a, A]$ be a*

---

**Algorithm 1** Catoni-OFUL

**Input:** Parameter $\alpha > 0$, $\delta$ and $\hat{\beta}_t$ for each $t \in [T]$.
**for** t=1,2,…,T **do**
    Pick action $x_t = \mathrm{argmax}_{x\in\mathcal{X}_t}\max_{f\in\mathcal{F}_{t-1}} f(x)$;
    Observe the reward $y_t$;
    Let $\bar{\sigma}_t = \max\left(\alpha, \sigma_t, \sqrt{4\iota(\delta)L_f D_{\mathcal{F}_{t-1}}(x_t; x_{[t-1]}, \bar{\sigma}_{[t-1]})}\right)$;
    Estimate $\hat{f}_t$ in (3);
    Construct confidence set

$$\mathcal{F}_t := \left\{ f \in \mathcal{F}_{t-1} : \sum_{i\in[t]} \frac{1}{\bar{\sigma}_i^2}\left(f(x_i) - \hat{f}_t(x_i)\right)^2 \leq \hat{\beta}_t^2 \right\};$$

**end for**

---

*parameter, for some constants $a$, $A$ independent of $Z_i$. For an appropriate $\epsilon$ and any large enough $t$, with probability at least $1 - 2\delta$ we have uniformly for all $\theta \in [a, A]$:*

$$\left|\mathsf{Catoni}_\theta(\{Z_i\}_{i\in[t]}) - \bar{\mu}\right| \leq \frac{\theta\left(V + \sum_{i\in[t]}(\mu_i - \bar{\mu})^2\right)}{t}$$
$$+ \frac{4\iota_0^2}{\theta t} + \frac{\epsilon}{t},$$

*where $\iota_0$ contains log terms and is given in Appendix E.1.*

This inequality differs from the prior concentration results for the Catoni estimator as it is uniform for all $\theta \in [a, A]$. In the sequel, we use this flexibility to choose $\theta$ based on the samples. The formal version of the lemma and the proof are deferred to Appendix E.1.

**Algorithm**   By incorporating the Catoni estimator into the Optimism in the Face of Uncertainty Learning (OFUL)(Abbasi-Yadkori et al., 2011), we propose the Catoni-OFUL approach in Algorithm 1. Given failure probabilities $\delta$ and confidence parameters $\hat{\beta}_t$, the algorithm chooses the action $x_t$ with the highest optimistic reward by maximizing across all functions in a confidence set $\mathcal{F}_t$, as in the standard OFUL approach.

The key difference lies in the construction of a robust confidence set based on Catoni's mean estimator. We first define a per-sample weight $\bar{\sigma}_t$ as the maximum of a parameter $\alpha$, the variance $\sigma_t$ of the reward of $x_t$, and an uncertainty term based on the eluder coefficient $D_{\mathcal{F}_{t-1}}(x_t; x_{[t-1]}, \bar{\sigma}_{[t-1]})$.

Then, we define a robust estimator of $f^\star$, given the data, as the solution to the following saddle-point problem:

$$\hat{f}_t = \mathrm{argmin}_{\hat{f}\in\mathcal{F}}\max_{f'\in\mathcal{F}} L_t(\hat{f}, f') := \sum_{i\in[t]} \frac{1}{\bar{\sigma}_i^2}(f'(x_i) - \hat{f}(x_i))^2$$
$$+ 2t\mathsf{Catoni}_{\theta_t(\hat{f}, f')}(\{Z_i(\hat{f}, f')\}_{i\in[t]}), \qquad (3)$$

where we use the notation $Z_i(f, f') := \bar{\sigma}_i^{-2}(f(x_i) - f'(x_i))(f'(x_i) - y_i)$, $\iota(\delta)$ scales as $\widetilde{O}(\sqrt{\log(1/\delta)})$ and is

specified in Table 2, and the parameter $\theta_t(\hat{f}, f')$ is also specified in Table 2.

To understand this definition, we observe that $L_t(f, f')$ is a robust sample-based estimator of the true excess risk:

$$
\begin{aligned}
R_t(f, f') &:= \sum_{i \in [t]} \frac{1}{\bar{\sigma}_i^2} \Big[ \mathbb{E}_i[(f(x_i) - y_i)^2] - \mathbb{E}_i[(f'(x_i) - y_i)^2] \Big] \\
&= \sum_{i \in [t]} \frac{1}{\bar{\sigma}_i^2} \mathbb{E}_i \bigg[ (f(x_i) - f'(x_i))^2 \\
&\qquad + 2 \underbrace{(f(x_i) - f'(x_i))(f'(x_i) - y_i)}_{\mathcal{I}_i} \bigg],
\end{aligned}
$$
(4)

where the expectation $\mathbb{E}_i$ is taken with respect to the noise $\eta_i$. Since $\mathcal{I}_i$ is the only term that depends on the heavy-tailed noise, it is approximated by the $\theta$-robust Catoni estimator $\mathsf{Catoni}_\theta(\{Z_i(f, f')\}_{i \in [t]})$ in $L_t(f, f')$. Then, we include all the $f \in \mathcal{F}_{t-1}$ that have a small weighted squared loss to $\hat{f}_t$ in the confidence set $\mathcal{F}_t$.

*Remark* 3.3. Since the min-max optimization in (3) can be hard to solve, we provide an alternative (Algorithm 3 in Appendix D), where we construct a candidate set first similar to the confidence set, and then choose an estimator from the candidates randomly. This approach can improve the optimization efficiency and ensure the same regret bound as Theorem 3.4. Similar results also apply to the unknown variance case in the next section.

**Theorem 3.4** (Informal). *Under Algorithm 1 with appropriate choices of the parameters $\alpha, \lambda, \upsilon$ and $\hat{\beta}$, with probability $1 - 2\delta$, we can bound the regret by*

$$
\begin{aligned}
R_T = \widetilde{O}\Big( &L_f \sqrt{\sum_{t \in [T]} \sigma_t^2 \cdot \dim_{\frac{1}{\sqrt{T}}, T}(\mathcal{F}) \cdot \log \mathcal{N}(\mathcal{F}, \upsilon)} \\
&+ L_f \cdot \log \mathcal{N}(\mathcal{F}, \upsilon) \cdot \dim_{\frac{1}{\sqrt{T}}, T}(\mathcal{F}) \Big).
\end{aligned}
$$

The formal version of the theorem and appropriate choices of the hyperparameters are provided in Appendix B.2. The variance dependence in our theorem matches the lower bound in Theorem 3.1. Specifically, for the deterministic case where $\sigma_t = 0$ for all $t \in [T]$, the bound is reduced to $\widetilde{O}(\log \mathcal{N}(\mathcal{F}, \upsilon) \cdot \dim_{\frac{1}{\sqrt{T}}, T}(\mathcal{F}))$, and in the worst case where $\sigma_t = \Theta(1)$ for all $t \in [T]$, the bound becomes $\widetilde{O}(\sqrt{T \dim_{\frac{1}{\sqrt{T}}, T}(\mathcal{F}) \cdot \log \mathcal{N}(\mathcal{F}, \upsilon)})$. We note that the bound depends only polylogarithmically on $R$, improving upon most prior results as observed in Table 1.

### 3.3. Proof Sketch

To illustrate the intuition clearly, we ignore the covering number in this subsection, and assume that the function space $\mathcal{F}$ is finite. The detailed proof considers an infinite function space and uses the uniform covering number. The novelty of the proof lies in the following two parts.

**Part I: Concentration of excess loss**  Recall that for any $f, f' \in \mathcal{F}$, we formulate the excess loss $L_t(f, f')$ to estimate the excess loss $R_t(f, f')$ in (4) under heavy-tailed noise. Here the conditional expectation of the variable $Z_i(f, f')$ is $\mathbb{E}[Z_i|x_i] = (f(x_i) - f'(x_i))(f'(x_i) - f^\star(x_i))/\bar{\sigma}_i^2$. Since the standard Hoeffding's inequality leads to the error dependent on the uniform noise bound $R$, which can be extremely large in our setting, we can utilize the robustness of the Catoni estimator via Lemma 3.2 to obtain the following lemma.

**Lemma 3.5.** *For all large enough time steps $t$ and two fixed $f, f' \in \mathcal{F}_{t-1}$, with a proper choice of parameters $\alpha$, $\hat{\beta}_t$ and $\epsilon$, we have with probability at least $1 - \delta/N^2$,*

$$
\begin{aligned}
&\Big| L_t(f, f') - R_t(f, f') \Big| \\
&= 2 \Big| t \mathsf{Catoni}_{\theta_t(f, f')}(\{Z_i(f, f')\}_{i \in [t]}) - \sum_{i \in [t]} \mathbb{E}[Z_i(f, f')|x_i] \Big| \\
&\leq \frac{1}{2} V_t(f, f') + \frac{1}{3} \hat{\beta}_t^2,
\end{aligned}
$$

*where $V_t(f, f') = \sum_{i \in [t]} (f(x_i) - f'(x_i))^2 / \bar{\sigma}_i^2$.*

The values of the parameters are shown in Lemma B.2, and the proof is deferred to Appendix B.2. Importantly, $\hat{\beta}_t^2$ in the lemma above only has $\log R$ dependence.

**Part II: Sharpness of the confidence set**  Next, we show that for our choice of $\hat{\beta}_t$, the true function $f^\star \in \mathcal{F}_t$ with a high probability for $t \in [T]$ that are appropriately large. To prove this, we define $L_t(f) := \max_{f' \in \mathcal{F}} L_t(f, f')$, apply Lemma 3.5 with $f = \hat{f}_t$, and take a minimum over $f \in \mathcal{F}$ on both sides of the inequality to get the following result.

**Lemma 3.6.** *Under the conditions of Lemma 3.5, we have for all large enough $t \in [T]$ with probability at least $1 - \delta$,*

$$
\begin{aligned}
L_t(\hat{f}_t) &\geq \max_{f' \in \mathcal{F}} \Big\{ V_t(\hat{f}_t, f^\star) - V_t(f', f^\star) - \frac{1}{2} V_t(\hat{f}_t, f') - \frac{1}{3} \hat{\beta}_t^2 \Big\} \\
&= \frac{2}{3} V_t(\hat{f}_t, f^\star) - \frac{1}{3} \hat{\beta}_t^2,
\end{aligned}
$$

*where the maximizer for $f'$ is $f'_{\max} = \frac{2}{3} f^\star + \frac{1}{3} \hat{f}_t$.*

The proof is deferred to Appendix B.2. An analogous argument also yields an upper bound $L_t(f^\star) \leq \hat{\beta}_t^2/3$, as shown in Appendix B.2. Furthermore, since $\hat{f}_t$ is the minimizer of $L_t(\cdot)$, we have

$$
0 \geq L_t(\hat{f}_t) - L_t(f^\star) \geq \frac{2}{3} V_t(\hat{f}_t, f^\star) - \frac{2}{3} \hat{\beta}_t^2,
$$

which leads to $f^\star \in \mathcal{F}_t$.

Ultimately, if the event $f^\star \in \mathcal{F}_t$ for a large enough $t$ happens, the regret can be bounded by using the definition of $\hat{\beta}_t$ and the definition of the eluder dimension. Since this part is standard, we defer the details to Appendix B.2.

**Algorithm 2** Variance-Agnostic Catoni Bandit

---

1: **Input:** Parameter $\gamma > 0$, $L = \lceil \log_2(1/\gamma) \rceil$, $l_\star = \lceil \log_2(1076\iota'(\delta)) \rceil$.
2: Initialize the estimators for all layers: $\lambda^l \leftarrow 2^{-2l}$, $\hat{\beta}_0^l \leftarrow 2^{-l+1}$, $\Psi_0^l \leftarrow \emptyset$ for all $l \in [l_\star, L]$.
3: **for** t=1,…,T **do**
4:    Observe $\mathcal{X}_t$, and initialize $\mathcal{X}_t^1 \leftarrow \mathcal{X}_t$, $l \leftarrow l_\star$.
5:    **while** $x_t$ is not specified **do**
6:      **if** $D_t^l(x) \leq \gamma$ for all $x \in \mathcal{X}_t^l$ **then**
7:         Choose $x_t$, $f_{t-1}^l \leftarrow \mathrm{argmax}_{x \in \mathcal{X}_t^l, f \in \mathcal{F}_{t-1}^l} f(x)$
8:         Observe $y_t$.
9:         **Break.**
10:      **else if** $D_t^l(x) \leq 2^{-l}$ for all $x \in \mathcal{X}_t^l$ **then**
11:         Update $\mathcal{X}_t^{l+1} \leftarrow \{x \in \mathcal{X}_t^l \mid \hat{f}_{t-1}^l(x) \geq \max_{x \in \mathcal{X}_t^l} \hat{f}_{t-1}^l(x) - 2^{-l+1}\hat{\beta}_{t-1}^l\}$.
12:      **else**
13:         Choose $x_t \in \mathcal{X}_t^l$ such that $D_t^l(x_t) > 2^{-l}$ and observe $y_t$.
14:         Update $w_t \leftarrow 2^l D_t^l(x_t)$.
15:         Update the index sets: $\Psi_t^l \leftarrow \Psi_{t-1}^l \cup \{t\}$ and $\Psi_t^{l'} \leftarrow \Psi_{t-1}^{l'}$ for $l' \neq l$.
16:         Optimize $\hat{f}_t^l$ as in (7), and choose the confidence set $\mathcal{F}_t^l$ defined in (9).
17:      **end if**
18:      Update $l \leftarrow l + 1$.
19:    **end while**
20:    For $l \in [L]$ s.t. $\Psi_t^l = \Psi_{t-1}^l$, $\hat{f}_t^l \leftarrow \hat{f}_{t-1}^l$, $\mathcal{F}_t^l \leftarrow \mathcal{F}_{t-1}^l$.
21: **end for**

---

# 4. Bandits with Unknown Variance

In this section, we generalize to the case where the noise variance $\mathbb{E}\eta_t^2$ for any $t \in [T]$ is unknown. In addition to the assumption that for any $f \in \mathcal{F}$, $\|f\|_\infty \leq L_f$, and $\|f^\star\|_\infty \in [0, 1]$, the following condition for the noise variance is required.

**Assumption 4.1.** For each time step $t \in [T]$, the noise $\eta_t$ satisfies that there exist positive constants $\sigma_\eta$ and $c_\eta$ such that $\mathbb{E}[\eta_t|\mathcal{F}_t] = 0$, $\mathbb{E}\eta_t^2 = \sigma_t^2 \leq \sigma_\eta^2$ and $\mathrm{Var}[\eta_t^2|\mathcal{F}_t] \leq c_\eta \mathrm{Var}[\eta_t|\mathcal{F}_t]$.

## 4.1. Algorithm

Since variances $\sigma_t$ are unknown, traditional variance weighting techniques necessitate an accurate estimation for the noise variance at each time step (Huang et al., 2024; Li & Sun, 2024). To circumvent the complicated variance estimation, we adapt the SupLinUCB-type (Chu et al., 2011) algorithm with adaptive variance-aware exploration from Zhao et al. (2023b) to propose Variance-Agnostic Catoni Bandit (VACB) in Algorithm 2, where we split the contexts $\{x_t\}_{t \in [T]}$ into $L$ subsets according to their uncertainty. For each level $l \in [L]$, let $\Psi_t^l$ denote the set of time indexes within $[t]$ when the estimator update happens. Specifically, we use the following short-hand notation of uncertainty with respect to history information in $\Psi_t^l$: for any $x \in \mathcal{X}$,

$$D_t^l(x) = \sup_{f, f' \in \mathcal{F}_{t-1}^l} \frac{|f(x) - f'(x)|}{\sqrt{\sum_{i \in \Psi_{t-1}^l}(f(x_i) - f'(x_i))^2/w_i^2 + \lambda^l}}. \tag{5}$$

At each time step $t$, starting from $l = l_\star$, if there exists a decision $x \in \mathcal{X}_t^l$ with sufficiently large uncertainty $D_t^l(x_t) > 2^{-l}$, this decision will be chosen; otherwise, all the actions $x \in \mathcal{X}_t^l$ that are far from the optimal reward $\max_{x \in \mathcal{X}_t^l} \hat{f}_{t-1}^l(x)$ are eliminated, and the remaining actions compose the decision set $\mathcal{X}_t^{l+1}$ at the next level. The process does not stop until (a) there exists an action with large uncertainty; (b) or the uncertainty of all the remaining decisions is small ($D_t^l(x_t) \leq \gamma$ for all $X_t \in \mathcal{X}_t^l$). If case (a) happens, we will construct the estimation for the current layer. Specifically, for each level $l$, the variance estimator $\widehat{\mathrm{Var}}_t^l$ uses plug-in:

$$\widehat{\mathrm{Var}}_t^l := t\mathsf{Catoni}_{\theta_{\mathrm{Var}}^{t,l}}\left(\left\{\frac{1}{w_i^2}(y_i - \hat{f}_{t-1}^l(x_i))^2\right\}_{i \in \Psi_t^l}\right) + \hat{b}_t^l, \tag{6}$$

where the detailed choice of bonus $\hat{b}_t^l$, the parameter $\theta_{\mathrm{Var}}^{t,l}$ and $\iota'(\delta)$ are provided in Table 3. Then, the function estimation follows Algorithm 1:

$$\hat{f}_t^l = \mathrm{argmin}_{\hat{f} \in \mathcal{F}_{t-1}^l} \max_{f' \in \mathcal{F}_{t-1}^l} L_t^l(\hat{f}, f') := \sum_{i \in \Psi_t^l} \frac{1}{w_i^2}(f'(x_i) - \hat{f}(x_i))^2$$
$$+ 2t\mathsf{Catoni}_{\theta_t'(\hat{f}, f')}(\{Z_i(\hat{f}, f')\}_{i \in \Psi_t^l}), \tag{7}$$

where $Z_i(f, f') := (f(x_i) - f'(x_i))(f'(x_i) - y_i)/w_i^2$, the parameter $\theta_t'(f, f')$ is given in Table 3. Essentially, the weight $w_i$ can substitute the per-round variance in normalizing the loss $L_t^l$, as we will show in Lemma 4.5 that $\widehat{\mathrm{Var}}_t^l$ can be upper and lower bounded by the true variance $\sum_{i \in \Psi_t^l} \sigma_i^2/w_i^2$ up to additive and multiplicative constants. Furthermore, since $\widehat{\mathrm{Var}}_t^l$ appears in the Catoni-mean's concentration (Lemma 3.2) for $t^{-1}\sum_{i \in \Psi_t^l} Z_i(f, f')$, we see that normalizing the losses with $w_i$ results in variance-aware concentration just like the known variance case.

For the two parameters require knowledge of the cumulative variance, we substitute the true summation with the optimistic variance estimator $\widehat{\mathrm{Var}}_t^l$: one is $\theta'(f, f')$ defined above, and the other is $\hat{\beta}_t^l$, which is iteratively computed:

$$(\hat{\beta}_t^l)^2 = \Theta\left((\iota'(\delta))^2 2^{-2l}\widehat{\mathrm{Var}}_t^l + \iota'(\delta)2^{-2l} + \Delta_\upsilon + \lambda^l\right), \tag{8}$$

where the specific value of $\hat{\beta}_t^l$ and $\Delta_\upsilon$ is provided in Table 3, and $\Delta_\upsilon$ is a small term depending on the parameter $\upsilon$ for the $\upsilon$-cover in Definition 2.1.

In summary, our algorithm needs to estimate only an aggregate variance instead of estimating the per-round vari-

ance exactly, as we would require for applying a variance-weighted directly in the agnostic setting. This requires access to another function class that can model variances, which we cleanly avoid. Finally, we define the confidence set

$$\mathcal{F}_t^l := \Big\{ f \in \mathcal{F}_{t-1}^l : \sum_{i \in \Psi_t^l} \frac{1}{w_i^2} \left( f(x_i) - \hat{f}_t^l(x_i) \right)^2 + \lambda^l \le (\hat{\beta}_t^l)^2 \Big\}. \tag{9}$$

### 4.2. Analysis

**Theorem 4.2** (Informal)**.** *Suppose that Assumption 4.1 holds. With appropriate choices of $\gamma$, $\iota'(\delta)$, $\upsilon$ and $\hat{\beta}_t^l$, if $T$ is large enough, with probability at least $1 - 3\delta$, we can bound the regret of Algorithm 2 by*

$$R_T = \widetilde{O}\Big( L_f \Big( \sum_{t \in [T]} \sigma_t^2 \cdot \log \mathcal{N}(\mathcal{F}, \upsilon) \Big)^{1/2} \cdot \dim_{1,T}(\mathcal{F})$$

$$+ L_f \dim_{1,T}(\mathcal{F}) (\log \mathcal{N}(\mathcal{F}, \upsilon))^{3/4} (\sqrt{c_\eta} + \sigma_\eta) \Big).$$

This theorem provides a variance-dependent upper bound when variances are unknown, which matches the upper bound when variances are observed (Theorem 3.4) up to a slightly worse dependence on the eluder dimension. The higher order dimension term arises in the analysis of the peeling technique.

When compared to the upper bound $\widetilde{O}(d\sqrt{\sum_{t \in [T]} \sigma_t^2} + d^{3.5}T^{1/4})$ (Li & Sun, 2024) for the linear setting with dimension $d$ and unknown variance, our bound gets rid of the dependence on $T^{1/4}$, which means that our bound is still optimal up to the dimension when the sum of variances is small: $\sum_{t \in [T]} \sigma_t^2 = o(\sqrt{T})$. We give more intuition on why our algorithm admits this sharper bound in the proof sketch below, with details deferred to Appendix C.

*Remark* 4.3. We compare our results here with a contemporary work (Jia et al., 2024). The main difference is that they assume that the noise is bounded such that $r_t \in [0, 1]$, and we consider heavy-tailed rewards. Besides, although we both get variance-dependent bounds, the focuses are distinct: they aim to obtain better regret bounds when the eluder dimension is larger than the number of actions, while we aim to obtain logarithmic dependence on the reward range. Hence, we develop algorithms based on the OFUL structure, while Jia et al. (2024) build on the SquareCB approach. Regardless of the dependence on reward range, for the weak adversary with revealed variance, our upper bound $\tilde{O}(\sqrt{\Lambda \cdot d_\mathcal{F} \log N})$ is incomparable to theirs $\tilde{O}(\sqrt{A\Lambda \log N} + d_\mathcal{F} \log N)$. For the strong adversary, we both use the peeling technique and thus, our bound is superior on the dependence of reward range.

**Proof Sketch** The main challenges for the variance-agnostic algorithm are: (I) how to obtain the concentration inequalities when the weights are not based on the noise variance; (II) how to make accurate substitutions for the sum of variance $\sum_{i \in [t]} \sigma_i^2 / w_i^2$ in the parameters; and (III) how to deal with the regret of each level $l$. The insight of solving challenges (I) and (III) basically follows previous work (Zhao et al., 2023b; Pacchiano, 2024), but for (II), because of the heavy-tailed setting, our contribution is designing the robust Catoni variance estimator, and demonstrating the estimator almost has the same order as the true average-variance $\sum_i \sigma_i^2 / w_i^2$ up to constants with logarithmic dependence on the reward range $R$. We address these challenges in the following three parts, respectively.

**Part I: Average variance bound for concentration** In this part, we study the concentration of the excess loss for each level. For clearer illustration, we omit level $l$ when there is no confusion and denote $D_i = D_i^l(x_i)$ for short. Distinct from the known variance case where one directly takes the variance $\sigma_t$ as weights to derive an upper bound for the variance of $Z_i(f, f')$, we start with an alternate bound in terms of the weights $w_i$:

$$S_t := \sum_{i \in \Psi_t} \text{Var}[Z_i(f, f')]$$

$$= \sum_{i \in \Psi_t} \mathbb{E}\Big[ \frac{1}{w_i^2} (f(x_i) - f'(x_i))^2 (f^\star(x_i) - y_i)^2 \,\Big|\, x_i \Big]$$

$$\le \sum_{i \in \Psi_t} \frac{(f(x_i) - f'(x_i))^2}{w_i^2} \cdot \frac{\sigma_i^2}{w_i^2}.$$

When the variance is known, and $w_i = \sigma_i$ as in Algorithm 1, the second term in the final inequality is uniformly equal to 1.

When the variance is unknown, we can no longer weight the variances, instead, we uniformly bound the first term and aggregate the second term as shown below:

$$S_t \le \max_{i \in \Psi_t} \frac{(f(x_i) - f'(x_i))^2}{w_i^2} \cdot \sum_{i \in \Psi_t} \frac{\sigma_i^2}{w_i^2}$$

$$\le \max_{i \in \Psi_t} \frac{D_i^2}{w_i^2} \cdot \underbrace{\Big( \sum_{\tau \in [i-1]} \frac{(f(x_\tau) - f'(x_\tau))^2}{w_\tau^2} + \lambda \Big)}_{\text{Uniform bound } \le 2^{-2l} \cdot 4\hat{\beta}_{t-1}^2} \cdot \sum_{i \in \Psi_t} \frac{\sigma_i^2}{w_i^2},$$

where the first inequality uses the definition of $D_i$, and the uniform bound holds because $D_i / w_i \le 2^{-l}$ at level $l$ from Algorithm 2. Also $f, f' \in \mathcal{F}_{t-1}$ implies that

$$\sum_{\tau \in [i-1]} \frac{(f(x_\tau) - f'(x_\tau))^2}{w_\tau^2} \le 2 \sum_{\tau \in [t-1]} \frac{(f(x_\tau) - \hat{f}_{t-1}(x_\tau))^2}{w_\tau^2}$$

$$+ 2 \sum_{\tau \in [t-1]} \frac{(f'(x_\tau) - \hat{f}_{t-1}(x_\tau))^2}{w_\tau^2}$$

$$\le 4\hat{\beta}_{t-1}^2.$$

Here the first inequality applies the Cauchy-Schwarz inequality, and the second inequality follows from the definition of $\mathcal{F}_{t-1}$.

Therefore, by not requiring a uniform bound on the closeness of the variances and the weights, we can successfully derive the concentration inequality for $L_t(f, f') - R_t(f, f')$.

**Lemma 4.4.** *Under Assumption 4.1 and Algorithm 2, we have $\sup_{i \in \Psi_t} D_i/w_i \leq 2^{-l} := \rho$, $w_i \geq 1$. Then, for a large enough $t \in [T]$ and for any $f, f' \in \mathcal{F}_{t-1}$, with probability at least $1 - \delta/TL$,*

$$
\left| L_t(f, f') - R_t(f, f') \right| \leq \frac{1}{2} V_t(f, f') + \frac{1}{2} \hat{\beta}_{t-1}^2
$$
$$
+ O\left( (\iota'(\delta))^2 \rho^2 \left( \frac{\left( \sum_{i \in \Psi_t} \sigma_i^2/w_i^2 \right)^2}{\widehat{\mathrm{Var}}_t} + \widehat{\mathrm{Var}}_t \right) + \Delta_v \right).
$$

The more involved version of this lemma and the proof is presented in Appendix C.1. Note that the variance estimator $\widehat{\mathrm{Var}}_t$ appears on the right-hand side of the inequality above because the parameter $\theta'_t(\hat{f}, f')$ cannot be directly set in terms of $\sum_{i \in \Psi_t} \sigma_i^2/w_i^2$, and we instead use the surrogate $\widehat{\mathrm{Var}}_t$. The $\widehat{\mathrm{Var}}_t$ dependence will be eliminated after demonstrating the close relationship between $\widehat{\mathrm{Var}}_t$ and $\sum_{i \in \Psi_t} \sigma_i^2/w_i^2$ in the next part.

**Part II: Accuracy of variance estimation.** For any $t \in [t]$, let $V_i(\hat{f}_{t-1}) = (y_i - \hat{f}_{t-1}(x_i))^2/w_i^2$. The conditional expectation of this term is

$$
\mathbb{E}[V_i(\hat{f}_{t-1})|x_i] = (\sigma_i^2 + (f^\star(x_i) - \hat{f}_{t-1}(x_i))^2)/w_i^2.
$$

Then, by using Lemma 3.2 and using an argument similar to Part I, we can control the concentration error

$$
\left| t\mathsf{Catoni}_{\theta_{\mathrm{Var}}^t}\left( \{V_i(\hat{f}_{t-1})\}_{i \in \Psi_t} \right) - \sum_{i \in \Psi_t} \mathbb{E}[V_i(\hat{f}_{t-1})|x_i] \right|.
$$

Hence, it suffices to bound the gap between the variance and the expectation:

$$
\left| \sum_{i \in \Psi_t} \mathbb{E}[V_i(\hat{f}_{t-1})|x_i] - \sum_{i \in \Psi_t} \frac{\sigma_i^2}{w_i^2} \right|
$$
$$
= \sum_{i \in \Psi_t} \frac{(f^\star(x_i) - \hat{f}_{t-1}(x_i))^2}{w_i^2}
$$
$$
= \sum_{i \in \Psi_t} \frac{(\hat{f}_{t-1}(x_i) - f^\star(x_i))^2}{w_i^2} + \frac{(\hat{f}_{t-1}(x_t) - f^\star(x_t))^2}{w_t^2}
$$
$$
\leq \hat{\beta}_{t-1}^2 + \frac{D_t^2}{w_t^2} \cdot \hat{\beta}_{t-1}^2 \leq (1 + 2^{-2l})\hat{\beta}_{t-1}^2,
$$

where the first inequality uses $f^\star \in \mathcal{F}_{t-1}$, the definition of $D_i$ and $D_i/w_i = 2^{-l}$.

Recalling the definition (6) of $\widehat{\mathrm{Var}}_t$, we derive the following accuracy guarantee of this estimate compared with the true aggregated weighted variance:

**Lemma 4.5.** *Under Algorithm 2 and the condition that $f^\star \in \mathcal{F}_{t-1}^l$, when $2^l$ is large enough, we have with probability at least $1 - 2\delta$ for all large enough $t \in [T]$,*

$$
\sum_{i \in \Psi_t} \frac{\sigma_i^2}{w_i^2} \leq 2\widehat{\mathrm{Var}}_t,
$$

$$
\widehat{\mathrm{Var}}_t \leq \frac{3}{2} \sum_{i \in \Psi_t} \frac{\sigma_i^2}{w_i^2} + O\left( \iota'(\delta)(\sigma_\eta^2 + c_\eta) + \Delta_v + \lambda \right). \tag{10}
$$

**Part III: Bounding the regret for each level $l$.** Conditioning on the high-probability events, we can show that for any time step $t \in \Psi_T^l$, the true optimal decision $x_t^\star = \mathrm{argmax}_{x \in \mathcal{X}_t} f^\star(x)$ remains in the candidate set $\mathcal{X}_t^l$ during the level-wise elimination in Algorithm 2, where $l$ is the level from which $x_t$ arises. By the definition of $\mathcal{X}_t^l$, we know that $x_t$ cannot be far from $x_t^\star$, thus we can demonstrate the following lemma.

**Lemma 4.6.** *Under Assumption 4.1 and Algorithm 2, if $f^\star \in \mathcal{F}_{t-1}^l$ and (10) hold for all large enough $t, l$, then, for all large enough $l$ and $t \in \Psi_T^l$, we have:*

$$
x_t^\star \in \mathcal{X}_t^l,
$$

*and the regret at the $l$-th level is bounded by*

$$
\sum_{t \in \Psi_T^l : t \geq \Theta(1)} (f^\star(x_t^\star) - f^\star(x_t)) \leq 2^{-l+3} \hat{\beta}_T^{l-1} \cdot |\Psi_T^l|.
$$

The details and the proof of the lemma are deferred to Appendix C.1. Hence, it suffices to bound the size of $\Psi_T^l$. We get via $D_t^l(x_t)/w_t = 2^{-l}$ that

$$
|\Psi_T^l| = 2^{2l} \cdot \sum_{i \in \Psi_{T,l}} \frac{(D_t^l(x_i))^2}{w_i^2} \leq 2^{2l} \dim_{1,T}(\mathcal{F}).
$$

Ultimately, by combining the results above the choice of $\hat{\beta}_t^{l-1}$ in (8), we can obtain the bound for each $l$. The final regret is obtained by summing the regret for level $l = l_*, \ldots, L$.

## 5. Conclusion

In this work, we consider contextual bandits under heavy-tailed rewards (rewards with a large range $R$) with general function approximation. The key novelty of our approach is the application of Catoni's mean estimator for non-linear settings based on the observation that excess loss estimation is the correct object to robustify. For the known-variance case, the Catoni-OFUL algorithm combines the adaptive Catoni estimator and the variance-weighted optimization. The algorithm enjoys a variance-based regret bound with only polynomial dependence on $R$. When the per-round variance is unknown, our proposed variance-agnostic Catoni

bandit algorithm carefully peels the samples based on their uncertainty and utilizes a plug-in estimator for the sum of variances. The algorithm also obtains a variance-based bound depending on $R$ logarithmically, but has a worse dependence on the eluder dimension. Improving this is left as a future direction. We also provide a lower bound to show that our regret bounds are optimal in the leading-order term.

For the future work, since the Catoni estimator is a general device from robust statistics, it might also be useful to investigate if it enables us to handle other forms of noise, such as adversarial corruption (He et al., 2022; Ye et al., 2023; 2024b). Additionally, while we obtain information-theoretic results in this paper, the algorithms are not easy to implement, both because OFUL-style algorithms are always tricky due to the version space structure, and the function-dependent choice of $\theta$ in the way we invoke the Catoni estimator makes things even harder. It would also be interesting to extend the results to general MDPs.

## Acknowledgments

This work was partially supported by NSF grant No. 2416897 and ONR grant No. N000142512318.

## Impact Statement

This paper presents work with the goal of advancing the field of reinforcement learning with heavy-tailed rewards. The utilization of the Catoni estimator, the variance-weighting technique, and the peeling approach under unknown variances can make online-decision making robust to heavy-tailed rewards and when the rewards have a large range. The work is primarily theoretical, with no immediate social consequences.

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

# A. Notation Table and Additional Related Works

## A.1. Notation Table

To improve the readability of this paper, we provide Tables 2 and 3 for the notations used in this paper.

| Notation | Description |
|---|---|
| $[n]$ | $\{1, \ldots, n\}$ |
| $R$ | The range for the noise $\eta_t, \forall t \in [T]$ |
| $\sigma_t$ | The variance for the noise $\eta_t$ |
| $L_f$ | The range for any function $f \in \mathcal{F}$ |
| $N$ | The $\upsilon$-covering number for the reward function class $\mathcal{F}$ |
| $D_{\mathcal{F}}(x, \sigma; x_{[t-1]}, \sigma_{[t-1]}),$ $D_{\mathcal{F}}^2(x; x_{[t-1]}, \sigma_{[t-1]})$ | The eluder coefficients in Definition 2.2 |
| $\dim(\mathcal{F}, X, \sigma),$ $\dim_{\alpha, T}(\mathcal{F})$ | The eluder dimension in Definition 2.2 |
| $\mathsf{Catoni}_\theta(\{Z_i\}_{i \in [t]})$ | Catoni estimator defined in (2) |
| $\iota_0^2$ | $4 \log \left( \frac{48R(1+2AR)t^2}{\min(1,a)\epsilon^2\delta} \log(A/a) \right)$ |
| $\iota(\delta)$ | $\Theta(\sqrt{\log(RL_f T \mathcal{N}(\mathcal{F}, \upsilon)/\delta)})$ |
| $\theta_t(f, f')$ | $\dfrac{2\iota(\delta)}{\sqrt{\sum_{i \in [t]}(f(x_i)-f'(x_i))^2/\bar{\sigma}_i^2 \cdot \left(1+(2\iota(\delta))^{-1}\sqrt{\hat{\beta}_{t-1}^2+\lambda}\right)+\epsilon^2}}$ |
| $\hat{\beta}_t$ | The confidence radius, $\Theta(\sqrt{\log(RL_f \mathcal{N}(\mathcal{F}, \upsilon)T/\delta)})$ |
| $\alpha$ | $1/\sqrt{T}$ |
| $\lambda$ | $\Theta(1)$ |
| $\upsilon$ | $O(1/L_f^{12}R^4T^{10})$ |

*Table 2.* The Table of Notations for the Known Variance Case.

## A.2. Additional Related Works

**Variance-weighted regression.** Variance-weighted regression has been studied for light-tailed noises for both contextual bandits and Markov Decision Processes (MDPs) with linear and general function approximation. Specifically, Zhou et al. (2021); Zhou & Gu (2022) apply variance-weighted regression to obtain second-order bounds for linear contextual bandits under the known variance case. They also use the weighting technique for linear mixture MDPs under unknown variance case, where they assume that the variance can be realized by a linear function class. Similar weighted regression also appears in MDPs with linear and general function approximation to achieve the optimal regret bound (Agarwal et al., 2023; He et al., 2023; Zhao et al., 2023a), and in the adversarial corruption settings to make the algorithm robust to adversarial attacks (He et al., 2022; Ye et al., 2023; 2024b;a).

**Heavy-tailed rewards in bandits and RL.** The topic of *robustness to heavy-tailed rewards* has received a considerable amount of attention recently. Bubeck et al. (2013) are the first to study heavy-tailed rewards in multi-armed bandits. More generally, robust mean estimators (Lugosi & Mendelson, 2019) such as median-of-means, truncated mean and Catoni's mean have been applied to linear contextual bandits (Medina & Yang, 2016; Shao et al., 2018; Xue et al., 2020; Huang et al., 2024; Li & Sun, 2024).

# B. Proofs for the Known Variance Setting

## B.1. Proof for the Lower Bound

*Proof of Theorem 3.1.* For any $0 \le \sigma \le 1/2, 0 \le \epsilon \le \sigma/2$ and $R > \sqrt{3}$, define three distributions

$$P_\sigma = \sigma(1+R^{-1}), \quad P_{\sigma,\epsilon}^+ = \begin{cases} 2\sigma, & \text{w.p. } \frac{\sigma+\epsilon}{2\sigma}, \\ 2\sigma R, & \text{w.p. } \frac{\sigma+\epsilon}{2\sigma R^2}, \\ 0, & \text{w.p. } 1 - \frac{(\sigma+\epsilon)(1+R^{-2})}{2\sigma}, \end{cases} \quad P_{\sigma,\epsilon}^- = \begin{cases} 2\sigma, & \text{w.p. } \frac{\sigma-\epsilon}{2\sigma}, \\ 2\sigma R, & \text{w.p. } \frac{\sigma-\epsilon}{2\sigma R^2}, \\ 0, & \text{w.p. } 1 - \frac{(\sigma-\epsilon)(1+R^{-2})}{2\sigma}, \end{cases}$$

| Notation | Description |
|---|---|
| $\sigma_\eta^2$ | The union bound for $\sigma_t^2$, $\forall\, t \in [T]$ |
| $c_\eta$ | The union bound of the ratio: $\mathrm{Var}[\eta_t^2|\mathcal{F}_t] \leq c_\eta \mathrm{Var}[\eta_t|\mathcal{F}_t]$ |
| $\Psi_t^l$ | The set of time steps when the update happens for level $l$ in Algorithm 2 |
| $\lambda^l$ | The parameter in the uncertainty $2^{-2l}$ for $l \in [L]$ |
| $l_\star$ | $\lceil \log_2(1076\iota'(\delta)) \rceil$ |
| $D_t^l(x)$ | $\sup_{f,f'\in\mathcal{F}_{t-1}^l} \dfrac{|f(x)-f'(x)|}{\sqrt{\sum_{i\in\Psi_{t-1}^l}(f(x_i)-f'(x_i))^2/w_i^2+\lambda^l}}$ |
| $\hat{b}_t^l$ | $14\iota'(\delta)(2\sigma_\eta^2 + c_\eta) + 43\Delta_v + 268\lambda^l$ |
| $\theta_{\mathrm{Var}}^{t,l}$ | $(4(2\sigma_\eta^2 + c_\eta + L_f^2 + 2^{-2l+4}\cdot(\hat{\beta}_{t-1}^l)^2))^{-1}$ |
| $\Delta_v$ | $\Theta(\mathrm{Poly}(L_f R v T))$ |
| $\iota'(\delta)$ | $\Theta\Big(\sqrt{\log\Big(RL_f(\sigma_\eta^2 + c_\eta + \Delta_v + \lambda^l)NLT/\delta\Big)}\Big)$ |
| $\theta_t^l(f,f')$ | $\dfrac{\iota'(\delta)}{\sqrt{2^{-2l}(\hat{\beta}_{t-1}^l)^2(\widehat{\mathrm{Var}}_t^l + V_t^l(f,f'))+2^{-4l}}}$ |
| $(\hat{\beta}_t^l)^2$ | The confidence radius, $2880(\iota'(\delta))^2 2^{-2l}\widehat{\mathrm{Var}}_t^l + 60\iota'(\delta)2^{-2l} + 12\Delta_{v,2} + 2\lambda^l$ |
| $\Delta_v$ | $\Theta\big(L_f v T^2 + L_f^4 R^2\iota'(\delta)v^{0.5}T^{3.5} + L_f^3 R^{1.5}v^{0.25}T^{1.25} + R^3 L_f^3 v T + \sqrt{RL_f v T}\big)$ |

*Table 3.* The Table of Notations for the Unknown Variance Case.

We have the means

$$\mu_\sigma = \sigma(1 + R^{-1}), \qquad \mu_{\sigma,\epsilon}^+ = (\sigma + \epsilon)(1 + R^{-1}), \qquad \mu_{\sigma,\epsilon}^- = (\sigma - \epsilon)(1 + R^{-1}),$$

and variance

$$
\begin{aligned}
V_\sigma &= 0, \\
V_{\sigma,\epsilon}^+ &= (\sigma + \epsilon)(4\sigma - (1 + R^{-1})^2\sigma - (1 + R^{-1})^2\epsilon) \leq 6\sigma^2, \\
V_{\sigma,\epsilon}^- &= (\sigma - \epsilon)(4\sigma - (1 + R^{-1})^2\sigma + (1 + R^{-1})^2\epsilon) \leq 2\sigma^2.
\end{aligned}
\tag{11}
$$

Thus, the rewards induced by the last two distributions $P_{\sigma,\epsilon}^+$, $P_{\sigma,\epsilon}^-$ have large $L_1$ norm ($2\sigma R$) and bounded variances.

Furthermore, we have

$$
\begin{aligned}
\mathrm{KL}(P_{\sigma,\epsilon}^-\|P_{\sigma,\epsilon}^+) &= \frac{\sigma - \epsilon}{2\sigma}\log\frac{\sigma - \epsilon}{\sigma + \epsilon} + \frac{\sigma - \epsilon}{2\sigma R^2}\log\frac{\sigma - \epsilon}{\sigma + \epsilon} + \left(1 - \frac{(\sigma - \epsilon)(1 + R^{-2})}{2\sigma}\right)\log\frac{\sigma + \epsilon - \frac{\sigma - \epsilon}{R^2}}{\sigma - \epsilon - \frac{\sigma + \epsilon}{R^2}} \\
&\leq (1 + R^{-2})\frac{\sigma - \epsilon}{2\sigma}\log\frac{\sigma - \epsilon}{\sigma + \epsilon} + (1 + R^{-2})\frac{\sigma + \epsilon}{2\sigma}\log\frac{\sigma + \epsilon}{\sigma - \epsilon} \\
&\leq 4(1 + R^{-2})\frac{\epsilon^2}{\sigma^2}.
\end{aligned}
$$

Fix a policy $\pi$. Now, we construct two 2-armed bandits and let the context space $\mathcal{X} = \emptyset$. For the first bandit $B_1$, the reward of the first arm $R_1(a_1) \sim P_\sigma$, and the reward of the second arm $R_1(a_2) \sim P_{\sigma,\epsilon}^-$. Thus, the first arm $a_1$ is the optimal arm for $B_1$. For the second bandit, we have $R_2(a_1) \sim P_\sigma$, and $R_2(a_2) \sim P_{\sigma,\epsilon}^+$, and the second arm is the optimal arm for $B_2$ but with a large variance. For $i = 1, 2$, let $\mathbb{P}_i$ denote the distribution generated by the bandit environment $B_i$, and let $\mathbb{E}_i$ denote the expectations under $\mathbb{P}_i$. Then, we have

$$\mathbb{E}_1[R_T] \geq \mathbb{P}_1(N_T(1) \leq T/2)\cdot\frac{T\epsilon}{2}, \quad \mathbb{E}_2[R_T] \geq \mathbb{P}_2(N_T(1) \geq T/2)\cdot\frac{T\epsilon}{2}.$$

Then, by Bretagnolle-Huber inequality, we have

$$\mathbb{E}_1[R_T] + \mathbb{E}_2[R_T] \geq \frac{T\epsilon}{2}\big(\mathbb{P}_1(N_T(1) \leq T/2) + \mathbb{P}_2(N_T(1) > T/2)\big) \geq \frac{T\epsilon}{4}\exp(-\mathrm{KL}(\mathbb{P}_1\|\mathbb{P}_2)).$$

We also have

$$\mathrm{KL}(\mathbb{P}_1\|\mathbb{P}_2) = \mathbb{E}_1[N_T(2)]\mathrm{KL}(P_{\sigma,\epsilon}^-\|P_{\sigma,\epsilon}^+) \leq \frac{4(1 + R^{-2})\mathbb{E}_1[N_T(2)]\epsilon^2}{\sigma^2}.$$

Thus, we have

$$\mathbb{E}_1[R_T] + \mathbb{E}_2[R_T] \geq \frac{T\epsilon}{4} \exp\left(-\frac{4(1+R^{-2})\mathbb{E}_1[N_T(2)]\epsilon^2}{\sigma^2}\right)$$

By choosing $\epsilon = \sqrt{\sigma^2/4(1+R^{-2})\mathbb{E}_1[N_T(2)]}$, we have

$$\mathbb{E}_1[R_T] + \mathbb{E}_2[R_T] \geq \frac{T}{4e}\sqrt{\frac{\sigma^2}{4(1+R^{-2})\mathbb{E}_1[N_T(2)]}}$$

Since the variance accumulates only when the arm $x_2$ is pulled,

$$T \geq \mathbb{E}_1[N_T(2)] = \mathbb{E}_1\Big[\sum_{t=1}^{T} I(x_t = x_2)\Big] \geq \frac{\mathbb{E}_1[\sum_{t=1}^{T}\sigma_t^2]}{6\sigma^2} = \Omega(\sqrt{\sigma^2\mathbb{E}_1[N_T(2)]}),$$

which implies that

$$\max_i\{\mathbb{E}_i[R_T]\} \geq \Omega\Big(\mathbb{E}_1\sum_{t=1}^{T}\sigma_t^2\Big).$$

Symetrically, we can also get

$$\max_i\{\mathbb{E}_i[R_T]\} \geq \Omega\Big(\mathbb{E}_2\sum_{t=1}^{T}\sigma_t^2\Big).$$

$\square$

### B.2. Proof for Theorem 3.4

**Theorem B.1** (Formal version of Theorem 3.4). *Under Algorithm 1 with the parameter* $\alpha = 1/\sqrt{T}$, $\lambda = \Theta(1)$, $\upsilon = O(1/L_f^{12}R^4T^{10})$ *and*

$$\hat{\beta}_t = \Theta(\sqrt{\log(RL_f\mathcal{N}(\mathcal{F},\upsilon)T/\delta)}), \tag{12}$$

*with probability* $1 - 2\delta$, *we can bound the regret by*

$$R_T = \widetilde{O}\Big(L_f\sqrt{\sum_{t\in[T]}\sigma_t^2 \cdot \dim_{\frac{1}{\sqrt{T}},T}(\mathcal{F}) \cdot \log\mathcal{N}(\mathcal{F},\upsilon)}$$
$$+ L_f \cdot \log\mathcal{N}(\mathcal{F},\upsilon) \cdot \dim_{\frac{1}{\sqrt{T}},T}(\mathcal{F})\Big).$$

**Notations** In the following analysis, we use the short-hand notation for any $f, f' \in \mathcal{F}$

$$V_t(f, f') := \sum_{i\in[t]} \frac{(f(x_i) - f'(x_i))^2}{\bar{\sigma}_i^2}.$$

Recall that we define the excess loss and expected loss: for any $f, f' \in \mathcal{F}$

$$L_t(f, f') = \sum_{i\in[t]} \frac{1}{\bar{\sigma}_i^2}(f'(x_i) - f(x_i))^2 + 2t\mathsf{Catoni}_{\theta_t(f,f')}(\{Z_i(f,f')\}_{i\in[t]}),$$

$$R_t(f, f') = \sum_{i\in[t]} \frac{1}{\bar{\sigma}_i^2}\mathbb{E}_i\big[(f(x_i) - f'(x_i))^2 + 2(f(x_i) - f'(x_i))(f'(x_i) - y_i)\big]$$

$$= \sum_{i\in[t]} \frac{1}{\bar{\sigma}_i^2}\big[(f(x_i) - f'(x_i))^2 + 2(f(x_i) - f'(x_i))(f'(x_i) - f^\star(x_i))\big],$$

where we define

$$Z_i(f, f') = \frac{1}{\bar{\sigma}_i^2}(f(x_i) - f'(x_i))(f'(x_i) - y_i),$$

$$\theta_t(f, f') = \frac{2\iota(\delta)}{\sqrt{V_t(f, f')\left(1 + (2\iota(\delta))^{-1}\sqrt{\hat{\beta}_{t-1}^2 + \lambda}\right) + \epsilon^2}},$$

$$\iota(\delta) = \sqrt{\log\left(\frac{720R^2 L_f^3 N^2 T^5}{\delta}\right)}.$$

We also use the short-hand notation for the covering number $N := \mathcal{N}(\mathcal{F}, \upsilon)$.

**Part I: Concentration of excess loss**   To begin with, we focus on proving the concentration between $L_t(f, f')$ and $R_t(f, f')$. We first consider two fixed functions $f, f' \in \mathcal{F}$.

**Lemma B.2** (Formal Version of Lemma 3.5). *For each time step $t \geq 3\iota^2(\delta)$ and two fixed $f, f' \in \mathcal{F}_{t-1}$, if we take $\alpha = 1/\sqrt{T}$ and $\epsilon = 1$, we have with probability at least $1 - \delta/N^2$,*

$$\left| L_t(f, f') - R_t(f, f') \right| = 2\left| t\mathsf{Catoni}_{\theta_t(f,f')}(\{Z_i(f, f')\}_{i\in[t]}) - \sum_{i\in[t]} \frac{1}{\bar{\sigma}_i^2}(f(x_i) - f'(x_i))(f'(x_i) - f^\star(x_i)) \right|$$

$$\leq \frac{1}{2}V_t(f, f') + 16\iota(\delta)(1 + \frac{\sqrt{\lambda}}{2}) + 32\iota^2(\delta) + 5\iota(\delta) + \frac{1}{6}\hat{\beta}_{t-1}^2.$$

*Proof.* We first compute the expectation of $Z_i(f, f')$ as

$$\mu_i(f, f') = \frac{1}{\bar{\sigma}_i^2}(f(x_i) - f'(x_i))(f'(x_i) - f^\star(x_i)),$$

and bound the sum of variance by

$$\sum_{i\in[t]} \mathrm{Var}[Z_i(f, f')] = \sum_{i\in[t]} \mathbb{E}\left[\frac{1}{\bar{\sigma}_i^4}(f(x_i) - f'(x_i))^2(f^\star(x_i) - y_i)^2\right]$$

$$\leq \sum_{i\in[t]} \frac{(f(x_i) - f'(x_i))^2}{\bar{\sigma}_i^2} = V_t(f, f').$$

We can also bound $\theta_t(f, f') \in [a, A]$ by choosing

$$A = \frac{\iota(\delta)}{\epsilon}, \qquad a = \frac{\iota(\delta)}{\sqrt{8L_f^2 t/\alpha^2 + \epsilon^2}}.$$

Hence, given choice of $\alpha = 1/\sqrt{T}$ and $\epsilon = 1$, we have

$$\log\left(\frac{48R(1 + 2AR)t^2}{\min(1, a)\epsilon^2 \cdot (\delta/N^2 T)}\log(A/a)\right) \leq \log\left(\frac{720R^2 L_f^3 T^4}{\delta/N^2 T}\right) \leq \iota^2(\delta)$$

Thus, for any time step

$$t \geq 6\iota^2(\delta) \geq 4\iota^2(\delta) + 2\log\left(\frac{48R(1 + 2AR)t^2}{\min(1, a)\epsilon^2 \cdot (\delta/N^2 T)}\log(A/a)\right),$$

by using Lemma 3.2 with $\alpha = 1/\sqrt{T}$ and $\epsilon = 1$, we have with probability at least $1 - \delta/N^2 T$,

$$\left| t \mathsf{Catoni}_{\theta_t(f, f')}(\{Z_i(f, f')\}_{i \in [t]}) - \sum_{i \in [t]} \frac{1}{\bar{\sigma}_i^2} (f(x_i) - f'(x_i))(f'(x_i) - f^\star(x_i)) \right|$$

$$\leq \theta_t(f, f') \left( V_t(f, f') + \sum_{i \in [t]} \frac{1}{\bar{\sigma}_i^4} (f(x_i) - f'(x_i))^2 (f'(x_i) - f^\star(x_i))^2 \right) + \frac{4\iota^2(\delta)}{\theta_t(f, f')} + \epsilon$$

$$\leq \theta_t(f, f') V_t(f, f') \left( 1 + \max_{i \in [t]} \frac{1}{\bar{\sigma}_i^2} (f'(x_i) - f^\star(x_i))^2 \right) + \frac{4\iota^2(\delta)}{\theta_t(f, f')} + \epsilon$$

$$\leq \theta_t(f, f') V_t(f, f') \left( 1 + \frac{1}{2\iota(\delta)} \cdot \sqrt{\hat{\beta}_{t-1}^2 + \lambda} \right) + \frac{4\iota^2(\delta)}{\theta_t(f, f')} + \epsilon, \tag{13}$$

where the last inequality uses the definition of the weight $\bar{\sigma}_i^2 \geq 4\iota(\delta) L_f D_f(x_i, x_{[i-1]}, \bar{\sigma}_{[i-1]})$ and $f, f' \in \mathcal{F}_{t-1} \subset \mathcal{F}_{i-1}$ to get for any $i \in [t]$

$$\frac{1}{\bar{\sigma}_i^2} (f'(x_i) - f^\star(x_i))^2 \leq \frac{L_f}{\bar{\sigma}_i^2} \cdot \sup_{f, f' \in \mathcal{F}_{i-1}} \frac{|f'(x_i) - f^\star(x_i)|}{\sqrt{\sum_{k \in [i-1]} \frac{1}{\bar{\sigma}_k^2} (f'(x_k) - f^\star(x_k))^2 + \lambda}} \cdot \sqrt{\sum_{k \in [i-1]} \frac{1}{\bar{\sigma}_k^2} (f'(x_k) - f^\star(x_k))^2 + \lambda}$$

$$\leq \frac{1}{4\iota(\delta)} \sqrt{2 \sum_{k \in [t-1]} \frac{1}{\bar{\sigma}_k^2} (f'(x_k) - \hat{f}_{t-1}(x_k))^2 + 2 \sum_{k \in [i-1]} \frac{1}{\bar{\sigma}_k^2} (f^\star(x_k) - \hat{f}_{t-1}(x_k))^2 + \lambda}$$

$$\leq \frac{1}{2\iota(\delta)} \cdot \sqrt{\hat{\beta}_{t-1}^2 + \lambda},$$

where the second inequality uses the Cauchy-Schwartz inequality, and the last inequality uses the definition of $\mathcal{F}_{i-1}$. Via the choice of

$$\theta_t(f, f') = \frac{2\iota(\delta)}{\sqrt{V_t(f, f')\left(1 + (2\iota(\delta))^{-1}\sqrt{\hat{\beta}_{t-1}^2 + \lambda}\right) + \epsilon^2}},$$

we get the right-hand side of (13) is upper-bounded by

$$4\iota(\delta) \sqrt{V_t(f, f')\left(1 + (2\iota(\delta))^{-1}\sqrt{\hat{\beta}_{t-1}^2 + \lambda}\right) + \epsilon^2} + \epsilon$$

$$\leq 4\iota(\delta) \sqrt{1 + \frac{\sqrt{\lambda}}{2}} \cdot \sqrt{V_t(f, f')} + 2\sqrt{2\iota(\delta)} \cdot \sqrt{V_t(f, f')\hat{\beta}_{t-1}} + 5\iota(\delta)\epsilon$$

$$\leq \frac{1}{4} V_t(f, f') + 16\iota(\delta)(1 + \frac{\sqrt{\lambda}}{2}) + \frac{1}{4} V_t(f, f') + 96\iota^2(\delta) + \frac{1}{6}\hat{\beta}_{t-1}^2 + 5\iota(\delta)$$

$$= \frac{1}{2} V_t(f, f') + 16\iota(\delta)(1 + \frac{\sqrt{\lambda}}{2}) + 96\iota^2(\delta) + 5\iota(\delta) + \frac{1}{6}\hat{\beta}_{t-1}^2,$$

which concludes the proof by taking the union bound over $t \in [T]$. $\qquad \square$

**Lemma B.3.** *For any $f, f' \in \mathcal{F}$, there exist two $f_\upsilon, f'_\upsilon \in \mathcal{F}_\upsilon$ such that $\|f - f_\upsilon\|_\infty \leq \upsilon$, $\|f' - f'_\upsilon\|_\infty \leq \upsilon$. Then, we have*

$$\left| \mathsf{Catoni}_{\theta_t(f, f')}(\{Z_i(f, f')\}_{i \in [t]}) - \mathsf{Catoni}_{\theta_t(f_\upsilon, f'_\upsilon)}(\{Z_i(f_\upsilon, f'_\upsilon)\}_{i \in [t]}) \right|$$

$$\leq \frac{360 L_f^4 R^2 \iota(\delta) \sqrt{\upsilon t}}{\alpha^4 \epsilon^2} + \frac{90 L_f^3 R^{1/2}(\upsilon t)^{1/4}}{\iota(\delta) \alpha^{5/2} \epsilon^{1/2}}.$$

*Proof.* From the definitions of $Z_i(f, f')$, we have $Z_i(f, f') \leq 2L_f(2L_f + R)/\alpha^2$, and

$$|Z_i(f, f') - Z_i(f_v, f'_v)| \leq \left| \frac{1}{\bar{\sigma}_i^2}(f(x_i) - f'(x_i))(f'(x_i) - y_i) - \frac{1}{\bar{\sigma}_i^2}(f_v(x_i) - f'_v(x_i))(f'_v(x_i) - y_i) \right|$$

$$\leq \left| \frac{1}{\bar{\sigma}_i^2}\left(f(x_i) - f_v(x_i) - f'(x_i) + f'_v(x_i)\right)(f'(x_i) - y_i) - \frac{1}{\bar{\sigma}_i^2}\left(f_v(x_i) - f'_v(x_i)\right)(f'_v(x_i) - f(x_i)) \right|$$

$$\leq \frac{6L_f v}{\alpha^2}.$$

From the definitions $\theta_t(f, f')$, we get $\theta_t(f, f') \leq \iota(\delta)/\epsilon$ and

$$|\theta_t(f, f') - \theta_t(f_v, f'_v)| \leq \left| \frac{\iota(\delta)}{\sqrt{\sum_{i \in [t]}(f(x_i) - f'(x_i))^2/\bar{\sigma}_i^2 + \epsilon^2}} - \frac{\iota(\delta)}{\sqrt{\sum_{i \in [t]}(f_v(x_i) - f'_v(x_i))^2/\bar{\sigma}_i^2 + \epsilon^2}} \right|$$

$$\leq \frac{\iota(\delta)}{\epsilon} \cdot \sqrt{\sum_{i \in [t]} \frac{1}{\bar{\sigma}_i^2}\left[(f(x_i) - f'(x_i))^2 - (f_v(x_i) - f'_v(x_i))^2\right]}$$

$$\leq \frac{\iota(\delta)}{\epsilon\alpha} \cdot \sqrt{8L_f vt}.$$

Combining the two inequalities above, we have

$$\Delta = \frac{1}{t}\sum_{i \in [t]} \theta_t(f, f')\left|Z_i(f, f') - Z_i(f_v, f'_v)\right| + \frac{6L_f(2L_f + R)}{\alpha^2} \cdot |\theta_t(f, f') - \theta_t(f_v, f'_v)|$$

$$\leq \frac{\iota(\delta)}{\epsilon} \cdot \frac{6L_f v}{\alpha^2} + \frac{6L_f(2L_f + R)}{\alpha^2} \cdot \frac{\iota(\delta)}{\epsilon\alpha} \cdot \sqrt{8L_f vt}$$

$$\leq \frac{60L_f^{2.5}R\iota(\delta)\sqrt{vt}}{\alpha^3\epsilon}.$$

Then, by invoking Lemma E.2, we deduce that

$$\left| \mathsf{Catoni}_{\theta_t(f, f')}(\{Z_i(f, f')\}_{i \in [t]}) - \mathsf{Catoni}_{\theta_t(f_v, f'_v)}(\{Z_i(f_v, f'_v)\}_{i \in [t]}) \right|$$

$$\leq \frac{1 + 2\theta_t(f, f') \cdot 2L_f(2L_f + R)/\alpha^2}{\theta_t(f, f')} \cdot \Delta + \sqrt{\frac{2\Delta}{(\theta_t(f, f'))^2}}$$

$$\leq \frac{360L_f^4 R^2 \iota(\delta)\sqrt{vt}}{\alpha^4\epsilon^2} + \frac{90L_f^3 R^{1/2}(vt)^{1/4}}{\iota(\delta)\alpha^{5/2}\epsilon^{1/2}}.$$

$\square$

Then, it follows the analysis for any $f, f' \in \mathcal{F}$ which uses the uniform cover.

**Lemma B.4.** *If we take $v = O(1/L_f^{12}R^4T^{10})$, $\alpha = 1/\sqrt{T}$, $\epsilon = 1$, and $\hat{\beta}_t = \Theta(\iota(\delta)(1 + \lambda^{1/4}))$. For any $f, f' \in \mathcal{F}$ and any $t \geq 3\iota^2(\delta)$, with probability at least $1 - \delta$,*

$$\left| L_t(f, f') - R_t(f, f') \right| = 2\left| t\mathsf{Catoni}_{\theta_t(f, f')}(\{Z_i(f, f')\}_{i \in [t]}) - \sum_{i \in [t]} \frac{1}{\bar{\sigma}_i^2}(f(x_i) - f'(x_i))(f'(x_i) - f^\star(x_i)) \right|$$

$$\leq \frac{1}{2}V_t(f, f') + \frac{1}{3}\hat{\beta}_t^2.$$

*Proof.* For any $f, f' \in \mathcal{F}$, there exist two $f_v, f'_v \in \mathcal{F}_v$ such that $\|f - f_v\|_\infty \leq v$, $\|f' - f'_v\|_\infty \leq v$. by invoking Lemma B.3 with $\alpha = 1/\sqrt{T}, \epsilon = 1$, we have

$$\left| \mathsf{Catoni}_{\theta_t(f, f')}(\{Z_i(f, f')\}_{i \in [t]}) - \mathsf{Catoni}_{\theta_t(f_v, f'_v)}(\{Z_i(f_v, f'_v)\}_{i \in [t]}) \right|$$

$$\leq 360L_f^4 R^2 \iota(\delta)v^{0.5}T^{2.5} + \frac{90L_f^3 R^{0.5}v^{0.25}T^{1.5}}{\iota(\delta)} := \Delta_c.$$

Additionally, we get

$$\left| R_t(f, f') - R_t(f_\upsilon, f'_\upsilon) \right|$$

$$= \sum_{i \in [t]} \frac{1}{\hat{\sigma}_i^2} \left| \mathbb{E}_i[(f(x_i) - y_i)^2] - \mathbb{E}_i[(f'(x_i) - y_i)^2] - E_i[(f_\upsilon(x_i) - y_i)^2] + \mathbb{E}_i[(f'_\upsilon(x_i) - y_i)^2] \right|$$

$$\leq 12 L_f \upsilon T^2.$$

Thus, by using Lemma 3.5 with a union bound over $f_\upsilon, f'_\upsilon \in \mathcal{F}_\upsilon$, we have with probability at least $1 - \delta$,

$$\left| L_t(f, f') - R_t(f, f') \right|$$

$$= \left| L_t(f, f') - L_t(f_\upsilon, f'_\upsilon) + L_t(f_\upsilon, f'_\upsilon) - R_t(f_\upsilon, f'_\upsilon) + R_t(f_\upsilon, f'_\upsilon) - R_t(f, f') \right|$$

$$\leq \left| L_t(f_\upsilon, f'_\upsilon) - R_t(f_\upsilon, f'_\upsilon) \right| + 18 L_f \upsilon T^2 + T \Delta_c$$

$$\leq \frac{1}{2} V_t^2(f_\upsilon, f'_\upsilon) + 16\iota(\delta)\left(1 + \frac{\sqrt{\lambda}}{2}\right) + 96\iota^2(\delta) + 5\iota(\delta) + \frac{1}{6}\hat{\beta}_{t-1}^2 + 18 L_f \upsilon T^2 + T \Delta_c$$

$$\leq \frac{1}{2} V_t(f, f') + 4 L_f \upsilon T + 16\iota(\delta)\left(1 + \frac{\sqrt{\lambda}}{2}\right) + 96\iota^2(\delta) + 5\iota(\delta) + \frac{1}{6}\hat{\beta}_{t-1}^2 + 18 L_f \upsilon T^2 + T \Delta_c$$

$$\leq \frac{1}{2} V_t(f, f') + \frac{1}{3}\hat{\beta}_t^2,$$

where the last inequality holds since $\hat{\beta}_{t-1}$ satisfies that

$$\hat{\beta}_{t-1}^2 \geq 6 \left( 16\iota(\delta)\left(1 + \frac{\sqrt{\lambda}}{2}\right) + 96\iota^2(\delta) + 5\iota(\delta) + 24 L_f \upsilon T^2 + T \Delta_c \right).$$

$\square$

### Part II: Sharpness of the confidence set

**Lemma B.5** (Formal version of Lemma 3.6)**.** *If we take $\upsilon = O(1/L_f^{12} R^4 T^{10})$, $\alpha = 1/\sqrt{T}$, $\epsilon = 1$, and under Algorithm 1 with $\hat{\beta}_t = \Theta(\iota(\delta)(1 + \lambda^{1/4}))$, we have for all large enough $t \in [T]$ with probability at least $1 - \delta$,*

$$L_t(\hat{f}_t) \geq \frac{2}{3} V_t(\hat{f}_t, f^\star) - \frac{1}{3}\hat{\beta}_t^2,$$

$$L_t(f^\star) \leq \frac{1}{3}\hat{\beta}_t^2,$$

*where the maximizer for $f'$ is $f'_{\max} = \frac{2}{3}f^\star + \frac{1}{3}\hat{f}_t$.*

*Proof of Lemma 3.6.* By invoking Lemma B.4 with taking minimum over $f' \in \mathcal{F}$ on the both sides of the inequality and $f = \hat{f}_t$, we have with probability at least $1 - \delta$

$$\max_{f' \in \mathcal{F}_{t-1}} L_t(\hat{f}_t, f') \geq \max_{f' \in \mathcal{F}_{t-1}} \left\{ R_t(\hat{f}_t, f') - \frac{1}{2} V_t(\hat{f}_t, f') - \frac{1}{3}\hat{\beta}_t^2 \right\},$$

which implies that

$$L_t(\hat{f}_t) \geq \max_{f' \in \mathcal{F}_{t-1}} \left\{ V_t(\hat{f}_t, f^\star) - V_t(f', f^\star) - \frac{1}{2} V_t(\hat{f}_t, f') - \frac{1}{3}\hat{\beta}_t^2 \right\}$$

$$= \max_{f' \in \mathcal{F}_{t-1}} \left\{ -\frac{3}{2} \sum_{i \in [t]} \frac{(f'(x_i) - \frac{2}{3}f^\star(x_i) - \frac{1}{3}\hat{f}_t(x_i))^2}{w_i^2} + \frac{2}{3} V_t(\hat{f}_t, f^\star) - \frac{1}{3}\beta_t^2 \right\}$$

$$= \frac{2}{3} V_t(\hat{f}_t, f^\star) - \frac{1}{3}\hat{\beta}_t^2,$$

where the minimizer is $f'_{\max} = \frac{2}{3}f^\star + \frac{1}{3}\hat{f}_t$.

Additionally, by using Lemma 3.5 with taking minimum over $f' \in \mathcal{F}$ on the both sides of the inequality and $f = f^\star$, we have with probability at least $1 - \delta$

$$
\begin{aligned}
L_t(f^\star) &\leq \max_{f' \in \mathcal{F}_{t-1}} \left\{ R_t(f^\star, f') + \frac{1}{2}V_t(f', f^\star) + \frac{1}{3}\hat{\beta}_t^2 \right\} \\
&= \max_{f' \in \mathcal{F}_{t-1}} \left\{ -V_t(f', f^\star) + \frac{1}{2}V_t(f', f^\star) + \frac{1}{3}\hat{\beta}_t^2 \right\} \\
&\leq \frac{1}{3}\hat{\beta}_t^2,
\end{aligned}
$$

where the minimizer of $f' = f^\star$. $\qquad\square$

**Lemma B.6.** *If we take $\upsilon = O(1/L_f^{12}R^4T^{10})$, $\alpha = 1/\sqrt{T}$, $\epsilon = 1$, and under Algorithm 1 with $\hat{\beta}_t = \Theta(\iota(\delta)(1 + \lambda^{1/4}))$, with probability at least $1 - \delta$, we have $f^\star \in \mathcal{F}_t$.*

*Proof.* We use the notation $L_t(f) := \max_{f' \in \mathcal{F}} L_t(f, f')$, and recall that

$$
\begin{aligned}
R_t(f, f') &= \sum_{i \in [t]} \frac{1}{w_i^2} \mathbb{E}_i \left[ (f(x_i) - f^\star(x_i) + f^\star(x_i) - y_i)^2 - (f'(x_i) - f^\star(x_i) + f^\star(x_i) - y_i)^2 \right] \\
&= \sum_{i \in [t]} \frac{1}{w_i^2} \mathbb{E}_i \left[ (f(x_i) - y_i)^2 + (f^\star(x_i) - y_i)^2 - (f'(x_i) - y_i)^2 - (f^\star(x_i) - y_i)^2 \right] \\
&= \sum_{i \in [t]} \frac{1}{w_i^2} \mathbb{E}_i \left[ (f(x_i) - y_i)^2 - (f'(x_i) - y_i)^2 \right] \\
&= V_t(f, f^\star) - V_t(f', f^\star).
\end{aligned}
$$

Since $\hat{f}_t = \operatorname{argmin}_{f \in \mathcal{F}_{t-1}} L_t(f)$, we invoke Lemma B.5 to get

$$
0 \geq L_t(\hat{f}_t) - L_t(f^\star) \geq \frac{2}{3}V_t(\hat{f}_t, f^\star) - \frac{2}{3}\hat{\beta}_t^2,
$$

which means that

$$
V_t(\hat{f}_t, f^\star) = \sum_{i \in [t]} \frac{1}{\bar{\sigma}_i^2} (\hat{f}_t(x_i) - f^\star(x_i))^2 \leq \hat{\beta}_t^2.
$$

$\qquad\square$

**Part III: Bounding the regret conditioning on good events.** We now recall the definition that $\mathcal{T} := \{t \in [T] : t > 7\iota^2(\delta_{n,T})\}$, we further denote the good events $\mathcal{E}_0 = \{f^\star \in \cap_{t \in \mathcal{T}} \mathcal{F}_t\}$.

*Proof of Theorem 3.4.* Conditioning on both good events $\mathcal{E}_0$, we use the notation $f_t(x) = \operatorname{argmax}_{f \in \mathcal{F}_{t-1}} f(x)$ can bound the regret of $t \in \mathcal{T}$ by

$$
\begin{aligned}
&\max_{x \in \mathcal{X}_t} f^\star(x) - f^\star(x_t) \\
&\quad \leq \max_{x \in \mathcal{X}_t} f_t(x) - f^\star(x_t) \leq f_t(x_t) - f^\star(x_t) \\
&\quad \leq \bar{\sigma}_t D_{\mathcal{F}_{t-1}}(x_t, \bar{\sigma}_t; x_{[t-1]}, \bar{\sigma}_{[t-1]}) \cdot \sqrt{ \sum_{i \in [t-1]} \frac{1}{\bar{\sigma}_i^2} \left( f_t(x_i) - \hat{f}_{t-1}(x_i) + \hat{f}_{t-1}(x_i) - f^\star(x_i) \right)^2 + \lambda } \\
&\quad \leq \bar{\sigma}_t D_{\mathcal{F}_{t-1}}(x_t, \bar{\sigma}_t; x_{[t-1]}, \bar{\sigma}_{[t-1]}) \cdot \sqrt{ 2 \sum_{i \in [t-1]} \frac{1}{\bar{\sigma}_i^2} \left( f_t(x_i) - \hat{f}_{t-1}(x_i) \right)^2 + 2 \sum_{i \in [t-1]} \frac{1}{\bar{\sigma}_i^2} \left( \hat{f}_{t-1}(x_i) - f^\star(x_i) \right)^2 + \lambda } \\
&\quad \leq 2\bar{\sigma}_t D_{\mathcal{F}_{t-1}}(x_t, \bar{\sigma}_t; x_{[t-1]}, \bar{\sigma}_{[t-1]}) \cdot \hat{\beta}_{t-1},
\end{aligned}
$$

where the second inequality follows from $X_t = \operatorname{argmax}_{x \in \mathcal{X}_t} f_t(x)$, the second inequality uses the definition of uncertainty $D_{\mathcal{F}_{t-1}}$, and the last inequality holds due to $f^\star, \hat{f}_{t-1} \in \mathcal{F}_{t-1}$.

Combining this with the range bound that $\|f^\star\|_\infty \leq L_f$, the cumulative regret is bounded by

$$R_T = \sum_{t \in [T]} (\max_{x \in \mathcal{X}_t} f^\star(x) - f^\star(x_t))$$
$$\leq 2L_f \cdot 7\iota^2(\delta) + 2L_f \sum_{t-1 \in \mathcal{T}} \min\left(1, \bar{\sigma}_t D_{\mathcal{F}_{t-1}}(x_t, \bar{\sigma}_t; x_{[t-1]}, \bar{\sigma}_{[t-1]})\hat{\beta}_{t-1}\right). \tag{14}$$

To finally bound the regret, we bound the second term in RHS of $R_T$ expression in (14) respectively. These steps mainly follow Lemma 4.4 in (Zhou & Gu, 2022). We can decompose the terms by considering $\mathcal{I}_1 = \{t - 1 \in \mathcal{T} | D_{\mathcal{F}_{t-1}}(x_t, \bar{\sigma}_t; x_{[t-1]}, \bar{\sigma}_{[t-1]}) \geq 1\}$ and $\mathcal{I}_2 = \{t - 1 \in \mathcal{T}, t \notin \mathcal{I}_1\}$.

For the first set, we bound its size naively by

$$|\mathcal{I}_1| \leq \sum_{t \in \mathcal{I}_1} \min\left(1, D^2_{\mathcal{F}_{t-1}}(x_t, \bar{\sigma}_t; x_{[t-1]}, \bar{\sigma}_{[t-1]})\right) \leq \dim_{\alpha,T}(\mathcal{F}).$$

For the second set, we bound the summation of terms of interest contraining on $\mathcal{I}_2$ by

$$\sum_{t \in \mathcal{I}_2} \bar{\sigma}_t \sqrt{\hat{\beta}_{t-1}^2 + \lambda} \cdot D_{\mathcal{F}_{t-1}}(x_t, \bar{\sigma}_t; x_{[t-1]}, \bar{\sigma}_{[t-1]})$$
$$\leq \sum_{t \in \mathcal{I}_2, \bar{\sigma}_t = \sigma_t \text{ or } \alpha} \bar{\sigma}_t \sqrt{\hat{\beta}_{t-1}^2 + \lambda} \cdot D_{\mathcal{F}_{t-1}}(x_t, \bar{\sigma}_t; x_{[t-1]}, \bar{\sigma}_{[t-1]})$$
$$+ \sum_{t \in \mathcal{I}_2, \bar{\sigma}_t = \sqrt{4\iota(\delta_{n,t,l})L_f D_{\mathcal{F}_{t-1}}(x_t; x_{[t-1]}, \bar{\sigma}_{[t-1]})}} \bar{\sigma}_t \sqrt{\hat{\beta}_{t-1}^2 + \lambda} \cdot D_{\mathcal{F}_{t-1}}(x_t, \bar{\sigma}_t; x_{[t-1]}, \bar{\sigma}_{[t-1]})$$
$$\overset{(i)}{\leq} \sum_{t \in [T]} (\sigma_t + \alpha) \hat{\beta}_{t-1} \cdot D_{\mathcal{F}_{t-1}}(x_t, \bar{\sigma}_t; x_{[t-1]}, \bar{\sigma}_{[t-1]}) + \sum_{t \in [T]} 32 L_f \iota(\delta_{n,t,l}) \hat{\beta}_{t-1} \cdot D^2_{\mathcal{F}_{t-1}}(x_t, \bar{\sigma}_t; x_{[t-1]}, \bar{\sigma}_{[t-1]})$$
$$\overset{(ii)}{\leq} \sqrt{2 \sum_{t \in [T]} \hat{\beta}_{t-1}^2 (\sigma_t^2 + \alpha^2)} \sqrt{\dim_{\alpha,T}(\mathcal{F})} + 16 L_f \iota(\delta_{n,t,l}) \max_{t \in [T]} \hat{\beta}_{t-1} \cdot \dim_{\alpha,T}(\mathcal{F}).$$

Here for $(i)$ we use the condition for each distinct set and for $(ii)$ we use Cauchy-Schwarz inequality for the first term and the definition of $\dim_\alpha$ for both terms.

Consequently plugging these back in (14) and take supremum over $x : |x| = T$, we conclude that with probability at least $1 - 2\delta$,

$$R_T = O\left(L_f \cdot \iota^2(\delta_{n,T}) + L_f \dim_{\alpha,T}(\mathcal{F}) + L_f^2 \cdot \iota(\delta_{n,t,l}) \cdot \max_{t \in [T]} \hat{\beta}_{t-1} \cdot \dim_{\alpha,T}(\mathcal{F})\right.$$
$$\left. L_f + \sqrt{\sum_{t \in [T]} \hat{\beta}_{t-1}^2 (\sigma_t^2 + \alpha^2)} \cdot \sqrt{\dim_{\alpha,T}(\mathcal{F})}\right)$$
$$= \widetilde{O}\left(L_f \cdot \log\mathcal{N}(\mathcal{F}, \upsilon) \cdot \dim_{\frac{1}{\sqrt{T}},T}(\mathcal{F}) + L_f \sqrt{\sum_{t \in [T]} \sigma_t^2} \cdot \sqrt{\dim_{\frac{1}{\sqrt{T}},T}(\mathcal{F}) \cdot \log\mathcal{N}(\mathcal{F}, \upsilon)}\right),$$

where for the last inequality we pick $\lambda = \Theta(1)$, $\alpha = 1/\sqrt{T}$ and $\upsilon = O(1/L_f^{12} R^4 T^{10})$. $\qquad \square$

# C. Proofs for Unknown Variance

## C.1. Proof of Theorem 4.2

**Theorem C.1** (Formal Version of Theorem 4.2). *Suppose that Assumption 4.1 holds. Under Algorithm 2 with $\gamma = 1/(\sigma_\eta T^{3/2})$ and $\iota'(\delta) = \Theta(\sqrt{\log\left(RL_f(\sigma_\eta^2 + c_\eta + \Delta_\upsilon + \lambda^l)NLT/\delta\right)})$, if $T \geq 14(\iota'(\delta))^2$, with probability at least $1 - 3\delta$, we can bound the regret by*

$$R_T = \widetilde{O}\Big( L_f \Big( \sum_{t \in [T]} \sigma_t^2 \cdot \log \mathcal{N}(\mathcal{F}, \upsilon) \Big)^{1/2} \cdot \dim_{1,T}(\mathcal{F})$$
$$+ L_f \dim_{1,T}(\mathcal{F})(\log \mathcal{N}(\mathcal{F}, \upsilon))^{3/4}(\sqrt{c_\eta} + \sigma_\eta) \Big).$$

**Notations**    In the following analysis, we use the short-hand notation for any $f, f' \in \mathcal{F}$

$$V_t^l(f, f') := \sum_{i \in \Psi_t^l} \frac{(f(x_i) - f'(x_i))^2}{w_i^2}.$$

Recall that we define the excess loss and expected loss: for each $t \in [T]$, $l \in [L]$, and any $f, f' \in \mathcal{F}$

$$L_t^l(f, f') = \sum_{i \in \Psi_t^l} \frac{1}{w_i^2}(f'(x_i) - f(x_i))^2 + 2t\mathsf{Catoni}_{\theta_t(f,f')}(\{Z_i(f, f')\}_{i \in \Psi_t^l}),$$

$$R_t^l(f, f') = \sum_{i \in \Psi_t^l} \frac{1}{w_i^2}\mathbb{E}_i\big[(f(x_i) - f'(x_i))^2 + 2(f(x_i) - f'(x_i))(f'(x_i) - y_i)\big]$$

$$= \sum_{i \in \Psi_t^l} \frac{1}{w_i^2}\big[(f(x_i) - f'(x_i))^2 + 2(f(x_i) - f'(x_i))(f'(x_i) - f^\star(x_i))\big],$$

where we define

$$Z_i(f, f') = \frac{1}{w_i^2}(f(x_i) - f'(x_i))(f'(x_i) - y_i),$$

$$\theta_t^l(f, f') = \frac{\iota'(\delta)}{\sqrt{2^{-2l}(\hat{\beta}_{t-1}^2 + \lambda)(\overline{\mathrm{Var}}_t^l + V_t^l(f, f')) + \epsilon^2}},$$

$$\iota'(\delta) = \Theta\Big(\sqrt{\log\left(RL_f(\sigma_\eta^2 + c_\eta + \Delta_\upsilon + \lambda^l)NLT/\delta\right)}\Big).$$

The proof is decomposed into four main parts.

In the following parts, we will show that the following two events both hold with a high probability:

$$\mathcal{E}_{\mathrm{var}}^t := \left\{ \sum_{i \in \Psi_t^l} \frac{\sigma_i^2}{w_i^2} \leq 2\widehat{\mathrm{Var}}_t^l, \text{ for } l \in [L], \ 2^l \geq 1076\iota'(\delta) \right\}, \tag{15}$$

$$\mathcal{E}_{\mathrm{conv}}^t := \left\{ f^\star \in \mathcal{F}_t^l, \text{ for } l \in [L], \ 2^l \geq 1076\iota'(\delta) \right\}.$$

We will prove the events hold conditioned on each other sequentially for $t = \Theta(1), \ldots, T$. We also use the short-hand notation for the covering number $N := \mathcal{N}(\mathcal{F}, \upsilon)$.

**Part I: Concentration of excess loss**    First of all, we also need to prove the concentration of excess loss for each $l \in [L]$.

In the following lemma, for conciseness, we neglect level $l$ in the concentration analysis. Later, we will apply the result for each $l \in [L]$.

**Lemma C.2** (Formal version of 4.4). *Under Assumption 4.1, given $\{x_i, y_i, w_i\}_{i \in [t]}$, we define for $i \in [t-1]$, $\mathcal{F}_i = \{f \in \mathcal{F}_{i-1} : \sum_{l \in [i]} (f(x_l) - \hat{f}_i(x_i))^2 / w_l^2 + \lambda \leq \hat{\beta}_i^2\}$, and $D_i = \sup_{f, f' \in \mathcal{F}_{i-1}} \frac{|f(x_i) - f'(x_i)|}{\sqrt{\sum_{l=1}^{i-1} (f(x_l) - f'(x_l))^2 / w_l^2 + \lambda}}$. Suppose that $\sup_{i \in [t]} \frac{D_i}{w_i} \leq \rho$, $w_i \geq 1$, $(\iota'(\delta))^2 \geq \Theta\left(\log\left(RL_f(\sigma_\eta^2 + c_\eta + \Delta_v + \lambda^l)NLT/\delta \cdot \hat{\beta}_{t-1}\right)\right)$ and $\rho \leq \min\{1, 1/(16\sqrt{3}\iota'(\delta))\}$.*

*Then, for the time step $t$ such that the following event happens $\mathcal{E}_t = \left\{t \geq 4(\iota'(\delta))^2 \frac{\sum_{i \in [t]} \sigma_i^2 / w_i^2}{\widehat{\text{Var}}_t} + 6(\iota'(\delta))^2\right\}$, and for any $f, f' \in \mathcal{F}_{t-1}$, if we take $\epsilon = \rho^2$, with probability at least $1 - \delta/TL$,*

$$\left|L_t(f, f') - R_t(f, f')\right| \leq \frac{1}{2} V_t(f, f') + \frac{1}{3}\left(\frac{1}{2}\hat{\beta}_{t-1}^2 + 6\left(48(\iota'(\delta))^2 \rho^2 \left(\frac{\left(\sum_{i \in [t]} \sigma_i^2 / w_i^2\right)^2}{\widehat{\text{Var}}_t} + \widehat{\text{Var}}_t\right) + 5\iota'(\delta)\rho^2 + \Delta_{v,2}\right)\right),$$

*where $\widehat{\text{Var}}_t = t\text{Catoni}_{\theta_{\text{Var}}^t}\left(\left\{\frac{1}{w_i^2}(y_i - \hat{f}_{t-1}(x_i))^2\right\}_{i \in [t]}\right) + 14\iota'(\delta)(2\sigma_\eta^2 + c_\eta) + 43\Delta_v + 268\lambda$, and $\Delta_{v,2} = \Theta(L_f \upsilon T^2 + L_f^4 R^2 \iota'(\delta)\upsilon^{0.5}T^{3.5} + L_f^3 R^{1.5}\upsilon^{0.25}T^{1.25})$, $\theta_{\text{Var}}^t = (4(2\sigma_\eta^2 + c_\eta + L_f^2 + 16\rho^2 \cdot \hat{\beta}_{t-1}^2))^{-1}$.*

*Proof.* At each time step $t \in [T]$, for two fixed $f, f' \in \mathcal{F}_{t-1}$, we first compute the expectation of $Z_i(f, f')$ as

$$\mu_i(f, f') = \frac{1}{w_i^2}(f(x_i) - f'(x_i))(f'(x_i) - f^\star(x_i)).$$

Additionally, we deduce that

$$\frac{(f(x_i) - f'(x_i))^2}{w_i^2} \leq \frac{1}{w_i^2} \cdot \sup_{f, f' \in \mathcal{F}_{i-1}} \frac{|f(x_i) - f'(x_i)|^2}{\sum_{l \in [i-1]} (f(x_l) - f'(x_l))^2 / w_l^2 + \lambda} \cdot \left(\sum_{l \in [i-1]} \frac{(f(x_l) - f'(x_l))^2}{w_l^2} + \lambda\right)$$

$$\leq \frac{D_i^2}{w_i^2} \cdot \left(2\sum_{l \in [t-1]} \frac{(f(x_l) - \hat{f}_{t-1}(x_l))^2}{w_l^2} + 2\sum_{l \in [i-1]} \frac{(f'(x_l) - \hat{f}_{t-1}(x_l))^2}{w_l^2} + \lambda\right)$$

$$\leq 4\rho^2 \hat{\beta}_{t-1}^2,$$

where the first inequality uses $f, f' \in \mathcal{F}_{t-1} \subset \mathcal{F}_{i-1}$, the second inequality uses the definition of $D_i$ and the Cauchy-Schwarz inequality, and the last inequality follows from $D_i / w_i \leq \rho$ and $f, f' \in \mathcal{F}_{t-1}$. Thus, we bound the sum of variance by

$$\sum_{i \in [t]} \text{Var}[Z_i(f, f')] = \sum_{i \in [t]} \mathbb{E}\left[\frac{1}{w_i^4}(f(x_i) - f'(x_i))^2 (f^\star(x_i) - y_i)^2 \,\Big|\, x_i\right]$$

$$\leq \sum_{i \in [t]} \frac{(f(x_i) - f'(x_i))^2}{w_i^2} \cdot \frac{\sigma_i^2}{w_i^2}$$

$$\leq 4\rho^2 \hat{\beta}_{t-1}^2 \sum_{i \in [t]} \frac{\sigma_i^2}{w_i^2}.$$

Similarly, we can bound the sum of $\mu_i^2$ by

$$\sum_{i \in [t]} \mu_i^2(f, f') \leq \sum_{i \in [t]} \frac{1}{w_i^4}(f(x_i) - f'(x_i))^2 (f'(x_i) - f^\star(x_i))^2$$

$$\leq V_t(f, f') \max_{i \in [t]} \frac{(f'(x_i) - f^\star(x_i))^2}{w_i^2}$$

$$\leq V_t(f, f') \max_{i \in [t]} \frac{D_i^2}{w_i^2} \cdot \left(\sum_{l \in [t-1]} (f'(x_l) - f^\star(x_l))^2 / w_l^2 + \lambda\right)$$

$$\leq V_t(f, f') \cdot 4\rho^2 \hat{\beta}_{t-1}^2.$$

We can also get the upper and lower bound of $\theta_t(f, f')$:

$$\theta_t(f, f') = \frac{\iota'(\delta)}{\sqrt{\rho^2 \hat{\beta}_{t-1}^2 (\widehat{\mathrm{Var}}_t + V_t(f, f')) + \epsilon^2}} \leq \frac{\iota'(\delta)}{\epsilon} := A.$$

Besides, by using Claim A.14 from Wagenmaker et al. (2022), we know that

$$\mathsf{Catoni}_{\theta_{\mathrm{Var}}^t}\left(\left\{\frac{1}{w_i^2}(y_i - \hat{f}_{t-1}(x_i))^2\right\}_{i \in [t]}\right) \leq \max_{i \in [t]} \frac{1}{w_i^2}(y_i - \hat{f}_t(x_i))^2 \leq \max_{i \in [t]}\left(2\eta_i^2 + (f^\star(x_i) - \hat{f}_t(x_i))^2\right) \leq 2R^2 + 4L_f^2,$$

which indicates the lower bound of $\theta_t(f, f')$:

$$\theta_t(f, f') \geq \frac{\iota'(\delta)}{\sqrt{(T(2R^2 + 8L_f^2) + 17\iota'(\delta)(2\sigma_\eta^2 + c_\eta) + 43\Delta_v + 268\lambda^l)\rho^2 \hat{\beta}_{t-1}^2 + \rho^4}} := a$$

Hence, given choice of $\epsilon = \rho^2$, we have

$$\log\left(\frac{48R(1 + 2AR)t^2}{\min(1, a)\epsilon^2 \cdot (\delta/N^2TL)} \log(A/a)\right)$$

$$\leq \log\left(\frac{144R^2T^2}{\rho^2\delta/N^2TL} \cdot \left(\frac{A}{a}\right)^2\right)$$

$$\leq \log\left(\frac{144R^2T^2}{\rho^2\delta/N^2TL} \cdot \left((T(2R^2 + 8L_f^2) + 17\iota'(\delta)(2\sigma_\eta^2 + c_\eta) + 43\Delta_v + 268\lambda^l)\rho^2 \hat{\beta}_{t-1}^2 + \rho^4\right)\right)$$

$$\leq (\iota'(\delta))^2,$$

where the last inequality holds since

$$(\iota'(\delta))^2 \geq \Theta\left(\log\left(RL_f(\sigma_\eta^2 + c_\eta + \Delta_v + \lambda^l)NLT/\delta \cdot \hat{\beta}_{t-1}\right)\right).$$

Thus, since the following condition holds for time step

$$\mathcal{E}_t = \left\{t \geq 4(\iota'(\delta))^2 \frac{\sum_{i \in [t]} \frac{\sigma_i^2}{w_i^2}}{\widehat{\mathrm{Var}}_t} + 6(\iota'(\delta))^2\right\},$$

we have

$$t \geq 4(\iota'(\delta))^2 \frac{\sum_{i \in [t]} \frac{\sigma_i^2}{w_i^2} + V_t(f, f')}{\widehat{\mathrm{Var}}_t + V_t(f, f')} + 2\log\left(\frac{48R(1 + 2AR)t^2}{\min(1, a)\epsilon^2 \cdot (\delta/N^2TL)} \log(A/a)\right),$$

by combining the results above and using Lemma 3.2 with $\epsilon = \rho^2$ and the choice of $\theta_t(f, f')$, we have with probability at

least $1 - \delta/N^2 TL$,

$$\left| t\mathsf{Catoni}_{\theta_t(f,f')}(\{Z_i(f,f')\}_{i\in[t]}) - \sum_{i\in[t]} \frac{1}{w_i^2}(f(x_i) - f'(x_i))(f'(x_i) - f^\star(x_i)) \right|$$

$$\leq \theta_t(f,f')\left( \sum_{i\in[t]} \mathrm{Var}[Z_i(f,f')] + \sum_{i\in[t]} \mu_i^2(f,f') \right) + \frac{4(\iota'(\delta))^2}{\theta_t(f,f')} + \rho^2$$

$$\leq \frac{\iota'(\delta)}{\sqrt{\rho^2\hat{\beta}_{t-1}^2(\widehat{\mathrm{Var}}_t + V_t(f,f')) + \epsilon^2}} \cdot 4\rho^2\hat{\beta}_{t-1}^2 \left( \sum_{i\in[t]} \frac{\sigma_i^2}{w_i^2} + V_t(f,f') \right)$$

$$\quad + 4\iota'(\delta)\sqrt{\rho^2\hat{\beta}_{t-1}^2(\widehat{\mathrm{Var}}_t + V_t(f,f')) + \epsilon^2} + 1$$

$$\leq 4\iota'(\delta)\rho\hat{\beta}_{t-1}\left( \frac{\sum_{i\in[t]}\sigma_i^2/w_i^2}{\sqrt{\widehat{\mathrm{Var}}_t}} + \sqrt{\widehat{\mathrm{Var}}_t} + 2\sqrt{V_t(f,f')} \right) + 4\iota'(\delta)\rho^2 + \rho^2$$

$$\leq \frac{1}{12}\hat{\beta}_{t-1}^2 + 48(\iota'(\delta))^2\rho^2 \left( \frac{(\sum_{i\in[t]}\sigma_i^2/w_i^2)^2}{\widehat{\mathrm{Var}}_t} + \widehat{\mathrm{Var}}_t + 4V_t(f,f') \right) + 5\iota'(\delta)\rho^2$$

$$\leq \frac{1}{4}V_t(f,f') + \frac{1}{12}\hat{\beta}_{t-1}^2 + 48(\iota'(\delta))^2\rho^2 \left( \frac{(\sum_{i\in[t]}\sigma_i^2/w_i^2)^2}{\widehat{\mathrm{Var}}_t} + \widehat{\mathrm{Var}}_t \right) + 5\iota'(\delta)\rho^2, \tag{16}$$

where the fourth inequality uses the Cauchy-Schwarz inequality and $\lambda = O(1)$, and the last inequality holds due to the condition that $\rho \leq 1/16\sqrt{3}\iota'(\delta)$.

Then, for any $f, f' \in \mathcal{F}_{t-1}$, there exist two $f_\upsilon, f'_\upsilon \in \mathcal{F}_\upsilon$ such that $\|f - f_\upsilon\|_\infty \leq \upsilon$, $\|f' - f'_\upsilon\|_\infty \leq \upsilon$. Similar to Lemma B.3, we have

$$\left| \mathsf{Catoni}_{\theta_t(f,f')}(\{Z_i(f,f')\}_{i\in[t]}) - \mathsf{Catoni}_{\theta_t(f_\upsilon,f'_\upsilon)}(\{Z_i(f_\upsilon,f'_\upsilon)\}_{i\in[t]}) \right|$$
$$\leq 360L_f^4 R^2\iota'(\delta)\upsilon^{0.5}T^{0.5} + 90L_f^3 R^{0.5}\upsilon^{0.25}T^{0.25} := \Delta_{\upsilon,1}.$$

Additionally, we get

$$\left| R_t(f,f') - R_t(f_\upsilon,f'_\upsilon) \right|$$

$$= \sum_{i\in[t]} \frac{1}{\bar{\sigma}_i^2} \left| \mathbb{E}_i[(f(x_i) - y_i)^2] - \mathbb{E}_i[(f'(x_i) - y_i)^2] - E_i[(f_\upsilon(x_i) - y_i)^2] + \mathbb{E}_i[(f'_\upsilon(x_i) - y_i)^2] \right|$$

$$\leq 12L_f\upsilon T^2.$$

Thus, by using (16) with a union bound over $f_\upsilon, f'_\upsilon \in \mathcal{F}_\upsilon$, we have with probability at least $1 - \delta/TL$,

$$\left| L_t(f,f') - R_t(f,f') \right|$$

$$= \left| L_t(f,f') - L_t(f_\upsilon,f'_\upsilon) + L_t(f_\upsilon,f'_\upsilon) - R_t(f_\upsilon,f'_\upsilon) + R_t(f_\upsilon,f'_\upsilon) - R_t(f,f') \right|$$

$$\leq \left| L_t(f_\upsilon,f'_\upsilon) - R_t(f_\upsilon,f'_\upsilon) \right| + 24L_f\upsilon T^2 + T\Delta_{\upsilon,1}$$

$$\leq \frac{1}{4}V_t(f,f') + \frac{1}{12}\hat{\beta}_{t-1}^2 + 48(\iota'(\delta))^2\rho^2 \left( \frac{(\sum_{i\in[t]}\sigma_i^2/w_i^2)^2}{\widehat{\mathrm{Var}}_t} + \widehat{\mathrm{Var}}_t \right) + 5\iota'(\delta)\rho^2 + \Delta_{\upsilon,2}$$

$$\leq \frac{1}{4}V_t(f,f') + \frac{1}{6}\beta_t^2,$$

where we define $\Delta_{\upsilon,2} = 24L_f\upsilon T^2 + T\Delta_{\upsilon,1}$ the last inequality holds due to the definition of $\beta_t$. $\qquad \square$

**Lemma C.3.** *Under the conditions of Lemma 4.4, assume that $f^\star \in \mathcal{F}_{t-1}$, and the estimator $\hat{f}_t$ is*

$$\hat{f}_t = \underset{\hat{f} \in \mathcal{F}_{t-1}}{\text{argmin}} \max_{f' \in \mathcal{F}_{t-1}} L_t(\hat{f}, f') := \sum_{i \in [t]} \frac{1}{\bar{\sigma}_i^2} (f'(x_i) - \hat{f}(x_i))^2 + 2t\text{Catoni}_{\theta_t(\hat{f}, f')}(\{Z_i(\hat{f}, f')\}_{i \in [t]}).$$

*Then, for the time step $t$ such that the following event happens*

$$\mathcal{E}_t = \left\{ t \geq 4(\iota'(\delta))^2 \frac{\sum_{i \in [t]} \frac{\sigma_i^2}{w_i^2}}{\widehat{\text{Var}}_t} + 6(\iota'(\delta))^2 \right\},$$

*and all $l \in [L]$, if we take $\epsilon = 2^{-2l}$, with probability at least $1 - \delta/TL$,*

$$\sum_{i \in [t]} \frac{(f^\star(x_i) - \hat{f}_t(x_i))^2)}{w_i^2} + \lambda \leq \frac{1}{2}\hat{\beta}_{t-1}^2 + 6\left( 48(\iota'(\delta))^2 \rho^2 \left( \frac{\left(\sum_{i \in [t]} \sigma_i^2/w_i^2\right)^2}{\widehat{\text{Var}}_t} + \widehat{\text{Var}}_t \right) + 5\iota'(\delta)\rho^2 + \Delta_{v,2} \right) + \lambda := \beta_t^2.$$

*Proof.* Let $L_t(f) := \max_{f' \in \mathcal{F}} L_t(f, f')$ and

$$(\beta_t')^2 = \frac{1}{2}\hat{\beta}_{t-1}^2 + 6\left( 48(\iota'(\delta))^2 \rho^2 \left( \frac{\left(\sum_{i \in [t]} \sigma_i^2/w_i^2\right)^2}{\widehat{\text{Var}}_t} + \widehat{\text{Var}}_t \right) + 5\iota'(\delta)\rho^2 + \Delta_{v,2} \right)$$

Thus, by invoking Lemma 4.4 with taking minimum over $f' \in \mathcal{F}_{t-1}$ on the both sides of the inequality and $f = \hat{f}_t$, we have with probability at least $1 - \delta$

$$\begin{aligned}
L_t(\hat{f}_t) &\geq \max_{f' \in \mathcal{F}_{t-1}} \left\{ R_t(\hat{f}_t, f') - \frac{1}{2}V_t(\hat{f}_t, f') - \frac{1}{3}(\beta_t')^2 \right\} \\
&\geq \max_{f' \in \mathcal{F}_{t-1}} \left\{ V_t(\hat{f}_t, f^\star) - V_t(f', f^\star) - \frac{1}{2}V_t(\hat{f}_t, f') - \frac{1}{3}(\beta_t')^2 \right\} \\
&= \max_{f' \in \mathcal{F}_{t-1}} \left\{ -\frac{3}{2}\sum_{i \in [t]} \frac{(f'(x_i) - \frac{2}{3}f^\star(x_i) - \frac{1}{3}\hat{f}_t(x_i))^2}{w_i^2} + \frac{2}{3}V_t(\hat{f}_t, f^\star) - \frac{1}{3}(\beta_t')^2 \right\} \\
&= \frac{2}{3}V_t(\hat{f}_t, f^\star) - \frac{1}{3}(\beta_t')^2,
\end{aligned}$$

where we take $f' = \frac{2}{3}f^\star + \frac{1}{3}\hat{f}_t$.

Additionally, by using Lemma 3.5 with taking minimum over $f' \in \mathcal{F}$ on the both sides of the inequality and $f = f^\star$, we have with probability at least $1 - \delta$

$$\begin{aligned}
L_t(f^\star) &\leq \max_{f' \in \mathcal{F}_{t-1}} \left\{ R_t(f^\star, f') + \frac{1}{2}V_t(f^\star, f') + \frac{1}{3}(\beta_t')^2 \right\} \\
&= \max_{f' \in \mathcal{F}_{t-1}} \left\{ -V_t(f', f^\star) + \frac{1}{2}V_t(f', f^\star) + \frac{1}{3}(\beta_t')^2 \right\} \\
&\leq \frac{1}{3}(\beta_t')^2,
\end{aligned}$$

where we take $f' = f^\star$.

Since $\hat{f}_t = \text{argmin}_{f \in \mathcal{F}_{t-1}} L_t(f)$, we have

$$0 \geq L_t(\hat{f}_t) - L_t(f^\star) \geq \frac{2}{3}V_t(\hat{f}_t, f^\star) - \frac{2}{3}(\beta_t')^2,$$

which means that

$$V_t(\hat{f}_t, f^\star) + \lambda = \sum_{i \in [t]} \frac{1}{\bar{\sigma}_i^2}(\hat{f}_t(x_i) - f^\star(x_i))^2 + \lambda \leq (\beta_t')^2 + \lambda = \beta_t^2.$$

$\square$

**Part II: Event $\mathcal{E}_{\text{var}}^{t,l}$ holds with high probability.** In this part, we focus on the relationship between the sum of true variance $\sum_{i\in[t]} \sigma_i^2/w_i^2$ and the estimation $\widehat{\text{Var}}_t$ conditioned on $\mathcal{E}_{\text{conv}}^{t-1}$. Recall that we define the events $\mathcal{E}_{\text{var}}^t$ and $\mathcal{E}_{\text{conv}}^t$ in (15). First, we will show that event $\mathcal{E}_{\text{var}}^t$ conditioned on $\mathcal{E}_{\text{conv}}^{t-1}$ holds with high probability.

First of all, we provide a lemma for the $\upsilon$-cover.

**Lemma C.4.** *For any $f \in \mathcal{F}$, there exist $f_\upsilon \in \mathcal{F}_\upsilon$ such that $\|f - f_\upsilon\|_\infty \le \upsilon$. Then, we have*

$$\left| \text{Catoni}_{\theta'}\left( \left\{ \frac{1}{w_i^2}(y_i - f(x_i))^2 \right\}_{i\in[t]} \right) - \text{Catoni}_{\theta'}\left( \left\{ \frac{1}{w_i^2}(y_i - f_\upsilon(x_i))^2 \right\}_{i\in[t]} \right) \right|$$

$$\le 4(R + 2L_f)^3\upsilon + 2\sqrt{(R + 2L_f)\upsilon}.$$

*Proof.* We have

$$\frac{1}{w_i^2}(y_i - f(x_i))^2 \le 2R^2 + 8L_f^2,$$

and

$$\left| \frac{1}{w_i^2}(y_i - f(x_i))^2 - \frac{1}{w_i^2}(y_i - f_\upsilon(x_i))^2 \right| \le |(f_\upsilon(x_i) - f(x_i))(2y_i - f(x_i) - f_\upsilon(x_i))|$$

$$\le \upsilon(2R + 4L_f).$$

From the definitions $\theta_{\text{Var}}^t$, we get

$$\theta_{\text{Var}}^t = (2(36L_f\rho\hat{\beta}_{t-1} + c_\eta + 2\sigma_\eta^2) + \epsilon)^{-1} \le 1/\epsilon$$

Combining the two inequalities above, we have

$$\frac{1}{t}\sum_{i\in[t]} \theta_{\text{Var}}^t \left| \frac{1}{w_i^2}(y_i - f(x_i))^2 - \frac{1}{w_i^2}(y_i - f_\upsilon(x_i))^2 \right|$$

$$\le \frac{1}{\epsilon} \cdot \upsilon(2R + 4L_f).$$

Then, by invoking Lemma E.2 with $\Delta = \frac{1}{\epsilon} \cdot \upsilon(2R + 4L_f)$ and taking $\epsilon = \rho^2$, we deduce that

$$\left| \text{Catoni}_{\theta_{\text{Var}}^t}(\{Z_i(f, f')\}_{i\in[t]}) - \text{Catoni}_{\theta_t(f_\upsilon, f'_\upsilon)}(\{Z_i(f_\upsilon, f'_\upsilon)\}_{i\in[t]}) \right|$$

$$\le \frac{1 + 2\theta_{\text{Var}}^t \cdot 2R^2 + 8L_f^2}{\theta_{\text{Var}}^t} \cdot \upsilon(2R + 4L_f) + \sqrt{\frac{2\upsilon(2R + 4L_f)}{(\theta_{\text{Var}}^t)^2}}$$

$$\le 4(R + 2L_f)^3\upsilon + 2\sqrt{(R + 2L_f)\upsilon}.$$

$\square$

**Lemma C.5.** *Under the same condition as Lemma 4.4, and assuming that $f^\star \in \mathcal{F}_{t-1}$, we use the Catoni estimator*

$$\overline{\text{Var}}_t(\hat{f}_t) = t\text{Catoni}_{\theta_{\text{Var}}^t}\left( \left\{ \frac{1}{w_i^2}(y_i - \hat{f}_{t-1}(x_i))^2 \right\}_{i\in[t]} \right),$$

*where $\theta_{\text{Var}}^t = (12(2\sigma_\eta^2 + c_\eta + L_f^2 + 16\rho^2 \cdot \hat{\beta}_{t-1}^2))^{-1}$, and suppose that $\rho \le 1/(1076\iota'(\delta))$ and $\hat{\beta}_{t-1}^2 \le 4320(\iota'(\delta))^2\rho^2 \sum_{i=1}^{t-1} \frac{\sigma_i^2}{w_i^2} + 16 \times 5660(\iota'(\delta))^3 2^{-2l}(2\sigma_\eta^2 + c_\eta) + 120\iota'(\delta)\rho^2 + 26\Delta_\upsilon + 4\lambda$, where $\Delta_\upsilon$ is defined in Table 3.*

*Then, for the time step $t$ such that the following event happens*

$$\mathcal{E}_t = \left\{ t \ge 4(\iota'(\delta))^2 \frac{\sum_{i\in[t]} \frac{\sigma_i^2}{w_i^2}}{\widehat{\text{Var}}_t} + 6(\iota'(\delta))^2 \right\},$$

*we have with probability $1 - 2\delta/TL$*

$$\left| \overline{\mathrm{Var}}_t(\hat{f}_t) - \frac{1}{2} \sum_{i=1}^{t} \frac{\sigma_i^2}{w_i^2} \right| \leq 14\iota'(\delta)(2\sigma_\eta^2 + c_\eta) + 43\Delta_\upsilon + 268\lambda.$$

*where $\Delta_\upsilon = \Theta(L_f \upsilon T^2 + L_f^4 R^2 \iota'(\delta) \upsilon^{0.5} T^{3.5} + L_f^3 R^{1.5} \upsilon^{0.25} T^{1.25} + R^3 L_f^3 \upsilon T + \sqrt{RL_f \upsilon T}).$*

*Proof.* For a fixed $f \in \mathcal{F}_{t-1}$, let

$$V_i(f) = \frac{1}{w_i^2}(y_i - f(x_i))^2 = \frac{1}{w_i^2}\left(\eta_i^2 + (f^\star(x_i) - f(x_i))^2 + 2\eta_i(f^\star(x_i) - f(x_i))\right).$$

We know that

$$|V_i(f)| \leq R^2 + 8L_f^2 + 2\upsilon^2 := R'.$$

We can calculate the conditional mean of $Z_i$:

$$\mu_i(f) = \frac{1}{w_i^2}\left(\sigma_i^2 + (f^\star(x_i) - f(x_i))^2\right),$$

and the sum of variance of $Z_i$:

$$\sum_{i=1}^{t} \mathrm{Var}[V_i(f)] = \sum_{i=1}^{t} \frac{1}{w_i^4} \mathbb{E}\left[(\eta_i^2 - \sigma_i^2)^2 + 4\eta_i^2(f^\star(x_i) - f(x_i))^2\right]$$

$$\leq \sum_{i=1}^{t} \frac{1}{w_i^4} \sigma_i^2 \left(c_\eta + 4(f^\star(x_i) - f(x_i))^2\right)$$

$$\leq \sum_{i=1}^{t} \frac{\sigma_i^2}{w_i^2} \cdot \left(c_\eta + 16\rho^2 \cdot \hat{\beta}_{t-1}^2\right),$$

where the second inequality holds since

$$\frac{1}{w_i^2}(f^\star(x_i) - f(x_i))^2 \leq \frac{1}{w_i^2} \cdot \sup_{f,f' \in \mathcal{F}_{i-1}} \frac{|f(x_i) - f'(x_i)|^2}{\sum_{k \in [i-1]} \frac{1}{\sigma_k^2}(f(x_k) - f'(x_k))^2 + \lambda} \cdot \left(\sum_{k \in [i-1]} \frac{1}{\sigma_k^2}(f(x_k) - f^\star(x_k))^2 + \lambda\right)$$

$$\leq \rho^2 \left(2 \sum_{k \in [t-1]} \frac{1}{\sigma_k^2}(f(x_k) - \hat{f}_{t-1}(x_k))^2 + 2 \sum_{k \in [i-1]} \frac{1}{\sigma_k^2}(f^\star(x_k) - \hat{f}_{t-1}(x_i))^2 + \lambda\right) \cdot$$

$$\leq 4\rho^2 \cdot \hat{\beta}_{t-1}^2, \tag{17}$$

where the second inequality uses the Cauchy-Schwartz inequality, and the last inequality uses the definition of $\mathcal{F}_{i-1}$.

Then, we have

$$\sum_{i=1}^{t} \mu_i^2(f) \leq \sum_{i=1}^{t} \frac{2}{w_i^4}\left(\sigma_i^4 + (f^\star(x_i) - f(x_i))^4\right)$$

$$\leq 2 \sum_{i=1}^{t} \frac{\sigma_i^4}{w_i^4} + V_t(f, f^\star) \cdot \max_{i \in [t]} \frac{f^\star(x_i) - f(x_i))^2}{w_i^2}$$

$$\leq 2\sigma_\eta^2 \sum_{i=1}^{t} \frac{\sigma_i^2}{w_i^2} + V_t(f, f^\star) \cdot 4\rho^2 \cdot \hat{\beta}_{t-1}^2,$$

where the first inequality uses the Cauchy-Schwarz inequality, the last inequality holds due to Assumption 4.1 and (17).

Then, since from $\theta_{\text{Var}}^t = (12(2\sigma_\eta^2 + c_\eta + L_f^2 + 16\rho^2 \cdot \hat{\beta}_{t-1}^2))^{-1}$ we have

$$(\theta_{\text{Var}}^t)^2 \left( \sum_{i=1}^t \text{Var}[V_i(f)] + \sum_{i=1}^t (\mu_i - \bar{\mu})^2 \right)$$

$$\leq (\theta_{\text{Var}}^t)^2 \left( \sum_{i=1}^t \frac{\sigma_i^2}{w_i^2} \cdot \left( 2\sigma_\eta^2 + c_\eta + 16\rho^2 \hat{\beta}_{t-1}^2 \right) + V_t(f, f^\star) \cdot 4\rho^2 \hat{\beta}_{t-1}^2 \right)$$

$$\leq (\theta_{\text{Var}}^t)^2 \left( t\sigma_\eta^2 \left( 2\sigma_\eta^2 + c_\eta + 16\rho^2 \hat{\beta}_{t-1}^2 \right) + t \cdot 4L_f^2 \cdot 4\rho^2 \hat{\beta}_{t-1}^2 \right)$$

$$\leq \frac{t}{6},$$

by the choice of $\iota'(\delta)$ and the condition $\mathcal{E}_t$ holds, we get

$$t = \frac{t}{6} + \frac{5t}{6} \geq (\theta_{\text{Var}}^t)^2 \left( \sum_{i=1}^t \text{Var}[V_i(f)] + \sum_{i=1}^t (\mu_i - \bar{\mu})^2 \right) + 2\log(NTL/\delta).$$

Therefore, we can apply Lemma E.1 and with a union bound over the covering set of $\mathcal{F}_{t-1}$, which is denoted as $\mathcal{F}_{t-1,\upsilon}$, to obtain with probability at least $1 - \delta/TL$, for any $f \in \mathcal{F}_{t-1,\upsilon}$

$$\left| \overline{\text{Var}}_t(f) - \sum_{i=1}^t \mu_i(f) \right| \leq \theta_{\text{Var}}^t \left( \sum_{i=1}^t \text{Var}[V_i(f)] + \sum_{i=1}^t \mu_i^2(f) \right) + 2\frac{\log(NTL/\delta)}{\theta_{\text{Var}}^t}$$

$$\leq \theta_{\text{Var}}^t \left( \sum_{i=1}^t \frac{\sigma_i^2}{w_i^2} \cdot \left( 2\sigma_\eta^2 + c_\eta + 16\rho^2 \hat{\beta}_{t-1}^2 \right) + V_t(f, f^\star) \cdot 4\rho^2 \hat{\beta}_{t-1}^2 \right) + 2\frac{\log(NTL/\delta)}{\theta_{\text{Var}}^t}. \tag{18}$$

Then, for the estimator $\hat{f}_{t-1}$, there exists a $f_\upsilon \in \mathcal{F}_{t-1,\upsilon}$ such that $\|\hat{f}_{t-1} - f_\upsilon\|_\infty \leq \upsilon$. By invoking Lemma C.4, we have

$$\left| \text{Catoni}_{\theta_{\text{Var}}^t} \left( \left\{ \frac{1}{w_i^2}(y_i - \hat{f}_{t-1}(x_i))^2 \right\}_{i \in [t]} \right) - \text{Catoni}_{\theta_{\text{Var}}^t} \left( \left\{ \frac{1}{w_i^2}(y_i - f_\upsilon(x_i))^2 \right\}_{i \in [t]} \right) \right|$$

$$\leq 4(R + 2L_f)^3 \upsilon + 2\sqrt{(R + 2L_f)\upsilon}.$$

Also, we can get

$$\left| \sum_{i=1}^t \mu_i(\hat{f}_{t-1}) - \sum_{i=1}^t \mu_i(f_\upsilon) \right| \leq 4L_f \upsilon T.$$

Combining the results above and by the choice of $\theta_{\text{Var}}^t$, we obtain that with probability at least $1 - \delta$,

$$\left| \overline{\text{Var}}_t(\hat{f}_{t-1}) - \sum_{i=1}^t \mu_i(\hat{f}_{t-1}) \right|$$

$$\leq \left| \overline{\text{Var}}_t(f_\upsilon) - \sum_{i=1}^t \mu_i(f_\upsilon) \right| + T(4(R + 2L_f)^3 \upsilon + 2\sqrt{(R + 2L_f)\upsilon}) + 4L_f \upsilon T$$

$$\leq \theta_{\text{Var}}^t \left( \sum_{i=1}^t \frac{\sigma_i^2}{w_i^2} \cdot \left( 2\sigma_\eta^2 + c_\eta + 16\rho^2 \hat{\beta}_{t-1}^2 \right) + V_t(f_\upsilon, f^\star) \cdot 4\rho^2 \hat{\beta}_{t-1}^2 \right) + 2\frac{\iota'(\delta)}{\theta_{\text{Var}}^t} + \Delta_{\upsilon,3}$$

$$\leq \theta_{\text{Var}}^t \left( \sum_{i=1}^t \frac{\sigma_i^2}{w_i^2} \cdot \left( 2\sigma_\eta^2 + c_\eta + 16\rho^2 \hat{\beta}_{t-1}^2 \right) + (V_t(\hat{f}_{t-1}, f^\star) + 4L_f \upsilon) \cdot 4\rho^2 \hat{\beta}_{t-1}^2 \right) + 2\frac{\iota'(\delta)}{\theta_{\text{Var}}^t} + \Delta_{\upsilon,3}$$

where $\Delta_{\upsilon,3} = 4(R + 2L_f)^3 \upsilon T + 2\sqrt{(R + 2L_f)\upsilon}T + 4L_f \upsilon T$. Further, we have

$$\sum_{i=1}^t \mu_i(\hat{f}_{t-1}) - \sum_{i=1}^t \frac{\sigma_i^2}{w_i^2} = V_t(\hat{f}_{t-1}, f^\star),$$

and

$$V_t(\hat{f}_{t-1}, f^\star) = \sum_{i \in [t-1]} \frac{(\hat{f}_{t-1}(x_i) - f^\star(x_i))^2}{w_i^2} + \frac{(\hat{f}_{t-1}(x_t) - f^\star(x_t))^2}{w_t^2}$$

$$\leq \hat{\beta}_{t-1}^2 + \frac{D_t^2}{w_t^2} \cdot \hat{\beta}_{t-1}^2$$

$$\leq (1 + \rho^2)\hat{\beta}_{t-1}^2,$$

where the first inequality uses the definition of $D_t$ and $f^\star \in \mathcal{F}_{t-1}$. Combining the results above and the value of $\theta_{\text{Var}}^t = (12(2\sigma_\eta^2 + c_\eta + 16\rho^2\hat{\beta}_{t-1}^2))^{-1}$ leads to

$$\left| \overline{\text{Var}}_t(\hat{f}_t) - \sum_{i=1}^t \frac{\sigma_i^2}{w_i^2} \right|$$

$$\leq \theta_{\text{Var}}^t \left( \sum_{i=1}^t \frac{\sigma_i^2}{w_i^2} \cdot \left( 2\sigma_\eta^2 + c_\eta + 16\rho^2\hat{\beta}_{t-1}^2 \right) + ((1+\rho^2)\hat{\beta}_{t-1}^2 + 4L_f\upsilon T) \cdot 4\rho^2\hat{\beta}_{t-1}^2 \right) + 2\frac{\iota'(\delta)}{\theta_{\text{Var}}^t} + \Delta_{v,3} + (1+\rho^2)\hat{\beta}_{t-1}^2$$

$$\leq \frac{1}{4}\sum_{i=1}^t \frac{\sigma_i^2}{w_i^2} + \frac{1+\rho^2}{16}\hat{\beta}_{t-1}^2 + \frac{1}{4}L_f\upsilon T + 8\iota'(\delta)(2\sigma_\eta^2 + c_\eta) + 64\rho^2\hat{\beta}_{t-1}^2 + \Delta_{v,3} + (1+\rho^2)\hat{\beta}_{t-1}^2$$

$$\leq \frac{1}{4}\sum_{i=1}^t \frac{\sigma_i^2}{w_i^2} + 67\Big( 4320(\iota'(\delta))^2\rho^2 \sum_{i=1}^{t-1} \frac{\sigma_i^2}{w_i^2} + 16 \times 5660(\iota'(\delta))^3 2\rho^2(2\sigma_\eta^2 + c_\eta) + 120\iota'(\delta)\rho^2 + 26\Delta_{v,2} + 4\lambda \Big)$$

$$+ \frac{1}{4}L_f\upsilon T + \Delta_{v,3}$$

$$\leq \frac{1}{2}\sum_{i=1}^t \frac{\sigma_i^2}{w_i^2} + 67\Big( 16 \times 5660(\iota'(\delta))^3\rho^2(2\sigma_\eta^2 + c_\eta) + 26\Delta_{v,2} + 4\lambda \Big) + 8\iota'(\delta)(2\sigma_\eta^2 + c_\eta) + \Delta_v$$

$$\leq \frac{1}{2}\sum_{i=1}^t \frac{\sigma_i^2}{w_i^2} + 14\iota'(\delta)(2\sigma_\eta^2 + c_\eta) + 43\Delta_v + 268\lambda,$$

where the second inequality uses $\rho \leq 1$, and the third inequality uses the value of $\hat{\beta}_{t-1}^2$, the last second inequalities holds since we know from $\rho \leq 1/(1076\iota'(\delta))$ that $67 \times 4320(\iota'(\delta))^2\rho^2 \leq 1/4$, and the last inequality also holds due to $67 \times 4 \times 12^3(\iota'(\delta))^2\rho^2 \leq 2$, and we define $\Delta_v = 67 \times 26\Delta_{v,2} + \Delta_{v,3} + \frac{1}{4}L_f\upsilon T = \Theta(L_f\upsilon T^2 + L_f^4 R^2\iota'(\delta)\upsilon^{0.5}T^{3.5} + L_f^3 R^{1.5}\upsilon^{0.25}T^{1.25} + R^3 L_f^3\upsilon T + \sqrt{RL_f\upsilon T})$, which concludes the proof. There is a fixable error. Now, I change the plug-in in $\widehat{\text{Var}}_t$ from $\hat{f}_t$ to $\hat{f}_{t-1}$, and change the analysis above, so now the upper and lower bound is reasonable. $\square$

**Lemma C.6** (Formal version of Lemma 4.5). *Recall the definition of the variance estimation from* (6):

$$\widehat{\text{Var}}_t^l = \overline{\text{Var}}_t^l + 14\iota'(\delta)(2\sigma_\eta^2 + c_\eta) + 43\Delta_v + 268\lambda^l,$$

*where* $\theta_{\text{Var}}^{t,l} = (4(2\sigma_\eta^2 + c_\eta + L_f^2 + 2^{-2l+4} \cdot \hat{\beta}_{t-1}^2))^{-1}$ *and* $\Delta_v = \Theta(L_f\upsilon T^2 + L_f^4 R^2\iota'(\delta)\upsilon^{0.5}T^{3.5} + L_f^3 R^{1.5}\upsilon^{0.25}T^{1.25} + R^3 L_f^3\upsilon T + \sqrt{RL_f\upsilon T})$. *Then, conditioned on* $\mathcal{E}_{\text{conv}}^{t-1}$, *when* $2^l \geq 1076\iota'(\delta)$ *we have with probability at least* $1 - 2\delta$ *for all* $t \geq 14(\iota'(\delta))^2$,

$$\frac{1}{2}\sum_{i \in \Psi_t^l} \frac{\sigma_i^2}{w_i^2} \leq \widehat{\text{Var}}_t^l \leq \frac{3}{2}\sum_{i \in \Psi_t^l} \frac{\sigma_i^2}{w_i^2} + 2\Big( 14\iota'(\delta)(2\sigma_\eta^2 + c_\eta) + 43\Delta_v + 268\lambda^l \Big),$$

*which implies that* $\cup_{t \in [T]: t \geq 14(\iota'(\delta))^2} \mathcal{E}_{\text{var}}^t$ *holds.*

*Proof of Lemma 4.5.* If we suppose that $\mathcal{E}_t$ happens, since $\mathcal{E}_{\text{conv}}^{t-1}$ holds true, we can apply Lemma C.5 to each $l \in [L]$ satisfying that $2^l \geq 1076\iota'(\delta)$ with $\rho = 2^{-l}$ and obtain with probability $1 - 2\delta$,

$$\left| \overline{\text{Var}}_t(\hat{f}_t) - \frac{1}{2}\sum_{i=1}^t \frac{\sigma_i^2}{w_i^2} \right| \leq 14\iota'(\delta)(2\sigma_\eta^2 + c_\eta) + 43\Delta_v + 268\lambda^l,$$

which indicates the desired result according to the definition of $\widehat{\mathrm{Var}}_t^l$.

Furthermore, we show that when $t \geq 14(\iota'(\delta))^2$, $\mathcal{E}_t$ holds true.

$$t \geq 14(\iota'(\delta))^2 = 4(\iota'(\delta))^2 \times 2 + 6(\iota'(\delta))^2$$

$$\geq 4(\iota'(\delta))^2 \frac{\sum_{i \in \Psi_t^l} \frac{\sigma_i^2}{w_i^2}}{\widehat{\mathrm{Var}}_t^l} + 6(\iota'(\delta))^2,$$

where the second inequality uses $\frac{1}{2} \sum_{i \in \Psi_t^l} \frac{\sigma_i^2}{w_i^2} \leq \widehat{\mathrm{Var}}_t^l$. $\qquad\qquad\square$

## Part III: Sharpness of the confidence set

**Lemma C.7.** *Conditioned on $\mathcal{E}_{\mathrm{var}}^t$, if we take $\upsilon = O(1/L_f^{12} R^6 (\iota'(\delta))^2 T^7)$, $\epsilon = 2^{-2l}$, and take the confidence parameter as* (8):

$$(\hat{\beta}_t^l)^2 = 2880(\iota'(\delta))^2 2^{-2l} \widehat{\mathrm{Var}}_t^l + 60\iota'(\delta)2^{-2l} + 12\Delta_{\upsilon,2} + 2\lambda^l,$$

*where $\Delta_\upsilon = \Theta\big(L_f \upsilon T^2 + L_f^4 R^2 \iota'(\delta) \upsilon^{0.5} T^{3.5} + L_f^3 R^{1.5} \upsilon^{0.25} T^{1.25} + R^3 L_f^3 \upsilon T + \sqrt{R L_f \upsilon T}\big)$. We have*

$$(\hat{\beta}_t^l)^2 \leq 4320(\iota'(\delta))^2 2^{-2l} \sum_{i \in \Psi_t^l} \frac{\sigma_i^2}{w_i^2} + 16 \times 5660(\iota'(\delta))^3 2^{-2l}(2\sigma_\eta^2 + c_\eta) + 120\iota'(\delta)2^{-2l} + 26\Delta_\upsilon + 4\lambda^l.$$

*Additionally, with probability at least $1 - \delta$, the following event occurs:*

$$\cup_{t \in [T]: t \geq 3(\iota'(\delta))^2} \mathcal{E}_{\mathrm{conv}}^t = \big\{f^\star \in \mathcal{F}_t^l, \text{ for } l \in [L], \ 2^l \geq 1076\iota'(\delta)\big\}.$$

*Proof.* By invoking Lemma C.3 for $l$ with $\rho = 2^{-l}$, we get with probability at least $1 - \delta$,

$$\sum_{i \in \Psi_t^l} \frac{(f^\star(x_i) - \hat{f}_t^l(x_i))^2)}{w_i^2} + \lambda^l$$

$$\leq \frac{1}{2}(\hat{\beta}_{t-1}^l)^2 + 6\left(48(\iota'(\delta))^2 2^{-2l}\left(\frac{\left(\sum_{i \in \Psi_t^l} \sigma_i^2/w_i^2\right)^2}{\widehat{\mathrm{Var}}_t^l} + \widehat{\mathrm{Var}}_t^l\right) + 5\iota'(\delta)2^{-2l} + \Delta_{\upsilon,2}\right) + \lambda^l. \qquad (19)$$

Then, since Corollary C.6 implies that with probability at least $1 - 2\delta$,

$$\frac{1}{2}\sum_{i \in \Psi_t^l}\frac{\sigma_i^2}{w_i^2} \leq \widehat{\mathrm{Var}}_t^l \leq \frac{3}{2}\sum_{i \in \Psi_t^l}\frac{\sigma_i^2}{w_i^2} + 2\left(14\iota'(\delta)(2\sigma_\eta^2 + c_\eta) + 43\Delta_\upsilon + 268\lambda^l\right), \qquad (20)$$

we get with probability at least $1 - 2\delta$,

$$\frac{\left(\sum_{i \in \Psi_t^l} \sigma_i^2/w_i^2\right)^2}{\widehat{\mathrm{Var}}_t^l} + \widehat{\mathrm{Var}}_t^l \leq 5\widehat{\mathrm{Var}}_t^l.$$

Hence, we deduce that

$$\sum_{i \in \Psi_t^l} \frac{(f^\star(x_i) - \hat{f}_t^l(x_i))^2)}{w_i^2} + \lambda^l$$

$$\leq \frac{1}{2}(\hat{\beta}_{t-1}^l)^2 + 6\left(48(\iota'(\delta))^2 2^{-2l} 5\widehat{\mathrm{Var}}_t^l + 5\iota'(\delta)2^{-2l} + \Delta_{\upsilon,2}\right) + \lambda^l$$

$$\leq \frac{1}{2}(\hat{\beta}_{t-1}^l)^2 + 1440(\iota'(\delta))^2 2^{-2l}\widehat{\mathrm{Var}}_t^l + 30\iota'(\delta)2^{-2l} + 6\Delta_{\upsilon,2} + \lambda^l$$

$$\leq (\hat{\beta}_t^l)^2,$$

where the last inequality uses the definition of $\hat{\beta}_t^l$.

Moreover, we define

$$B_\beta^l = 16 \times 5660(\iota'(\delta))^3 2^{-2l}(2\sigma_\eta^2 + c_\eta) + 120\iota'(\delta)2^{-2l} + 26\Delta_v + 4\lambda^l,$$

and deduce from (20) that

$$
\begin{aligned}
(\hat{\beta}_t^l)^2 =& 2880(\iota'(\delta))^2 2^{-2l}\widehat{\mathrm{Var}}_t^l + 60\iota'(\delta)2^{-2l} + 12\Delta_{v,2} + 2\lambda^l \\
\leq& 4320(\iota'(\delta))^2 2^{-2l} \sum_{i \in \Psi_t^l} \frac{\sigma_i^2}{w_i^2} + 5660(\iota'(\delta))^2 2^{-2l}\left(14\iota'(\delta)(2\sigma_\eta^2 + c_\eta) + 43\Delta_v + 268\lambda^l\right) + 60\iota'(\delta)2^{-2l} + 12\Delta_{v,2} + 2\lambda^l \\
\leq& 4320(\iota'(\delta))^2 2^{-2l} \sum_{i \in \Psi_t^l} \frac{\sigma_i^2}{w_i^2} + \frac{1}{2}B_\beta^l + \frac{1}{2}B_\beta^l \\
=& 4320(\iota'(\delta))^2 2^{-2l} \sum_{i \in \Psi_t^l} \frac{\sigma_i^2}{w_i^2} + 16 \times 5660(\iota'(\delta))^3 2^{-2l}(2\sigma_\eta^2 + c_\eta) + 120\iota'(\delta)2^{-2l} + 26\Delta_v + 4\lambda^l,
\end{aligned}
$$

where the last inequality holds since $2^l \geq 1076\iota'(\delta)$. $\qquad\square$

**Part IV: Bounding the regret conditioning on good events.** Recall the notation for the eluder coefficient for each layer

$$D_t^l(x) = \sup_{f,f' \in \mathcal{F}_{t-1}^l} \frac{|f(x) - f'(x)|}{\sqrt{\sum_{i \in \Psi_{t-1}^l} \frac{1}{w_i^2}(f(x_i) - f'(x_i))^2 + \lambda^l}}.$$

**Lemma C.8.** *Let $l_\star = \lceil \log_2(1076\iota'(\delta)) \rceil$. Under Assumption 4.1 and Algorithm 2, if $\cup_{t \geq 14(\iota'(\delta))^2}\mathcal{E}_{\mathrm{conv}}^t$ happens, then, for all $l \in [l_\star, L]$, $t \in \Psi_T^l$, and $X_t^* = \mathrm{argmax}_{x \in \mathcal{X}_t} f^\star(x) \in \mathcal{X}_t^l$, and the regret at the l-th level is bounded by*

$$\sum_{t \in \Psi_T^l : t \geq 14(\iota'(\delta))^2} (f^\star(x_t^*) - f^\star(x_t)) \leq 2^{-l+3}\hat{\beta}_T^{l-1}|\Psi_T^l|.$$

*Proof of Lemma 4.6.* First according to Algorithm 2, we will prove that for all $t \in \Psi_T^l$, $X_t^* \in \mathcal{X}_t^l$ by induction from $l_\star$ to $l$, where $l$ is the level from which $x_t$ arises, and note that Algorithm 2 starts from level $l_\star$.

$$(\hat{\beta}_{t-1}^l)^2 = \Theta\left((\iota'(\delta))^2 2^{-2l} \sum_{i \in \Psi_{t-1}^l} \sigma_i^2/w_i^2 + (\iota'(\delta))^3 2^{-2l}(\sigma_\eta^2 + c_\eta) + \Delta_v\right).$$

Assume that $X_t^* \in \mathcal{X}_t^{l_0}$ for some $l_0 \in [l_\star, l-1]$ and $\mathcal{X}_t^{l_0+1}$ exists. Since $\Psi_t^{l_0+1}$ exists only if $D_t^{l_0}(x) \leq 2^{-l_0}$ for all $x \in \mathcal{X}_t^{l_0}$. Then, we denote $X_t^{l_0} = \mathrm{argmax}_{x \in \mathcal{X}_t^{l_0}} \hat{f}_{t-1}^{l_0}(x)$ and deduce that

$$
\begin{aligned}
\hat{f}_{t-1}^{l_0}(x_t^*) - \hat{f}_{t-1}^{l_0}(x_t^{l_0}) \geq& f^\star(x_t^*) - f^\star(x_t^{l_0}) - \left|\hat{f}_{t-1}^{l_0}(x_t^*) - f^\star(x_t^*)\right| - \left|\hat{f}_{t-1}^{l_0}(x_t^{l_0}) - f^\star(x_t^{l_0})\right| \\
\geq& -D_t^{l_0}(x_t^*)\hat{\beta}_{t-1}^l - D_t^{l_0}(x_t^*)\hat{\beta}_{t-1}^l \\
\geq& -2^{-l_0+1}\hat{\beta}_{t-1}^l,
\end{aligned}
$$

where the second inequality uses $X_t^* = \mathrm{argmax}_{x \in \mathcal{X}_t} f^\star(x)$, and the last inequality holds by using $D_t^{l_0}(x_t^*) \leq 2^{-l_0}$, $D_t^{l_0}(x_t^{l_0}) \leq 2^{-l_0}$. Therefore, from the definition of $\mathcal{X}_t^{l_0+1} = \{x \in \mathcal{X}_t^{l_0} \mid \hat{f}_{t-1}^{l_0}(x) \geq \max_{x \in \mathcal{X}_t^{l_0}} \hat{f}_{t-1}^{l_0}(x) - 2^{-l_0+1}\hat{\beta}_{t-1}^{l-1}\}$, we obtain that $X_t^* \in \mathcal{X}_t^{l_0+1}$. Hence, via induction, we can prove that $X_t^* \in \mathcal{X}_t^l$.

Next, for the $l_0 + 1 \leq l$ since $X_t^*, X_t \in \mathcal{X}_t^l$, we know that

$$\hat{f}_{t-1}^{l-1}(x_t^*) - \hat{f}_{t-1}^{l-1}(x_t) \leq 2^{-l+2}\hat{\beta}_t^{l-1}, \tag{21}$$

and

$$D_t^{l-1}(x_t^*) \leq 2^{-l+1}, \quad D_t^{l-1}(x_t) \leq 2^{-l+1}. \tag{22}$$

Thus, we derive that

$$
\begin{aligned}
f^\star(x_t^*) - f^\star(x_t) &\leq \hat{f}_{t-1}^{l-1}(x_t^*) - \hat{f}_{t-1}^{l-1}(x_t) + \left| f^\star(x_t^*) - \hat{f}_{t-1}^{l-1}(x_t^*) \right| + \left| f^\star(x_t) - \hat{f}_{t-1}^{l-1}(x_t) \right| \\
&\leq 2^{-l+2}\hat{\beta}_t^{l-1} + D_t^{l-1}(x_t^*)\hat{\beta}_t^{l-1} + D_t^{l-1}(x_t)\hat{\beta}_t^{l-1} \\
&\leq 2^{-l+2}\hat{\beta}_t^{l-1} + 2^{-l+1}\hat{\beta}_t^{l-1} + 2^{-l+1}\hat{\beta}_t^{l-1} \\
&= 2^{-l+3}\hat{\beta}_t^{l-1},
\end{aligned}
$$

which implies that

$$
\begin{aligned}
\sum_{t \in \Psi_T^l : t \geq 14(\iota'(\delta))^2} (f^\star(x_t^*) - f^\star(x_t)) &\leq \sum_{t \in \Psi_T^l} 2^{-l+3}\hat{\beta}_t^{l-1} \\
&\leq 2^{-l+3}\hat{\beta}_T^{l-1}|\Psi_T^l|.
\end{aligned}
$$

$\square$

Now, we provide the proof for the main theorem.

*Proof of Theorem 4.2.* Recall the definition of two good events

$$
\begin{aligned}
\mathcal{E}_{\text{var}}^t &:= \left\{ \sum_{i \in \Psi_t^l} \frac{\sigma_i^2}{w_i^2} \leq 2\widehat{\text{Var}}_t^l, \text{ for } l \in [L], \ 2^l \geq 1076\iota'(\delta) \right\}, \\
\mathcal{E}_{\text{conv}}^t &:= \left\{ f^\star \in \mathcal{F}_t^l, \text{ for } l \in [L], \ 2^l \geq 1076\iota'(\delta) \right\}.
\end{aligned}
$$

By invoking Lemma C.7 and C.6, we have

$$\mathbb{P}\left(\cup_{t \in [T] : t \geq 3(\iota'(\delta))^2}(\mathcal{E}_{\text{conv}}^t \cap \mathcal{E}_{\text{var}}^t)\right) \geq 1 - 3\delta.$$

In the remaining proof, we suppose that $\cup_{t \in [T] : t \geq 3(\iota'(\delta))^2}(\mathcal{E}_{\text{conv}}^t \cap \mathcal{E}_{\text{var}}^t)$ holds. Let $l_\star$ be the minimum $l \in [L]$ such that $2^l \geq \max\{18c_\beta^2(\iota'(\delta))^2, 6 \times 48^2 L_f(\iota'(\delta))^2\}$.

We can decompose the regret into three parts

$$R_T = \underbrace{\sum_{l=l_\star}\sum_{t \in \Psi_T^l}(f^\star(x_t^*) - f^\star(x_t))}_{I_1} + \underbrace{\sum_{l=l_\star+1}^{L}\sum_{t \in \Psi_T^l}(f^\star(x_t^*) - f^\star(x_t))}_{I_2} + \underbrace{\sum_{t \in [T] \backslash \cup_{l \in [L]} \Psi_T^l}(f^\star(x_t^*) - f^\star(x_t))}_{I_3},$$

where we use the short notation $X_t^* = \operatorname{argmax}_{x \in \mathcal{X}_t} f^\star(x)$. For the term $I_1$, we have

$$
\begin{aligned}
I_1 &\leq 2l_\star L_f |\Psi_T^{l_\star}| \\
&\leq 2l_\star L_f 2^{2l_\star} \cdot \sum_{t \in \Psi_{T,l_\star}} \frac{(D_t^{l_\star}(x_t))^2}{w_t^2} \\
&\leq L_f l_\star 2^{2l_\star+1}\dim_{1,T}(\mathcal{F}) = \widetilde{O}(L_f \log(N) \cdot \dim_{1,T}(\mathcal{F})),
\end{aligned}
$$

where the second inequality holds due to $2^{-l_\star} = D_t^{l_\star}(x_t)/w_t$ from Algorithm 2, and the last inequality follows from the fact that $2^{l_\star} = \Theta(L_f(\iota'(\delta))^2)$.

For the term $I_2$, we invoke Lemma 4.6 to get

$$I_2 \leq \sum_{l \geq l_\star+1} \hat{\beta}_T^l 2^{-l+3} |\Psi_T^l| \leq 8L_f \sum_{l \geq l_\star} \hat{\beta}_T^l 2^{-l} \cdot 2^{2l} \sum_{i \in \Psi_{T,l}} \min\left\{1, \frac{(D_t^l(x_i))^2}{w_i^2}\right\}$$

$$\leq 8L_f \sum_{l \geq l_\star+1} \hat{\beta}_T^l 2^l \dim_{1,T}(\mathcal{F})$$

$$= \widetilde{O}\left(L_f \sqrt{\sum_{t \in [T]} \sigma_t^2 \cdot \log N} \cdot \dim_{1,T}(\mathcal{F}) + L_f(\log N)^{3/4} \dim_{1,T}(\mathcal{F})(\sqrt{c_\eta} + \sigma_\eta)\right),$$

where the last inequality holds since we know from Lemma C.7 that $\iota'(\delta) = \Theta\left(\sqrt{\log\left(RL_f(\sigma_\eta^2 + c_\eta + \Delta_v + \lambda^l)NLT/\delta\right)}\right)$ and

$$\hat{\beta}_t^l = \Theta\left(\iota'(\delta)2^{-l}\sqrt{\sum_{i \in \Psi_t^l} \sigma_i^2/w_i^2} + (\iota'(\delta))^{3/2}2^{-l}(\sigma_\eta + \sqrt{c_\eta}) + \sqrt{\Delta_v}\right).$$

For the term $I_3$, we have

$$I_3 \leq \sum_{t \in [T] \setminus \cup_{l \in [L]} \Psi_T^l} (f_{t-1}^{l_t}(x_t) - f^\star(x_t))$$

$$\leq \sum_{t \in [T] \setminus \cup_{l \in [L]} \Psi_T^l} \hat{\beta}_t^{l_t} \cdot \gamma$$

$$= \widetilde{O}\left(T \cdot \sigma_\eta \sqrt{T} \cdot \frac{1}{\sigma_\eta T^{3/2}}\right),$$

where the first inequality follows from the peeling rule that $D_{\mathcal{F}}^{t,l}(x_t) \leq \gamma$ for $t \in [T] \setminus \cup_{l \in [L]} \Psi_T^l$, and the second inequality uses the upper bound of $\hat{\beta}_t^{l_t}$ and the $\gamma = 1/\sigma_\eta T^{3/2}$.

Combining all three terms leads us to the eventual result. $\qquad\square$

# D. Another Algorithm: Construct Two Confidence Sets

In this section, we develop a substitution for the Catoni estimator by constructing another candidate set and picking one estimator out of the set instead of solving the min-max optimization as (3). Additionally, for simplicity, we consider the finite function space $\mathcal{F}$ with cardinality $N$ in this section. By standard analysis for the union bound over the cover set, we can extend the analysis to infinite function space with finite covering number as in Appendix B.2 and C.1.

## D.1. Known Variance

**Notations.** Recall the probability parameter as follows:

$$\delta_{n,t} := \frac{\delta}{N(T+1)}, \quad \delta_t := \sum_{n \in [N]} \delta_{n,t} = \frac{\delta}{T+1}. \tag{23}$$

Note again we have $\sum_{n \in [N], t \in [T]} \delta_{n,t} = \sum_{t \in [T]} \delta_t \leq \delta$.

We also define the following logarithmic factor throughout the analysis that

$$\iota(\delta_{n,t}) = \sqrt{\log\left(\frac{\sqrt{21} \cdot 288 \cdot L_f^2 R^2 T^{3.5}}{\delta_{n,t}}\right)} \geq \sqrt{\log\left(\frac{96R(1 + 2R/\epsilon)T^2}{\min(1, \alpha^2/\sqrt{20T}L_f^2)\epsilon^2 \delta_{n,t}}\right)},$$

where for the last inequality we choose $\alpha = 1/\sqrt{T}$, $\epsilon = 1$ and use the assumption that $R \geq 1$, $L_f \geq 1$ without loss of generality.

---

**Algorithm 3** Catoni-OFUL with Candidate Set

---

**Input:** Parameter $\delta_{n,t}$ and $\hat{\beta}_t$ for each $t \in [T]$.
**for** t=1,2,...,T **do**
  Pick action $x_t$ according to $\max_{f \in \mathcal{F}_{t-1}, x \in \mathcal{D}_t} f(x)$;
  Observe the reward $y_t$;
  Let the weight $\bar{\sigma}_t = \max\left(\sigma_t, \alpha, 4\sqrt{2\iota(\delta_{n,t})L_f \cdot D_{\mathcal{F}}(x_t; x_{[t-1]}, \bar{\sigma}_{[t-1]})}\right)$;
  Construct $\hat{\mathcal{F}}_t$ as defined in (24) and pick any $\hat{f}_t \in \hat{\mathcal{F}}_t$;
  Construct confidence set
$$\mathcal{F}_t := \left\{ f \in \mathcal{F} : \sum_{i \in [t]} \frac{1}{\bar{\sigma}_i^2} \left( f(x_i) - \hat{f}_t(x_i) \right)^2 \leq \hat{\beta}_t^2 \right\};$$

**end for**

---

Suppose that for each $t \in [T]$, the upper bound of the noise variance $\sigma_t^2$ is known. We now consider the following VOFUL algorithm tailored to this nonlinear function class setup. After specifying parameters $\delta_{n,t}$ and $\hat{\beta}_t$ for $t \in [T]$, and the weight

$$\bar{\sigma}_t = \max\left(\sigma_t, \alpha, 4\sqrt{2\iota(\delta_{n,t})L_f \cdot D_{\mathcal{F}}(x_t; x_{[t-1]}, \bar{\sigma}_{[t-1]})}\right)$$

depending on the variance $\sigma_t$ and eluder coefficient $D_{\mathcal{F}}$, we define the candidate set for the estimator as

$$\hat{\mathcal{F}}_t := \left\{ \hat{f} \in \mathcal{F} : \min_{f \in \mathcal{F}} \sum_{i \in [t]} \frac{1}{\bar{\sigma}_i^2} \left( f(x_i) - \hat{f}(x_i) \right)^2 + 2t \cdot \mathsf{Catoni}_{\theta(f,\hat{f})}(Z_1, \cdots, Z_t) \geq -\frac{1}{4}\hat{\beta}_t^2 \right\}, \tag{24}$$

where $Z_i(f, \hat{f}) = \frac{1}{\bar{\sigma}_i^2}(f(x_i) - \hat{f}(x_i))(\hat{f}(x_i) - y_i)$ and

$$\theta_t(f, \hat{f}) = \sqrt{\frac{\iota^2(\delta_{n,t})}{\sum_{i \in [t]} \frac{1}{\bar{\sigma}_i^2} \left( f(x_i) - \hat{f}(x_i) \right)^2 + \sum_{i \in [t]} \frac{2}{\bar{\sigma}_i^4} \left( f(x_i) - \hat{f}(x_i) \right)^4 + \epsilon^2}}.$$

This candidate set selects robust estimations for the true function $f^\star$, and we will prove in the sequel that the $f^\star$ belongs to $\hat{\mathcal{F}}_t$. Then, we choose any function $\hat{f}_t$ from $\hat{\mathcal{F}}_t$ and further construct the confidence set $\mathcal{F}_t$ with a small weighted square error. We will demonstrate that $f^\star \in \mathcal{F}_t$. Based on the principle of optimism in the face of uncertainty, we choose the greedy function $\hat{f}_t \in \mathcal{F}_t$ and the greedy action $X_t \in \mathcal{D}_t$.

**Theorem D.1.** *Under Algorithm 3 with the parameter* $\delta_{n,t} = \delta/N(T+1)$, $\iota(\delta_{n,t}) = \sqrt{\log\left(\sqrt{21} \cdot 288 \cdot L_f^2 R^2 T^{3.5}/\delta_{n,t}\right)}$, *and*

$$\hat{\beta}_t := \left[\left(8\left(8 \cdot 13^4 + 2 \cdot 13^2 + 13\right)\right)^{1/2} + 13\sqrt{2}\lambda^{1/4}\right]\iota(\delta_{n,t}), \tag{25}$$

*with probability* $1 - 2\delta$, *we can bound the regret by*

$$R_T = \widetilde{O}\left(L_f \cdot \log N \cdot \dim_{\frac{1}{\sqrt{T}}, T}(\mathcal{F}) + L_f \left(\sum_{t \in [T]} \sigma_t^2\right)^{1/2} \cdot \sqrt{\dim_{\frac{1}{\sqrt{T}}, T}(\mathcal{F}) \cdot \log N}\right).$$

We now divide the argument into the following three parts. In the analysis, we omit $(f, f')$ in $Z_i$ and $\theta$ when there is no confusion.

**Part I: With high probability $1 - \delta$, all the sets $\hat{\mathcal{F}}_t$ are non-empty for each $t \in [\widetilde{O}(1), T]$.**

**Lemma D.2.** *For any iteration $t \in [T], t \geq 7\iota^2(\delta_{n,T})$ and the set $\hat{\mathcal{F}}_t$ as constructed in (24), we have with probability at least $1 - \delta$, $\cap_{t \in [T], t \geq 7\iota^2(\delta_{n,T})}\hat{\mathcal{F}}_t \neq \emptyset$.*

*Proof.* It suffices to show that for each $t$, we have $\mathbb{P}(f^\star \notin \hat{\mathcal{F}}_t) \leq \delta_t$. When $f^\star \notin \hat{\mathcal{F}}_t$, there exists some $f^n \in \mathcal{F}$ such that

$$\sum_{i\in[t]} \frac{1}{\bar{\sigma}_i^2} (f^n(x_i) - f^\star(x_i))^2 + 2t \cdot \mathsf{Catoni}_\theta \left( \left\{ \frac{1}{\bar{\sigma}_i^2} (f^n(x_i) - f^\star(x_i)) (f^\star(x_i) - y_i) \right\}_{i\in[t]} \right) < -\frac{1}{4}\hat{\beta}_t^2,$$

which implies that

$$t \cdot \mathsf{Catoni}_\theta \left( \left\{ \frac{1}{\bar{\sigma}_i^2} (f^n(x_i) - f^\star(x_i)) (f^\star(x_i) - y_i) \right\}_{i\in[t]} \right) < \frac{-\frac{1}{4}\hat{\beta}_t^2 - \sum_{i\in[t]} \frac{1}{\bar{\sigma}_i^2} (f^n(x_i) - f^\star(x_i))^2}{2}.$$

Now we bound the probability that the above inequality happens. We use the notation for any $f, f' \in \mathcal{F}$

$$Z_i(f, f') = \frac{1}{\bar{\sigma}_i^2} (f(x_i) - f'(x_i)) (f'(x_i) - y_i),$$

which is short-notated as $Z_i$ when no confusion arises.

We apply Lemma E.1 to $Z_i(f^n, f^*)$. The variable $Z_i$ has conditional mean $\mu_i = \mathbb{E}Z_i = 0$, and sum of conditional variance

$$\sum_{i\in[t]} \mathbb{E}\frac{1}{\bar{\sigma}_i^4} (f^n(x_i) - f^\star(x_i))^2 (f^\star(x_i) - y_i)^2$$
$$\leq \sum_{i\in[t]} \frac{1}{\bar{\sigma}_i^2} (f^n(x_i) - f^\star(x_i))^2$$
$$:= V(f^n, f^*),$$

where the inequality uses the fact that $\mathbb{E} (f^\star(x_i) - y_i)^2 = \sigma_i^2 \leq \bar{\sigma}_i^2$. We can also bound $\theta$ by definition $\theta \in [a, A]$ where

$$A = \iota(\delta_{n,t,l})/\epsilon \text{ and } a = \frac{\iota(\delta_{n,t,l})}{\sqrt{20L_f^4 t/\alpha^4 + \epsilon^2}}.$$

Thus, we have $\log\left( \frac{48R(1+2AR)t^2}{\min(1,a)\epsilon^2\delta_{n,t}} \log(A/a) \right) \leq 3\iota^2(\delta_{n,t,l})$ given choice of $\alpha = 1/\sqrt{T}$ and $\epsilon = 1$, and thus for any iteration

$$t \geq 7\iota^2(\delta_{n,t,l}) \geq \iota^2(\delta_{n,t,l}) + 6\iota^2(\delta_{n,t,l}) \geq \iota^2(\delta_{n,t,l}) + 2\log\left( \frac{48R(1+2AR)t^2}{\min(1,a)\epsilon^2\delta_{n,t}} \log(A/a) \right),$$

by choice of $\theta = \sqrt{\frac{\iota^2(\delta_{n,t,l})}{\sum_{i\in[t]} \frac{1}{\bar{\sigma}_i^2}(f^n(x_i)-f^\star(x_i))^2 + \sum_{i\in[t]} \frac{2}{\bar{\sigma}_i^4}(f^n(x_i)-f^\star(x_i))^4 + \epsilon^2}}$, with probability at least $1 - \delta_{n,t}$, we have from

Lemma 3.2 that

$$t \cdot \left| \mathsf{Catoni}_\theta(\{Z_i\}_{i \in [t]}) \right|$$

$$\leq \iota(\delta_{n,t,l}) \frac{V + \sum_{i \in [t]} (\mu_i - \bar{\mu})^2}{\sqrt{V + \sum_{i \in [t]} \frac{2}{\bar{\sigma}_i^4} (f^n(x_i) - f^\star(x_i))^4 + \epsilon^2}} + 12\iota(\delta_{n,t,l}) \left( \sqrt{V + \sum_{i \in [t]} \frac{2}{\bar{\sigma}_i^4} (f^n(x_i) - f^\star(x_i))^4 + \epsilon^2} \right)$$

$$\qquad + \epsilon + t\bar{\mu}$$

$$\overset{(o)}{\leq} 13 \sqrt{V \cdot \left( 1 + \max_{i \in [t]} \frac{2}{\bar{\sigma}_i^2} (f^n(x_i) - f^\star(x_i))^2 \right) \cdot \iota(\delta_{n,t,l})} + 12\iota(\delta_{n,t,l})\epsilon + \epsilon$$

$$\overset{(i)}{\leq} 13 \sqrt{V \cdot \left( 1 + \max_{i \in [t]} \frac{4L_f}{\bar{\sigma}_i^2} D_\mathcal{F}(x_i; x_{[i-1]}, \bar{\sigma}_{[i-1]}) \sqrt{\sum_{k \in [i-1]} \frac{1}{\bar{\sigma}_k^2} (f^n(x_k) - f^\star(x_k))^2 + \lambda} \right) \iota(\delta_{n,t,l})}$$

$$\overset{(ii)}{\leq} 13\iota(\delta_{n,t,l}) \sqrt{\left( 1 + \frac{\sqrt{\lambda}}{8} \right) V + \frac{13}{2\sqrt{2}} V^{3/4} \left( \iota^2(\delta_{n,t,l}) \right)^{1/4}}$$

$$\overset{(iii)}{\leq} \frac{V}{4} + 169 \left( 1 + \frac{\sqrt{\lambda}}{8} \right) \iota^2(\delta_{n,t,l}) + \frac{V}{4} + 13^4 \iota^2(\delta_{n,t,l})$$

$$\overset{(iv)}{\leq} \frac{\frac{1}{4}\hat{\beta}_t^2 + \sum_{i \in [t]} \frac{1}{\bar{\sigma}_i^2} (f^n(x_i) - f^\star(x_i))^2}{2},$$

where we use $(o)$ that $\sqrt{a+b} \leq \sqrt{a} + \sqrt{b}$, and the inequality that $\log \left( \frac{48R(1+2AR)t^2}{\min(1,a)\epsilon^2 \delta_{n,t}} \log(A/a) \right) \leq 3\iota^2(\delta_{n,t,l})$; $(i)$ the range assumption that $|f(\cdot)| \leq L_f$, choice of $\epsilon = 1$ and definition of $D_\alpha$; $(ii)$ the choice of $\bar{\sigma}_i^2 \geq 32L_f D_\mathcal{F}(x_i; x_{[i-1]}, \bar{\sigma}_{[i-1]})\iota(\delta_{n,t,l})$ and $\sqrt{a+b} \leq \sqrt{a} + \sqrt{b}$; $(iii)$ triangle inequality that $\sqrt{ab} \leq \frac{a}{2c} + \frac{c \cdot b}{2}$ where we let $a = V, b = 169\iota^2(\delta_{n,t,l})$, $c = 2$ and similarly $a^{3/4}b^{1/4} \leq \frac{a}{c} + c^3 b$ where we let $a = V$, $b = \iota^2(\delta_{n,t,l})$, and $c = 13\sqrt{2}$; and finally $(iv)$ the definition of $\hat{\beta}_t$ in (25) so that $\hat{\beta}_t^2 \geq 8 \left( 13^4 + 13^2 \right) \iota^2(\delta_{n,t,l}) + 13^2\sqrt{\lambda}\iota^2(\delta_{n,t,l})$.

This implies that

$$\mathbb{P}\left( \sum_{i \in [t]} \frac{1}{\bar{\sigma}_t^2} (f^n(x_i) - f^\star(x_i))^2 + 2t \cdot \mathsf{Catoni}_\theta \left( \left\{ \frac{1}{\bar{\sigma}_i^2} (f^n(x_i) - f^\star(x_i))(f^\star(x_i) - y_i) \right\}_{i \in [t]} \right) < -\frac{1}{4}\hat{\beta}_t^2 \right) \leq \delta_{n,t}.$$

Thus taking a union bound on all $f = f^n \in \mathcal{F}$ and $t \in [T], t \geq 7\iota^2(\delta_{n,T})$, we can conclude that with probability at least $1-\delta$ (where $\delta = \sum_{n \in [N], t \in [T]} \delta_{n,t}$) for all iteration $7\iota^2(\delta_{n,T}) \leq t \leq T$, one has $f^\star \in \hat{\mathcal{F}}_t$, i.e. $\cap_{t \in [T], t \geq 7\iota^2(\delta_{n,T})} \hat{\mathcal{F}}_t \neq \emptyset$. $\qquad \square$

**Part II: With high probability $1 - \delta$, $f^\star \in \mathcal{F}_t$ for all $t \in [\widetilde{O}(1), T]$.** We first provide the following lemma.

**Lemma D.3.** *For any function $f^n \in \mathcal{F}$, let $Z_i = \frac{1}{\bar{\sigma}_i^2} (f^\star(x_i) - f^n(x_i))(f^n(x_i) - y_i)$ and $\theta$ as defined in (24), we have for any $t \geq 7\iota^2(\delta_{n,T})$ with probability $1 - \delta_{n,t}$,*

$$t \cdot \left| \mathsf{Catoni}_\theta(Z_1, \cdots, Z_t) + \frac{1}{t} \sum_{i \in [t]} \frac{1}{\bar{\sigma}_i^2} (f^\star(x_i) - f^n(x_i))^2 \right| \leq \frac{1}{4} \sum_{i \in [t]} \frac{1}{\bar{\sigma}_i^2} (f^\star(x_i) - f^n(x_i))^2 + \frac{\hat{\beta}_t^2}{8}.$$

*Proof.* In order to apply the concentration inequality in Lemma E.1, we first bound the following

$$V = \sum_{i \in [t]} \mathbb{E} \left[ (Z_i - \mu_i)^2 | \mathcal{H}_{i-1} \right] = \sum_{i \in [t]} \frac{1}{\bar{\sigma}_i^2} (f^\star(x_i) - f^n(x_i))^2,$$

and further

$$\sum_{i \in [t]} (\mu_i - \bar{\mu})^2 \le \sum_{i \in [t]} \mu_i^2 = \sum_{i \in [t]} \frac{1}{\bar{\sigma}_i^4} (f^\star(x_i) - f^n(x_i))^2 (f^n(x_i) - f^\star(x_i))^2 \le \sum_{i \in [t]} \frac{2}{\bar{\sigma}_i^4} (f^\star(x_i) - f^n(x_i))^4.$$

Now by choice of $\theta = \sqrt{\frac{\iota^2(\delta_{n,t,l})}{\sum_{i \in [t]} \frac{1}{\bar{\sigma}_i^2} (f^n(x_i) - f^\star(x_i))^2 + \sum_{i \in [t]} \frac{2}{\bar{\sigma}_i^4} (f^n(x_i) - f^\star(x_i))^4 + \epsilon^2}}$, we will have with probability $1 - \delta_{n,t}$, for

any $t \ge 7\iota^2(\delta_{n,T})$, it holds that

$$t \cdot \left| \mathsf{Catoni}_\theta(\{Z_i\}_{i \in [t]}) - \frac{1}{t} \sum_{i \in [t]} \frac{1}{\bar{\sigma}_i^2} (f^\star(x_i) - f^n(x_i)) (f^n(x_i) - f^\star(x_i)) \right|$$

$$\le 13 \sqrt{\left( \sum_{i \in [t]} \frac{1}{\bar{\sigma}_i^2} (f^\star(x_i) - f^n(x_i))^2 \right) \cdot \left( 1 + \max_{i \in [t]} \frac{2}{\bar{\sigma}_i^2} (f^\star(x_i) - f^n(x_i))^2 \right) \iota(\delta_{n,t,l})} + 13\iota(\delta_{n,t,l}).$$

We now proceed to bound $\max_{i \in [t]} \frac{2}{\bar{\sigma}_i^2} (f^n(x_i) - f^\star(x_i))^2$, we assume $|f^n(x) - f^\star(x)| \le 2L_f$ for any $x \in \mathcal{X}$, consequently we have

$$\max_{i \in [t]} \frac{2}{\bar{\sigma}_i^2} (f^n(x_i) - f^\star(x_i))^2 \le \max_{i \in [t]} \frac{4L_f}{\bar{\sigma}_i^2} \sqrt{(f^n(x_i) - f^\star(x_i))^2}$$

$$\le \max_{i \in [t]} \frac{4L_f}{\bar{\sigma}_i^2} D_\mathcal{F}(x_i; x_{[i-1]}, \bar{\sigma}_{[i-1]}) \sqrt{\sum_{k \in [i]} \frac{1}{\bar{\sigma}_k^2} (f^n(x_k) - f^\star(x_k))^2 + \lambda}$$

$$\le \frac{1}{8\iota(\delta_{n,t,l})} \sqrt{\sum_{i \in [t]} \frac{1}{\bar{\sigma}_i^2} (f^n(x_k) - f^\star(x_k))^2 + \lambda}$$

$$\le \frac{1}{8\iota(\delta_{n,t,l})} \sqrt{\sum_{i \in [t]} \frac{1}{\bar{\sigma}_i^2} (f^n(x_k) - f^\star(x_k))^2} + \frac{\sqrt{\lambda}}{8},$$

where for the last inequality we use the fact that $\bar{\sigma}_i^2 \ge 32 L_f \iota(\delta_{n,t,l}) D_\mathcal{F}^2(x_i; x_{[i-1]}, \bar{\sigma}_{[i-1]})$ by the choice of $\bar{\sigma}$.

Plugging this back we can conclude that

$$t \cdot \left| \mathsf{Catoni}_\theta(\{Z_i\}_{i \in [t]}) + \frac{1}{t} \sum_{i \in [t]} \frac{1}{\bar{\sigma}_i^2} (f^\star(x_i) - f^n(x_i))^2 \right| - 13\iota(\delta_{n,t,l})$$

$$\le 13 \sqrt{\left( 1 + \frac{\sqrt{\lambda}}{8} \right) \cdot \sqrt{\sum_{i \in [t]} \frac{1}{\bar{\sigma}_i^2} (f^\star(x_i) - f^n(x_i))^2 \iota(\delta_{n,t,l})}} + \frac{13}{2\sqrt{2}} \left( \sum_{i \in [t]} \frac{1}{\bar{\sigma}_i^2} (f^n(x_t) - f^\star(x_i))^2 \right)^{3/4} (\iota^2(\delta_{n,t,l}))^{1/4}$$

$$\le \frac{1}{8} \sum_{i \in [t]} \frac{1}{\bar{\sigma}_i^2} (f^n(x_t) - f^\star(x_i))^2 + 2 \cdot 13^2 \iota^2(\delta_{n,t,l}) \left( 1 + \frac{\sqrt{\lambda}}{8} \right) + \frac{1}{8} \sum_{i \in [t]} \frac{1}{\bar{\sigma}_i^2} (f^n(x_t) - f^\star(x_i))^2 + 8 \cdot 13^4 \iota^2(\delta_{n,t,l})$$

$$\implies t \cdot \left| \mathsf{Catoni}_\theta(\{Z_i\}_{i \in [t]}) + \frac{1}{t} \sum_{i \in [t]} \frac{1}{\bar{\sigma}_i^2} (f^\star(x_i) - f^n(x_i))^2 \right| \le \frac{1}{4} \sum_{i \in [t]} \frac{1}{\bar{\sigma}_i^2} (f^n(x_t) - f^\star(x_i))^2 + \frac{1}{8} \hat{\beta}_t^2,$$

where for the last inequality we use the definition of $\hat{\beta}_t$ so that $\hat{\beta}_t^2 \ge 8(8 \cdot 13^4 + 2 \cdot 13^2 + 13)\iota^2(\delta_{n,t,l}) + 2 \cdot 13^2 \sqrt{\lambda} \iota^2(\delta_{n,t,l})$. This concludes the proof of lemma. $\square$

**Corollary D.4.** *With high probability* $1 - \delta$ *where* $\delta = \sum_{n \in [N], t \in [T]} \delta_{n,t}$, *we have* $f^\star \in \mathcal{F}_t$ *for all* $t \in [T - 1]$ *satisfying* $t \ge 7\iota^2(\delta_{n,T})$.

*Proof.* We bound the probability by a union bound argument. Let $\mathcal{T} := \{t \in [T-1] : t \geq 7\iota^2(\delta_{n,T})\}$ and $Z_i' = \frac{1}{\sigma_i^2}(f^\star(x_i) - f^n(x_i))(f^n(x_i) - y_i)$, we have

$$\mathbb{P}\left(\exists\, t, 7\iota^2(\delta_{n,T}) \leq t \leq T, f^\star \notin \mathcal{F}_t\right) \leq \sum_{n\in[N], t\in\mathcal{T}} \mathbb{P}\left(\hat{f}_t = f^n, \sum_{i\in[t]} \frac{1}{\sigma_i^2}(f^\star(x_i) - f^n(x_i))^2 > \hat{\beta}_t^2\right)$$

$$\leq \sum_{n\in[N], t\in\mathcal{T}} \mathbb{P}\left(\begin{array}{c} \sum_{i\in[t]} \frac{1}{\sigma_i^2}(f^\star(x_i) - f^n(x_i))^2 + 2t\cdot\mathsf{Catoni}_\theta\left(\{Z_i'\}_{i\in[t]}\right) \geq -\frac{1}{4}\hat{\beta}_t^2 \\ \sum_{i\in[t]} \frac{1}{\sigma_i^2}(f^\star(x_i) - f^n(x_i))^2 > \hat{\beta}_t^2 \end{array}\right)$$

$$= \sum_{n\in[N], t\in\mathcal{T}} \mathbb{P}\left(\begin{array}{c} \sum_{i\in[t]} \frac{1}{\sigma_i^2}(f^\star(x_i) - f^n(x_i))^2 \leq \frac{1}{4}\hat{\beta}_t^2 + 2t\cdot\left|\mathsf{Catoni}_\theta\left(\{Z_i'\}_{i\in[t]}\right) + \frac{1}{t}\sum_{i\in[t]}\frac{1}{\sigma_i^2}(f^\star(x_i) - f^n(x_i))^2\right| \\ \sum_{i\in[t]} \frac{1}{\sigma_i^2}(f^\star(x_i) - f^n(x_i))^2 > \hat{\beta}_t^2 \end{array}\right)$$

$$\leq \sum_{n\in[N], t\in\mathcal{T}} \mathbb{P}\left(t\cdot\left|\mathsf{Catoni}_\theta\left(\{Z_i'\}_{i\in[t]}\right) + \frac{1}{t}\sum_{i\in[t]}\frac{1}{\sigma_i^2}(f^\star(x_i) - f^n(x_i))^2\right| > \frac{1}{4}\sum_{i\in[t]}\frac{1}{\sigma_i^2}(f^\star(x_i) - f^n(x_i))^2 + \frac{1}{8}\hat{\beta}_t^2\right)$$

$$\leq \sum_{n\in[N], t\in\mathcal{T}} \delta_{n,t} \leq \sum_{n\in[N], t\in[T]} \delta_{n,t} \leq \delta.$$

$\square$

**Part III: Bounding the regret conditioning on good events.** We now recall the definition that $\mathcal{T} := \{t \in [T] : t \geq 7\iota^2(\delta_{n,T})\}$, we further denote the good events $\mathcal{E} = \{\cap_{t\in\mathcal{T}}\hat{\mathcal{F}}_t \neq \emptyset\}$ and $\mathcal{E}' = \{f^\star \in \cap_{t\in\mathcal{T}}\mathcal{F}_t\}$.

*Proof of Theorem D.1.* Conditioning on both good events $\mathcal{E} \cap \mathcal{E}'$, we let $f^{n_t}$ be the function maximizer in set $\mathcal{F}_{t-1}$ we pick at step $t$ and can bound the regret by

$$R_T = \sum_{t\in[T]} r_t = \sum_{t\in[T]} (f^\star(x_t^\star) - f^\star(x_t)) \leq 2L_f\left(7\iota^2(\delta_{n,T}) + 2\right) + \sum_{t-1\in\mathcal{T}}(f^{n_t}(x_t) - f^\star(x_t))$$

$$\leq 2L_f\left(7\iota^2(\delta_{n,T}) + 2\right) + \sum_{t-1\in\mathcal{T}}\left(\left|f^{n_t}(x_t) - \hat{f}_{t-1}(x_t)\right| + \left|\hat{f}_{t-1}(x_t) - f^\star(x_t)\right|\right)$$

$$\overset{(i)}{\leq} O\left(L_f \cdot \iota^2(\delta_{n,T})\right) + \sum_{t-1\in\mathcal{T}}\left[\bar{\sigma}_t D_\mathcal{F}(x_t, \bar{\sigma}_t; x_{[t-1]}, \bar{\sigma}_{[t-1]}) \cdot \left(\sqrt{\sum_{i\in[t-1]}\frac{1}{\sigma_i^2}\left(f^{n_t}(x_i) - \hat{f}_{t-1}(x_i)\right)^2 + \lambda}\right.\right.$$

$$\left.\left. + \sqrt{\sum_{i\in[t-1]}\frac{1}{\sigma_i^2}\left(f^\star(x_i) - \hat{f}_{t-1}(x_i)\right)^2 + \lambda}\right)\right]$$

$$\overset{(ii)}{\leq} O\left(L_f \cdot \iota^2(\delta_{n,T})\right) + \sum_{t-1\in\mathcal{T}} 2\bar{\sigma}_t D_\mathcal{F}(x_t, \bar{\sigma}_t; x_{[t-1]}, \bar{\sigma}_{[t-1]}) \cdot \left(\sqrt{\hat{\beta}_{t-1}^2 + \lambda}\right),$$

where we use $(i)$ the definition of $D_\alpha$ for each $t \in [T]$ and $(ii)$ the definition of $\mathcal{F}_{t-1}$ and that $f^{n_t}, f^\star \in \mathcal{F}_{t-1}$ conditioning on $\mathcal{E}$ and $\mathcal{E}'$.

Combining this with the range bound that $[0, 1]$ of each individual reward one may receive by assumption, one can conclude that

$$R_T \leq O\left(L_f \cdot \iota^2(\delta_{n,T})\right) + 2L_f \sum_{t-1\in\mathcal{T}} \min\left(1, \bar{\sigma}_t D_\mathcal{F}(x_t, \bar{\sigma}_t; x_{[t-1]}, \bar{\sigma}_{[t-1]})\sqrt{\hat{\beta}_{t-1}^2 + \lambda}\right). \tag{26}$$

To finally bound the regret, we bound the second term in RHS of $R_T$ expression in (26) respectively. These steps mainly follow Lemma 4.4 in (Zhou & Gu, 2022). We can decompose the terms by considering $\mathcal{I}_1 = \{t - 1 \in \mathcal{T} | D_\mathcal{F}(x_t, \bar{\sigma}_t; x_{[t-1]}, \bar{\sigma}_{[t-1]}) \geq 1\}$ and $\mathcal{I}_2 = \{t - 1 \in \mathcal{T}, t \notin \mathcal{I}_1\}$.

For the first set, we bound its size naively by

$$|\mathcal{I}_1| \leq \sum_{t \in \mathcal{I}_1} \min\left(1, D_{\mathcal{F}}^2(x_t, \bar{\sigma}_t; x_{[t-1]}, \bar{\sigma}_{[t-1]})\right) \leq \dim_{\alpha, T}(\mathcal{F}).$$

For the second set, we bound the summation of terms of interest contraining on $\mathcal{I}_2$ by

$$\sum_{t \in \mathcal{I}_2} \bar{\sigma}_t \sqrt{\hat{\beta}_{t-1}^2 + \lambda} \cdot D_{\mathcal{F}}(x_t, \bar{\sigma}_t; x_{[t-1]}, \bar{\sigma}_{[t-1]})$$

$$\leq \sum_{t \in \mathcal{I}_2, \bar{\sigma}_t = \sigma_t \text{ or } \alpha} \bar{\sigma}_t \sqrt{\hat{\beta}_{t-1}^2 + \lambda} \cdot D_{\mathcal{F}}(x_t, \bar{\sigma}_t; x_{[t-1]}, \bar{\sigma}_{[t-1]})$$

$$+ \sum_{t \in \mathcal{I}_2, \bar{\sigma}_t = 4\sqrt{2\iota(\delta_{n,t,l}) L_f D_{\mathcal{F}}(x_t; x_{[t-1]}, \bar{\sigma}_{[t-1]})}} \bar{\sigma}_t \sqrt{\hat{\beta}_{t-1}^2 + \lambda} \cdot D_{\mathcal{F}}(x_t, \bar{\sigma}_t; x_{[t-1]}, \bar{\sigma}_{[t-1]})$$

$$\overset{(i)}{\leq} \sum_{t \in [T]} (\sigma_t + \alpha) \sqrt{\hat{\beta}_{t-1}^2 + \lambda} \cdot D_{\mathcal{F}}(x_t, \bar{\sigma}_t; x_{[t-1]}, \bar{\sigma}_{[t-1]}) + \sum_{t \in [T]} 32 L_f \iota(\delta_{n,t,l}) \sqrt{\hat{\beta}_{t-1}^2 + \lambda} \cdot D_{\mathcal{F}}^2(x_t, \bar{\sigma}_t; x_{[t-1]}, \bar{\sigma}_{[t-1]})$$

$$\overset{(ii)}{\leq} \sqrt{2 \sum_{t \in [T]} (\hat{\beta}_{t-1}^2 + \lambda)(\sigma_t^2 + \alpha^2)} \sqrt{\dim_{\alpha, T}(\mathcal{F})} + 16 L_f \iota(\delta_{n,t,l}) \max_{t \in [T]} \sqrt{\hat{\beta}_{t-1}^2 + \lambda} \cdot \dim_{\alpha, T}(\mathcal{F}).$$

Here for $(i)$ we use the condition for each distinct set and for $(ii)$ we use Cauchy-Schwarz inequality for the first term and the definition of $\dim_\alpha$ for both terms.

Consequently plugging these back in (26) and take supremum over $x : |x| = T$, we conclude that with probability at least $1 - 2\delta$,

$$R_T = O\left(L_f \cdot \iota^2(\delta_{n,T}) + \dim_{\alpha, T}(\mathcal{F}) + L_f \cdot \iota(\delta_{n,t,l}) \cdot \max_{t \in [T]} \sqrt{\hat{\beta}_{t-1}^2 + \lambda} \cdot \dim_{\alpha, T}(\mathcal{F})\right.$$

$$\left. + L_f \sqrt{\sum_{t \in [T]} \left(\hat{\beta}_{t-1}^2 + \lambda\right)(\sigma_t^2 + \alpha^2)} \cdot \sqrt{\dim_{\alpha, T}(\mathcal{F})}\right)$$

$$= \widetilde{O}\left(L_f \cdot \log N \cdot \dim_{\frac{1}{\sqrt{T}}, T}(\mathcal{F}) + L_f \sqrt{\sum_{t \in [T]} \sigma_t^2} \cdot \sqrt{\dim_{\frac{1}{\sqrt{T}}, T}(\mathcal{F}) \cdot \log N}\right),$$

where for the last inequality we pick $\lambda = \Theta(1)$, $\alpha = 1/\sqrt{T}$.

$\square$

### D.2. Unknown Variance

We first choose the probability parameter as follows:

$$\delta_{n,t,l} := \frac{\delta}{N(T+1)L}, \quad \delta_t := \sum_{l \in [L]} \sum_{n \in [N]} \delta_{n,t,l} = \frac{\delta}{T+1}. \tag{27}$$

Note again we have $\sum_{t \in [T]} \delta_t \leq \delta$.

---

**Algorithm 4** Variance-Agnostic Catoni Bandit with Candidate Set

---
1: **Input:** Parameter $\gamma > 0$, $L = \lceil \log_2(1/\gamma) \rceil$, $l_\star = \lceil \log_2(1076\iota'(\delta)) \rceil$.
2: Initialize the estimators for all layers: $\lambda^l \leftarrow 2^{-2l}$, $\hat{\beta}_0^l \leftarrow 2^{-l+1}$, $\Psi_0^l \leftarrow \emptyset$ for all $l \in [l_\star, L]$.
3: **for** t=1,...,T **do**
4:     Observe $\mathcal{X}_t$, and initialize $\mathcal{X}_t^1 \leftarrow \mathcal{X}_t$, $l \leftarrow l_\star$.
5:     **while** $x_t$ is not specified **do**
6:        **if** $D_t^l(x) \leq \gamma$ for all $x \in \mathcal{X}_t^l$ **then**
7:           Choose $x_t$, $f_{t-1}^l \leftarrow \arg\max_{x \in \mathcal{X}_t^l, f \in \mathcal{F}_{t-1}^l} f(x)$
8:           Observe $y_t$.
9:           **Break.**
10:        **else if** $D_t^l(x) \leq 2^{-l}$ for all $x \in \mathcal{X}_t^l$ **then**
11:           Update $\mathcal{X}_t^{l+1} \leftarrow \{x \in \mathcal{X}_t^l \mid \hat{f}_{t-1}^l(x) \geq \max_{x \in \mathcal{X}_t^l} \hat{f}_{t-1}^l(x) - 2^{-l+1}\hat{\beta}_{t-1}^l\}$.
12:        **else**
13:           Choose $x_t \in \mathcal{X}_t^l$ such that $D_t^l(x_t) > 2^{-l}$ and observe $y_t$.
14:           Update $w_t \leftarrow 2^l D_t^l(x_t)$.
15:           Update the index sets: $\Psi_t^l \leftarrow \Psi_{t-1}^l \cup \{t\}$ and $\Psi_t^{l'} \leftarrow \Psi_{t-1}^{l'}$ for $l' \neq l$.
16:           Pick any $\hat{f}_t^l \in \hat{\mathcal{F}}_t^l$ as in (28), and choose the confidence set $\mathcal{F}_t^l$ defined in (32).
17:        **end if**
18:        Update $l \leftarrow l + 1$.
19:     **end while**
20:     For $l \in [L]$ s.t. $\Psi_t^l = \Psi_{t-1}^l$, $\hat{f}_t^l \leftarrow \hat{f}_{t-1}^l$, $\mathcal{F}_t^l \leftarrow \mathcal{F}_{t-1}^l$.
21: **end for**

---

For each layer $l \in [L]$, define $Z_i = \frac{1}{\sigma_i^2}(f(x_i) - \hat{f}(x_i))(\hat{f}(x_i) - y_i)$ and

$$\hat{\mathcal{F}}_t^l := \left\{ \hat{f} \in \mathcal{F} : \min_{f \in \mathcal{F}} \frac{1}{\sum_{i \in \Psi_t^l} w_i^2} \left( f(x_i) - \hat{f}(x_i) \right)^2 + 2t \cdot \mathsf{Catoni}_{\theta(f,\hat{f})}(Z_1, \cdots, Z_t) \geq -\frac{1}{4}(\hat{\beta}_t^l)^2 \right\}, \tag{28}$$

$$\text{where } \theta(f, \hat{f}) = \sqrt{\frac{(\iota'(\delta_{n,t}))^2}{\sum_{i \in \Psi_t^l} \frac{1}{w_i^2} \left( f(x_i) - \hat{f}(x_i) \right)^2 + \sum_{i \in \Psi_t^l} \frac{2}{w_i^4} \left( f(x_i) - \hat{f}(x_i) \right)^4 + \epsilon^2}}, \tag{29}$$

and $\mathsf{Catoni}_\theta$ is the $\theta$-robust mean estimator for $\frac{1}{t} \sum_{i \in \Psi_t^l} \frac{1}{w_i^2} \left( f(x_i) - \hat{f}(x_i) \right) \left( \hat{f}(x_i) - f^\star(x_i) \right)$. The confidence radius is specified as

$$(\hat{\beta}_t^l)^2 = \Theta\left( (\iota'(\delta))^2 2^{-2l} \widehat{\mathrm{Var}}_t^l + \iota'(\delta) 2^{-2l} + \lambda^l \right), \tag{30}$$

where $\widehat{\mathrm{Var}}_t^l$ should satisfy event $\mathcal{E}_{\mathrm{var}}^{t,l}$ holds with high probability:

$$\widehat{\mathrm{Var}}_t^l := t\mathsf{Catoni}_{\theta_{\mathrm{Var}}^{t,l}} \left( \left\{ \frac{1}{w_i^2}(y_i - \hat{f}_{t-1}^l(x_i))^2 \right\}_{i \in \Psi_t^l} \right) + 14\iota'(\delta)(2\sigma_\eta^2 + c_\eta) + 268\lambda^l. \tag{31}$$

Then, we can define the confidence set

$$\mathcal{F}_t^l := \left\{ f \in \mathcal{F} : \sum_{i \in \Psi_t^l} \frac{1}{w_i^2} \left( f(x_i) - \hat{f}_t^l(X_i) \right)^2 \leq (\hat{\beta}_t^l)^2 \right\}. \tag{32}$$

The result is provided in the following theorem, where the bound is basically the same as the one-confidence set case in Theorem 4.2.

**Theorem D.5.** *Suppose that Assumption 4.1 holds. Under Algorithm 4 with $\gamma = 1/(L_f T^{3/2})$, if $T \geq 14\iota^2(\delta_{n,T,l})$, with probability at least $1 - 3\delta$, we can bound the regret by*

$$R_T = \widetilde{O}\Big( L_f \sqrt{\sum_{t \in [T]} \sigma_t^2 \cdot \dim_{\frac{1}{\sqrt{T}},T}(\mathcal{F}) \cdot \log N} + L_f \cdot \log N \cdot \dim_{\frac{1}{\sqrt{T}},T}(\mathcal{F}) \Big).$$

For the analysis, we now divide the argument into the following parts. For conciseness, we will omit the proofs similar to the previous analysis.

**Lemma D.6.** *Under Assumption 4.1, given $\{X_i, y_i, w_i\}_{i \in [t]}$, we define $D_i^2 = \sup_{f, f' \in \mathcal{F}} \frac{|f(x_i) - f'(x_i)|}{\sqrt{\sum_{l=1}^{i-1}(f(x_l) - f'(x_l))^2/w_i^2 + \lambda}}$. If $\sup_{i \in [t]} \frac{D_i}{w_i} \leq \rho$, and $\mathcal{E}_{\mathrm{var}}^{t,l} := \{\sum_{i \in [t]} \frac{\sigma_i^2}{w_i^2} \leq 2\widehat{\mathrm{Var}}_t\}$ holds true, then, for the time step $t$ such that*

$$\mathcal{E}_t = \Big\{ t \geq 4(\iota'(\delta_{n,t,l}))^2 \frac{\sum_{i \in [t]} \sigma_i^2/w_i^2}{\widehat{\mathrm{Var}}_t} + 6(\iota'(\delta_{n,t,l}))^2 \Big\},$$

*there exists some $f = f^n \in \mathcal{F}$ such that with probability at least $1 - \delta$,*

$$t \cdot \mathsf{Catoni}\left( \Big\{ \frac{1}{w_i^2} (f(x_i) - f^\star(x_i)) (f^\star(x_i) - y_i) \Big\}_{i \in [t]} \right) \geq \frac{-\frac{1}{4}\beta_t^2 - \sum_{i \in [t]} \frac{1}{w_i^2} (f(x_i) - f^\star(x_i))^2}{2}.$$

*Proof.* Now we bound the probability that the above inequality happens. We apply Lemma E.1 to

$$Z_i = \frac{1}{w_i^2} (f(x_i) - f^\star(x_i)) (f^\star(x_i) - y_i),$$

which has conditional mean $\mu_i = 0$, and sum of conditional variance bounded by

$$\sum_{i \in [t]} \mathrm{Var}[Z_i(f, f')] = \sum_{i \in [t]} \mathbb{E}\Big[ \frac{1}{w_i^4} (f(x_i) - f'(x_i))^2 (f^\star(x_i) - y_i)^2 \,\Big|\, x_i \Big]$$

$$\leq \sum_{i \in [t]} \frac{(f(x_i) - f'(x_i))^2}{w_i^2} \cdot \frac{\sigma_i^2}{w_i^2}$$

$$\leq 4\rho^2 \hat{\beta}_{t-1}^2 \sum_{i \in [t]} \frac{\sigma_i^2}{w_i^2} \leq 8\rho^2 \hat{\beta}_{t-1}^2 \widehat{\mathrm{Var}}_t.$$

Then, follow the proof steps of Lemma C.2, we can obtain that with probability at least $1 - \delta_{n,t}$, we have from Corollary 3.2 that

$$t \cdot \Big| \mathsf{Catoni}_{\theta_t(f,f')}(\{Z_i(f, f')\}_{i \in [t]}) \Big| \leq \frac{\frac{1}{4}(\hat{\beta}_t^l)^2 + V_t(f, f')}{2}.$$

Thus taking a union bound on all $f = f^n \in \mathcal{F}$, we can conclude that with probability at least $1 - \delta_t$,

$$t \cdot \Big| \mathsf{Catoni}_\theta(\{Z_i\}_{i \in [t]}) \Big| \leq \frac{\frac{1}{4}(\hat{\beta}_t^l)^2 + V_t(f, f')}{2}.$$

$\square$

**Lemma D.7.** *Conditioned on $\mathcal{E}_{\mathrm{var}}^{t,l}$, for any iteration $t \in [T], t$ satisfying $\mathcal{E}_t$, and any layer $l \in [L]$ and the set $\hat{\mathcal{F}}_t^l$ as constructed in (28), we have with probability at least $1 - \delta$, $\cap_{t \in [T], t \geq 7\iota^2(\delta_{n,T})} \hat{\mathcal{F}}_t^l \neq \emptyset$.*

*Proof.* It suffices to show that for each $t$ and $l$, we have $\mathbb{P}(f^\star \notin \hat{\mathcal{F}}_t^l) \leq \delta_t$. When $f^\star \notin \hat{\mathcal{F}}_t^l$, there exists some $f = f^n \in \mathcal{F}$ such that with probability at least $1 - \delta$

$$\sum_{i \in \Psi_t^l} \frac{1}{w_t^2} (f(x_i) - f^\star(x_i))^2 + 2t \cdot \mathsf{Catoni}_\alpha \left( \Big\{ \frac{1}{w_i^2} (f(x_i) - f^\star(x_i)) (f^\star(x_i) - y_i) \Big\}_{i \in \Psi_t^l} \right) < -\frac{1}{4}(\hat{\beta}_t^l)^2,$$

which implies that

$$t \cdot \mathsf{Catoni}\left(\left\{\frac{1}{w_i^2}\left(f(x_i) - f^\star(x_i)\right)\left(f^\star(x_i) - y_i\right)\right\}_{i \in \Psi_t^l}\right) < \frac{-\frac{1}{4}(\hat{\beta}_t^l)^2 - \sum_{i \in \Psi_t^l}\frac{1}{w_i^2}\left(f(x_i) - f^\star(x_i)\right)^2}{2}.$$

Lemma D.6 implies that

$$\mathbb{P}\left(\sum_{i \in [t]}\frac{1}{w_t^2}\left(f(x_i) - f^\star(x_i)\right)^2 + 2t \cdot \mathsf{Catoni}_\theta\left(\left\{\frac{1}{w_i^2}\left(f(x_i) - f^\star(x_i)\right)\left(f^\star(x_i) - y_i\right)\right\}_{i \in [t]}\right) < -\frac{1}{4}(\hat{\beta}_t^l)^2\right) \le \delta_{n,t,l}.$$

Thus taking a union bound on all $f = f^n \in \mathcal{F}$ and $t \in [T]$, $t$ satisfying $\mathcal{E}_t$, we can conclude that with probability at least $1 - \delta$ (where $\delta = \sum_{n \in [N], t \in [T]} \delta_{n,t,l}$) for all iteration $7\iota^2(\delta_{n,T,l}) \le t \le T$, one has $f^\star \in \hat{\mathcal{F}}_t^l$, i.e. $\cap_{t \in [T], t \ge 7\iota^2(\delta_{n,T,l})}\hat{\mathcal{F}}_t^l \ne \emptyset$. $\square$

**With probability at least** $1 - \delta$, $f^\star \in \mathcal{F}_t^l$ **for all** $t \in [\tilde{O}(1), T]$, $l \in [L]$.

**Corollary D.8.** *Suppose that Assumption 4.1 holds. With high probability $1 - \delta$ where $\delta = \sum_{l \in [L], n \in [N], t \in [T]} \delta_{n,t}$, we have*

$$\cup_{t \in [T]: t \ge 3(\iota'(\delta))^2}\mathcal{E}_{\text{conv}}^t := \left\{f^\star \in \mathcal{F}_t^l, \text{ for } l \in [L],\ 2^l \ge 1076\iota'(\delta)\right\}.$$

*Proof.* Conditioned on $\mathcal{E}_{\text{var}}^{t,l}$, we can follow the proof for the known-variance setting and get

$$f^\star \in \mathcal{F}_t^l.$$

For each $l \in [L]$, so it remains to prove that $\mathcal{E}_{\text{var}}^{t,l}$ holds true. Following similar analysis of Lemma C.6, we can show that conditioned on

$$\{f^\star \in \mathcal{F}_s^l,\ \forall l \in [L], s \in [7\iota^2(\delta_{n,T,l}), t - 1]\},$$

the event $\mathcal{E}_{\text{var}}^{t,l}$ holds with probability at least $1 - \delta$. $\square$

Recall the value of $\hat{\beta}_t^l$:

$$(\hat{\beta}_t^l)^2 = 39^2(\iota'(\delta_{n,t,l}))^2 2^{-2l}\widehat{\text{Var}}_t^l + 52^2(\iota'(\delta_{n,t,l}))^2 2^{-2l}\sqrt{\lambda} + 8 \times 52^4\iota^4(\delta_{n,t,l})2^{-4l} + \frac{\lambda}{4}.$$

We can also demonstrate the relation between $\hat{\beta}_t^l$ and the sum of variance $\sum_{i \in \Psi_t^l}\frac{\sigma_i^2}{w_i^2}$ in the following lemma via similar proofs with Lemma C.7.

**Lemma D.9.** *Conditioned on $\mathcal{E}_{\text{var}}^t$, if we take the confidence parameter as (8), We have*

$$(\hat{\beta}_t^l)^2 \le \Theta\left((\iota'(\delta))^2 2^{-2l}\sum_{i \in \Psi_t^l}\frac{\sigma_i^2}{w_i^2} + (\iota'(\delta))^3 2^{-2l}(\sigma_\eta^2 + c_\eta) + \iota'(\delta)2^{-2l} + \lambda^l\right).$$

*Additionally, with probability at least $1 - \delta$, the following event occurs:*

$$\cup_{t \in [T]: t \ge 3(\iota'(\delta))^2}\mathcal{E}_{\text{conv}}^t := \left\{f^\star \in \mathcal{F}_t^l, \text{ for } l \in [L],\ 2^l \ge 1076\iota'(\delta)\right\}.$$

Finally, following the proofs of Theorem D.1 and 4.2, we can obtain the result of Theorem D.5.

## E. Auxiliary Proofs

### E.1. Concentration Inequality for Catoni Estimator

**Lemma E.1** (Concentration for Catoni estimator, cf. Lemma 13 in (Wei et al., 2020))**.** *Let $Z_t$ be random variable adapted to filtration $\mathcal{H}_t$, suppose $\mathbb{E}[Z_i|\mathcal{H}_{i-1}] = \mu_i$, $\sum_{i \in [t]}\mathbb{E}\left[(Z_i - \mu_i)^2|\mathcal{H}_{i-1}\right] \le V$ for some fixed $V$. Let $\bar{\mu} := \frac{1}{t}\sum_{i \in [t]}\mu_i$, for some fixed parameter $\theta > 0$, we have for any $t \ge \theta^2\left(V + \sum_{i \in [t]}(\mu_i - \bar{\mu})^2\right) + 2\log(1/\delta)$, with probability at least $1 - 2\delta$,*

$$\left|\mathsf{Catoni}_\theta(\{Z_i\}_{i \in [t]}) - \bar{\mu}\right| \le \frac{\theta\left(V + \sum_{i \in [t]}(\mu_i - \bar{\mu})^2\right)}{t} + \frac{2\log(1/\delta)}{\theta t}.$$

We provide the following lemma used in proving Lemma 3.2

**Lemma E.2** (Sensitivity of Catoni estimator, cf. Lemma A.13 of (Wagenmaker et al., 2022)). *Consider some fixed* $Z = \{Z_i\}_{i \in [t]}$, $\tilde{Z} = \{\tilde{Z}_i\}_{i \in [t]}$ *satisfying* $|Z_i| \leq R$, $|\tilde{Z}_i| \leq R$ *for all* $i \in [t]$, *and some fixed* $\theta > 0$, $\tilde{\theta} > 0$. *Then, assuming that*

$$\Delta := \frac{1}{t} \sum_{i \in [t]} \theta |Z_i - \tilde{Z}_i| + 3R|\theta - \tilde{\theta}| \leq \frac{1}{18} \min\{1, \theta^2 R^2\},$$

*we will have*

$$\left| \mathrm{Catoni}_\theta(\{Z_i\}_{i \in [t]}) - \mathrm{Catoni}_{\tilde{\theta}}(\{\tilde{Z}_i\}_{i \in [t]}) \right| \leq \frac{1 + 2\theta R}{\theta} \Delta + \sqrt{\frac{2\Delta}{\theta^2}}.$$

**Lemma E.3** (Formal version of Lemma 3.2). *Let* $Z_t$ *be a random variable adapted to filtration* $\mathcal{H}_t$ *with a uniform bound* $|Z_t| \leq R$, $\mathbb{E}[Z_i | \mathcal{H}_{i-1}] = \mu_i$, $\sum_{i \in [t]} \mathbb{E}\left[(Z_i - \mu_i)^2 | \mathcal{H}_{i-1}\right] \leq V$ *for some fixed* $V$. *Let* $\bar{\mu} := \frac{1}{t} \sum_{i \in [t]} \mu_i$. *For any parameter* $\theta \in [a, A]$ *and given* $\epsilon \leq 24R(1 + 2AR)t^2$, *if* $t \geq \theta^2(V + \sum_{i \in [t]}(\mu_i - \bar{\mu})^2) + 2\log(\frac{48R(1+2AR)t^2}{\min(1,a)\epsilon^2\delta}\log(A/a))$, *with probability at least* $1 - 2\delta$,

$$\left| \mathrm{Catoni}_\theta(\{Z_i\}_{i \in [t]}) - \bar{\mu} \right| \leq \frac{\theta\left(V + \sum_{i \in [t]}(\mu_i - \bar{\mu})^2\right)}{t} + \frac{4\iota_0^2}{\theta t} + \frac{\epsilon}{t},$$

*where*

$$\iota_0^2 = 4\log\left(\frac{48R(1+2AR)t^2}{\min(1,a)\epsilon^2\delta}\log(A/a)\right).$$

*Proof of Lemma 3.2.* For any $\epsilon > 0$, set $\kappa = \frac{\epsilon^2 \cdot \min(1,a)}{24R(1+2AR)t^2} \leq 1$, we consider a set $\mathcal{A} = \{\theta = (1+\kappa)^j \cdot a \mid (1+\kappa)^j \cdot a \in [a, A], j \geq 0\}$, it is immediate to see $|\mathcal{A}| \leq \frac{48R(1+2AR)t^2}{\min(1,a)\epsilon^2}\log(A/a)$. Now for any $\theta \in \mathcal{A}$, we have by Lemma E.1 that with probability $1 - 2\delta/|\mathcal{A}|$, we have for any $t \geq \theta^2\left(V + \sum_{i \in [t]}(\mu_i - \bar{\mu})^2\right) + 2\log(|\mathcal{A}|/\delta)$,

$$\left| \mathrm{Catoni}_\theta(\{Z_i\}_{i \in [t]}) - \bar{\mu} \right| \leq \frac{\theta\left(V + \sum_{i \in [t]}(\mu_i - \bar{\mu})^2\right)}{t} + \frac{2\log(|\mathcal{A}|/\delta)}{\theta t}.$$

Thus by taking a union bound over $\theta \in \mathcal{A}$, we have with probability $1 - 2\delta$, for all $\theta \in \mathcal{A}$, it holds that for any $t \geq \theta^2\left(V + \sum_{i \in [t]}(\mu_i - \bar{\mu})^2\right) + 2\log(|\mathcal{A}|/\delta)$,

$$\left| \mathrm{Catoni}_\theta(\{Z_i\}_{i \in [t]}) - \bar{\mu} \right| \leq \frac{\theta\left(V + \sum_{i \in [t]}(\mu_i - \bar{\mu})^2\right)}{t} + \frac{2\log(|\mathcal{A}|/\delta)}{\theta t}.$$

Thus, for any $\theta' \in [a, A]$, we have there exists some $\theta \in \mathcal{A}$ such that the above bound holds true and $\frac{|\theta' - \theta|}{\theta} \leq \kappa \leq \min\left(\frac{\epsilon}{2 \cdot 3R(1+2AR)t}, \frac{a \cdot \epsilon^2}{24Rt^2}\right)$. Now by triangle inequality we can conclude that for any $\theta \in [a, A]$,

$$
\begin{aligned}
\left| \mathrm{Catoni}_\theta - \bar{\mu} \right| &\leq \left| \mathrm{Catoni}_{\theta_0} - \bar{\mu} \right| + \left| \mathrm{Catoni}_\theta - \mathrm{Catoni}_{\theta_0} \right| \\
&\leq \frac{\theta_0\left(V + \sum_{i \in [t]}(\mu_i - \bar{\mu})^2\right)}{t} + \frac{2\log(|\mathcal{A}|/\delta)}{\theta_0 t} + \frac{\epsilon}{t} \\
&\leq \frac{\theta\left(V + \sum_{i \in [t]}(\mu_i - \bar{\mu})^2\right)}{t} + \frac{4\log(|\mathcal{A}|/\delta)}{\theta t} + \frac{\epsilon}{t},
\end{aligned}
$$

where for the second inequality we use Lemma E.2. $\qquad\square$

