# OpenReview forum: "Catoni Contextual Bandits are Robust to Heavy-tailed Rewards"
_ICML.cc/2025/Conference — ICML 2025 spotlightposter_

### Official Review · Reviewer_PCY8 · 2025-03-06

**Overall Recommendation:** 3

**Summary:**

This paper studies contextual bandits with heacy-tailed reward. If the variance is known, the authors use catoni estimator to achieve an near optimal regret upper bound. If the variance is unknown, but the variance of variance is bounded by variance times a constant factor, then the authors propose another method that first uses catoni estimator to estimate the variance, and then use one more catoni estimator based on the estimated variance. They prove that it also achieves a near optimal regret upper bound.

=====After rebuttal=====

I read the rebuttal. My score remains.

**Claims And Evidence:**

Yes

**Essential References Not Discussed:**

No.

**Experimental Designs Or Analyses:**

N/A

**Methods And Evaluation Criteria:**

Yes

**Other Comments Or Suggestions:**

N/A

**Other Strengths And Weaknesses:**

Strengths:

- The paper's algorithm do have better regret upper bounds when the variances of arms are limited.

- The concentration for Catoni estimator is a novel contribution.


Weaknesses:

- In Algorithm 1, choosing the action $x_t$ and estimating the $\hat{f}_t$ seem require to use a double-oracle, and can be inefficient. Even if Algorithm 3 do some improvement on estimating the $\hat{f}_t$, it can still be inefficient when choosing $x_t$.

- There is no experiments, and I am wondering the applicabiity of these results.

- Some parts are not clear enough, please see questions below.

**Questions For Authors:**

1. In table 1, it seems that DistUCB has a better regret. In the footnote, it is explained that "DistUCB needs to estimate full reward distribution
 rather than just the mean", does it mean it will have a larger $ \tilde{d}_F$? If the function class is restricted to Gaussian or exponential, what is the difference between $ \tilde{d}_F$ and $ d_F$?

2. In Assumption 4.1, it is assumed that $Var[\eta^2] \le cVar[\eta]$. Is it common to have a small $c$ in practice? For example, if the noise distribution is exponential, this $c$ could be very large.

3. In Theorem 3.1, the regret lower bound depends on $\sigma_t$, which depends on $\pi_t$, i.e., the used policy. This is a little strange. In my opinion, the regret lower bound should be independent with the used policy. Can you explain more about this?

4. Why Eq. (4) holds? Do you miss a "[ ]"?

5. I can understand why equation (3) makes sense. But why we need to use this kind of "indirect" estimator? What if we just let $\hat{f}_t$ be that one with minimum $\sum_i E[(f(x_i) - y_i)^2]$? Could you explain more about this?

**Relation To Broader Scientific Literature:**

It could be helpful in real-world applications with heavy-tailed noises, for example, the waiting time in routing systems. But no such experiments are provided in the paper.

**Theoretical Claims:**

I do not check the detailed proofs in appendix.

---

> ### Author Rebuttal · Authors · 2025-03-31
>
> Thanks for your insightful suggestions.
>
> **Q1**: In Algorithm 1, choosing the action $x_t$ and estimating the $\hat f_t$ can be inefficient.
>
> **A1**: The double oracle is a standard way for optimism in online RL and bandits with general function approximation [1,2,3]. In linear function approximation, we can compute bonus $b_t$ efficiently [4] and choose $x_t=argmax_{x\in\mathcal X_t} \hat f_t(x) + b_t(x)$. However, there is no counterpart for general function approximation even when the reward range has a small bound. How to calculate the bonus efficiently is an open question even with bounded rewards.
>
> **Q2**: There is no experiments. Applicability?
>
> **A2**: Our work primarily focuses on theoretical analysis, and serves as a first step to use the Catoni estimator for nonlinear function approximation. We provide rigorous proofs demonstrating that the weighted Catoni estimator is robust against heavy-tailed noise. Although the proposed theoretical algorithm is not computationally efficient, our analysis serves as a first step toward designing robust algorithms for general function class. We believe the Catoni estimator and the variance-aware weighting technique offer valuable insights for practical use. Developing an efficient, practical version of the algorithm is a question for future work.
>
> **Q3**: In table 1, it seems that DistUCB has a better regret. Does footnote mean it has a larger $\tilde d_F$ If the function class is restricted to Gaussian or exponential, what is the difference between $\tilde d_F$ and $d_F$?
>
> **A3**: Note that DistUCB does not achieve a better regret bound due to its polynomial dependence on the reward range $R$. Moreover, the order on $\tilde d_F$ is generally incomparable to $d_F$. DistUCB assumes **realizability of the reward distribution**, i.e. true reward distribution $r = R + \epsilon$ belongs to the considered class of distributions. In contrast, we only assume realizability of the reward mean $R$. Therefore, the footnote indicates that their assumption is stronger than ours.
>
> According to Definition 5.2 in the DistUCB paper, $\tilde d_F$ measures the complexity of a class of distributions, while $d_F$ measures the complexity of a class of mean functions. These two notions are incomparable. Even in the Gaussian case, the total variation distance between $N(f, v_f)$ and $N(g, v_g)$ is incomparable to $(f - g)^2$. Further, the noise distributions may be complicated and not even satisfy the realizability.
>
> **Q4**: Assumption 4.1, is it common to have a small c in practice?
>
> **A4**: The fourth moment bound is required in most prior works (e.g. Li, et. al. and Huang, et. al) that handle the unknown variance case. Our assumption $Var[\eta^2] \le c Var[\eta]$ is equivalent to theirs when $Var[\eta]$ is bounded. Additionally, bounded second and fourth moment is already a substantially weaker condition than a bounded range within $[0,1]$.
>
> **Q5**: In Theorem 3.1, the regret lower bound depends on the used policy.
>
> **A5**: The intuition is as follows. Consider two bandit problems, each with two arms. These two bandits differ only in the reward distribution of the second arm. In both bandits, the first arm has a zero variance, and the second arm has variance $\sigma^2$. The maximum regret between these two instances depends on how often the second arm is chosen, which is given by $\mathbb{E}\sum_t \sigma_t^2 / \sigma^2$. Therefore, we obtain a lower bound directly related to the variances of the selected actions. The construction is not policy dependent, but a simple argument that there are two problem instances, and any algorithm incurs a large regret in one or the other.
>
> **Q6**: Why Eq. (4) holds? Do you miss a "[ ]"?
>
> **A6**: We miss a "[ ]" and will correct it in the revision.
>
> **Q7**: Why "indirect" estimator in (3)? What if we just let $\hat f_t$  be that one with minimum $\sum_iE[(f(x_i)-y_i)^2]$?
>
> **A7**: We have explained the intuition in Lines 203-219 right. Specifically, the reason is that directly solving
> $$
> \mathrm{argmin}_f Catoni(\{\frac{1}{\bar\sigma_i^2}(f(x_i)-y_i)^2\})
> $$
> does not work well. If we use the optimization above
>
> and use Lemma 3.2 with $Z_i=\frac{1}{\bar\sigma_i^2}(f(x_i)-y_i)^2$ that is on second-order of noise, then, we need to deal with the fourth moment of noises in the analysis even for the known variant case since Lemma 3.2 needs bounded variance of $Z_i$. In our approach, the indirect estimator helps to cancel the higher order of $y_i$ and make the term $(f(x_i)-f'(x_i))(f(x_i)-y_i)$ easier to analyze.
>
>
> [1] Jin C, et. al. Bellman eluder dimension: New rich classes of rl problems, and sample-efficient algorithms. NeurIPS, 2021.
>
> [2] Liu Q, et al. When is partially observable reinforcement learning not scary?. COLT, 2022.
>
> [3] Agarwal A, et. al. VO $ Q $ L: Towards Optimal Regret in Model-free RL with Nonlinear Function Approximation. COLT, 2023.
>
> [4] Abbasi-Yadkori Y, et. al.. Improved algorithms for linear stochastic bandits. NeurIPS, 2011.

---

### Official Review · Reviewer_Cd7h · 2025-03-06

**Overall Recommendation:** 3

**Summary:**

This paper studied the setting of variance-aware contextual bandit (or second order bandit). Specifically, this paper aims to develop algorithms whose regret is upper bounded by the variance of noise. Suppose the noise is between $[-R, R]$, all previous literatures which also studied this question obtained regret bounds that have polynomial dependence on $R$. This paper adopts the Catoni estimator from the robust statistics literature, and develops an algorithm whose regret upper bound is only log dependent on $R$.

**Claims And Evidence:**

Yes.

**Essential References Not Discussed:**

The following reference is also related to this paper. I suggest the authors provide comparison to results therein.

[1] Z Jia, J Qian, A Rakhlin, CY Wei "How Does Variance Shape the Regret in Contextual Bandits?"

**Experimental Designs Or Analyses:**

No experiments in this paper.

**Methods And Evaluation Criteria:**

Yes.

**Other Comments Or Suggestions:**

Typos:
1. Line 272, in the left: $\mathbb{E}[Z_i|x_i]$ to $\mathbb{E}[Z_i(f, f')|x_i]$
2. Line 754, right hand side: $E[\frac{1}{\bar{\sigma}_i^2}...]$ to $E[\frac{1}{\bar{\sigma}_i^4}...]$
3. In the description of Lemma 3.6 (also Lemma B.5 and the proof therein): the minimizer $f'$ to the maximize $f'$

Suggestions:
1. I suggest the authors use a different notation in $L_t(f, f')$, since it is confusing to $L_f$

**Other Strengths And Weaknesses:**

Strength:
1. The algorithm and analysis technique allows us to obtain a regret bound that only has logarithmic dependence on the noise scale. This provides a different perspective of variance-aware algorithms for contextual bandits.
2. This paper is well-written, except for a few typos.

Weakness:
1. The techniques used in this paper are not new. This paper is not the first paper that adopts the Catoni estimator in designing bandit algorithms.

**Questions For Authors:**

I have the following questions:
1. In the case where the variances are all known, the algorithm in the literature I referred above can achieve regret $\sqrt{A * \sum_{t=1}^T \sigma_t^2 * \log|F|} + d_{elu}$. However, the setting therein requires the noise scale bounded by 1. Is it possible to improve your results so that the dominating term also scales with $\sqrt{A * \sum_{t=1}^T \sigma_t^2 * \log|F|}$, while keeping the dependence on the scala of noise to be logarithmic?
2. In the current setup, the variance cannot depend on the action chosen. If the variance depends on the action chosen at each time step, will the results still work?

**Relation To Broader Scientific Literature:**

Previous literatures which studied the same problem all have regret bound polynomially depending on the noise scale, this paper developed an algorithm whose regret upper bound is only log dependent on the noise scale.

**Theoretical Claims:**

Yes, I checked the proofs for theoretical claims. All arguments seem to be correct, except from typos.

---

> ### Author Rebuttal · Authors · 2025-03-31
>
> Thanks for your constructive advice!
>
> **Q1**: The Jia, et. al. is also related to this paper. I suggest the authors provide a comparison to results therein.
>
> **A1**: We will cite their work and provide a comparison in the revision, and want to mention that Jia, et. al. is contemporary with our work. The main difference is that they assume that the noise is bounded such that $r_t\in[0,1]$, and we consider heavy-tailed rewards. Besides, although we both get variance-dependent bounds, the focuses are distinct: they aim to obtain better regret bounds when the eluder dimension is larger than the number of actions, while we aim to obtain logarithmic dependence on the reward range. Regardless of the dependence on reward range, for the weak adversary with revealed variance, our upper bound $O(\sqrt{\Lambda\cdot d_{elu} \log N})$ is incomparable to theirs $O(\sqrt{A\Lambda \log N} + d_{elu}\log N)$. For the strong adversary, we both use the peeling technique and thus, our bound is superior on the dependence of reward range.
>
> **Q2**: The techniques used in this paper are not new. This paper is not the first paper that adopts the Catoni estimator in designing bandit algorithms.
>
> **A2**: Although the Catoni estimator has previously been applied in bandit problems, it has not yet been used with nonlinear function approximation, so we provide new methods and analysis.
>
> To the best of our knowledge, our newly designed estimator in (3), which is based on excess risk, is the first one capable of handling heavy-tailed noise in a nonlinear function class. Other robust estimators, such as Huber regression and the median-of-means estimator (Lugosi and Mendelson, 2019), are limited to linear vector space structures.
>
> Additionally, we introduce a novel analysis method to establish concentration results. Existing analyses for linear heavy-tailed bandits rely explicitly on linear structures and do not generalize to the nonlinear case considered here. Furthermore, for the unknown variance scenario, we employ a peeling technique, which allows our approach to avoid the need for approximating the noise variance using a function class.
>
> **Q3**: In the case where the variances are all known, the algorithm in the literature I referred above can achieve regret $\sqrt{A*\sum_{t=1}^T\sigma_t^2*\log|F|}+d_{elu}$. However, the setting therein requires the noise scale bounded by 1. Is it possible to improve your results so that the dominating term also scales with $\sqrt{A*\sum_{t=1}^T\sigma_t^2*\log|F|}$, while keeping the dependence on the scala of noise to be logarithmic?
>
> **A3**: That is an interesting question for future work. Currently, we develop algorithms based on the OFUL structure, while Jia et al. build on the SquareCB approach. It might be feasible to connect the concentration arguments in future work. We will add this discussion to the final version.
>
> **Q4**: In the current setup, the variance cannot depend on the action chosen. If the variance depends on the action chosen at each time step, will the results still work?
>
> **A4**:  We would like to claim that our regret bounds in both Theorem 3.4 and 4.2 depend on the variance $\sigma_t$ of the chosen actions at each time step. $\sigma_t$ is defined in Line 128 left and is the variance of the noise of chosen actions.
>
> **Typos**: We will correct them in the revision.
>
> **Suggestions**: Thanks for the suggestions. We will use a difference notation to avoid confusion.
>
> [1] Jia, Zeyu, et al. How Does Variance Shape the Regret in Contextual Bandits?. NeurIPS, 2024.

---

### Official Review · Reviewer_HA2Q · 2025-03-14

**Overall Recommendation:** 4

**Summary:**

In this work, the authors proposes a novel algorithm to tackle the contextual bandit problem in the presence of heavy-tailed noise assumptions. In particular, they assume that the variance of the noise is finite and deal with the scenario in which (i) the noise variance is known to the learner, improving existing literature bounds and (ii) the noise variance is unknown to the learner, obtaining nearly matching regret bounds.

**Claims And Evidence:**

Yes, every claim is supported by a proof.

**Essential References Not Discussed:**

All the essential and related literature on contextual MABs has been discussed. For some additional literature on adaptation to the noise variance in HT MABs see the previous box.

**Experimental Designs Or Analyses:**

There is no experimental evaluation in the paper.

**Methods And Evaluation Criteria:**

There is no experimental evaluation in the paper.

**Other Comments Or Suggestions:**

there's a typo in Table 3, parameter $\lambda^l$: prameter -> parameter.

**Other Strengths And Weaknesses:**

The paper is well-written, and the approach's strength is clear and provable. Authors make sure to compare with the existing literature and highlight the extent of their improvement. Moreover, the proofs seem correct.

**Questions For Authors:**

I don't have any relevant question, except for the one already stated in the related works box.

**Relation To Broader Scientific Literature:**

As the authors discuss, their results strictly improve the ones from previous literature, especially in the known variance scenario. Regarding the adaptation to variance, it is less clear what different assumptions were made by previous works. In this work, the authors make a reasonable fourth-moment assumption, which already exists in the bandit literature [1]. Moreover, in heavy-tailed bandits, a dedicated sub-literature on adaptation to the unknown noise variance/1+epsilon moment exists [2,3,4]. I would be interested in knowing how this work relates to them: here, the focus is on the contextual scenario, but what happens if I cast these results to the unstructured MAB setting? Will they improve over the existing works?

References
[1] LATTIMORE, Tor. A scale free algorithm for stochastic bandits with bounded kurtosis. Advances in Neural Information Processing Systems, 2017, 30.
[2] HUANG, Jiatai; DAI, Yan; HUANG, Longbo. Adaptive best-of-both-worlds algorithm for heavy-tailed multi-armed bandits. In: international conference on machine learning. PMLR, 2022. p. 9173-9200.
[3] GENALTI, Gianmarco, et al. $(ε, u) $-Adaptive Regret Minimization in Heavy-Tailed Bandits. In: The Thirty Seventh Annual Conference on Learning Theory. PMLR, 2024. p. 1882-1915.
[4] CHEN, Yu, et al. uniINF: Best-of-Both-Worlds Algorithm for Parameter-Free Heavy-Tailed MABs. arXiv preprint arXiv:2410.03284, 2024.

**Theoretical Claims:**

I went through some of the proofs. Each of them seems correct to me.

---

> ### Author Rebuttal · Authors · 2025-03-31
>
> Thanks for your helpful advice!
>
> **Q1**: In heavy-tailed bandits, a dedicated sub-literature on adaptation to the unknown noise variance/1+epsilon moment exists [2,3,4]. I would be interested in knowing how this work relates to them: here, the focus is on the contextual scenario, but what happens if I cast these results to the unstructured MAB setting? Will they improve over the existing works?
>
> **A1**: We will cite [2,3,4] and include a discussion in the revised version. However, we want to highlight that our formulations and techniques differ significantly. Our work focuses on the contextual setting with general function approximation, whereas [2,3,4] consider the standard multi-armed bandit (MAB) setting. It is definitely possible to obtain sharper gap-dependent bounds in the MAB setting, but in the most general case, we can get $\sigma\sqrt{KT}$ bounds in the MAB literature, while one has $\Omega(d\sqrt{T})$ lower bounds even in the linear setting with large action spaces [1]. Consequently, the results are not directly comparable. We will add this discussion to the final version.
>
> Regarding algorithms, we adopt the OFUL framework combined with weighted Catoni estimators and peeling techniques. In contrast, [2] uses a skipping method based on Follow-the-Regularized-Leader, and [3,4] design adaptive algorithms capable of handling unknown $\alpha$ and unknown moment bounds.
>
> **Q2**: there's a typo in Table 3, parameter $\lambda^l$: prameter -> parameter.
>
> **A2**: Thanks for the correction. We will correct it in the revision.

---

### Official Review · Reviewer_24qD · 2025-03-25

**Overall Recommendation:** 4

**Summary:**

This paper introduces contextual bandit algorithms that are robust to heavy-tailed rewards by leveraging Catoni’s mean estimator from robust statistics. The authors propose two algorithms: Catoni-OFUL for the known-variance setting, and VACB, for the unknown-variance setting. Both algorithms achieve regret bounds that depend on the cumulative reward variance and scale only logarithmically with the reward range $R$, improving upon prior work that exhibits polynomial dependence on $R$. In the unknown-variance case, the authors avoid direct variance estimation by employing a peeling-based approach combined with a plug-in estimator derived from Catoni’s method. A matching lower bound is established in the known-variance setting, demonstrating the optimality of the leading-order term in the regret bound.

**Claims And Evidence:**

Most of the claims are well-supported.
1) Extensive comparisons (see Table 1) with existing algorithms highlight the advantages of the proposed methods in heavy-tailed reward settings.
2) The concentration results for Catoni’s estimator are carefully stated and integrated into the analysis with appropriate rigor.
3) The proof sketches and supporting lemmas provide a clear logical path from algorithm design to the stated regret bounds.

One minor concern: the challenges of solving the min-max optimization in Eq. (3) are acknowledged but not explored in depth. While Algorithm 3 is proposed as a more efficient alternative, the paper does not discuss its trade-offs in detail. For example, what are the theoretical or empirical pros and cons of this variant? Could a similar approach be extended to the unknown-variance setting?

**Essential References Not Discussed:**

No major omissions.

**Experimental Designs Or Analyses:**

There is no empirical evaluation in this paper. While acceptable for a theory-focused paper, a small experiment could have illustrated practical benefits of robustness. Nevertheless, the theoretical evaluation is comprehensive and grounded.

**Methods And Evaluation Criteria:**

N/A

**Other Comments Or Suggestions:**

See above.

**Other Strengths And Weaknesses:**

Novel and principled use of Catoni estimator in a contextual bandit setting.

**Questions For Authors:**

1) In the linear reward setting with known variances, does Catoni-OFUL incur slightly worse regret compared to existing algorithms such as AdaOFUL (as shown in Row 2 of Table 1)? Could the authors clarify whether this is due to the generality of the function class or an artifact of the analysis?

2) Theorem 4.2 achieves a variance-dependent regret bound that matches the known-variance case (Theorem 3.4) up to a slightly worse dependence on the eluder dimension. Could the authors provide more insight into whether this additional dependence is intrinsic to the peeling-based approach, or whether it might be improved with a different algorithmic or analytical technique?

**Relation To Broader Scientific Literature:**

N/A

**Theoretical Claims:**

Overall, the proofs are technically sound based on the provided sketches and lemmas.

For the known-variance case:
The authors leverage a uniform concentration inequality (Lemma 3.2) for Catoni’s estimator. This result is a non-trivial extension of prior Catoni bounds and is central to constructing valid confidence sets.

For the unknown-variance case:
The authors show how to control variance-normalized excess loss without exact variances. Lemma 4.4 demonstrates that the plug-in estimator for the cumulative variance (based on Catoni’s method) remains accurate up to logarithmic factors. The authors then carefully control the contribution to regret from different uncertainty levels using a peeling argument.

---

> ### Author Rebuttal · Authors · 2025-03-31
>
> Thanks for your constructive suggestions!
>
> **Q1**: The challenges of solving the min-max optimization in Eq. (3) are acknowledged but not explored in depth. While Algorithm 3 is proposed as a more efficient alternative, the paper does not discuss its trade-offs in detail. For example, what are the theoretical or empirical pros and cons of this variant? Could a similar approach be extended to the unknown-variance setting?
>
> **A1**: Theoretically, the two algorithms have the same regret bound. Computationally, Algorithm 3 is more efficient since it picks one function from the constructed candidate set instead of solving a min-max optimization. Despite this advantage, we chose to present Algorithm 1 in the main text because it is simpler, clearer, and easier to explain, and its design is better aligned with OFUL-style algorithms which readers might find familiar.
>
> For the extension to the unknown-variance case, Algorithm 3 can be extended to the unknown-variance case with similar techniques and obtain almost the same result. We will add the extension in the revision.
>
> **Q2**: There is no empirical evaluation in this paper. While acceptable for a theory-focused paper, a small experiment could have illustrated practical benefits of robustness. Nevertheless, the theoretical evaluation is comprehensive and grounded.
>
> **A2**: Our work primarily focuses on theoretical analysis, and serves as a first step to use the Catoni estimator for nonlinear function approximation. We provide rigorous proofs demonstrating that the weighted Catoni estimator is robust against heavy-tailed noise. Although the proposed theoretical algorithm is not computationally efficient, our analysis serves as a first step toward designing robust algorithms for general function class. We believe the Catoni estimator and the variance-aware weighting technique offer valuable insights for practical use. Developing an efficient, practical version of the algorithm is an important question for future work.
>
> **Q3**: In the linear reward setting with known variances, does Catoni-OFUL incur slightly worse regret compared to existing algorithms such as AdaOFUL (as shown in Row 2 of Table 1)? Could the authors clarify whether this is due to the generality of the function class or an artifact of the analysis?
>
> **A3**: The regret bounds match in the dominant term and are worse in $d$ for the non-dominant term. The reason for this difference is that the linear vector space allows a smaller variance-based weight for regression, with $d^{0.25}$ on the denominator according to Lemma B.1 and B.2 of Li, et. al. Their arguments do not generalize to the nonlinear structure, however, and we can only use $\sqrt{\log N\cdot D_{F_{t-1}}}$ ($\iota(\delta)=\Theta(\sqrt{\log N})$) in weights $\bar \sigma_t$ to balance the orders in Lemma B.2. The expense of using a larger weight is to get additional dependence on the non-dominating term (Lines 1013-1026). How to match this non-dominating term is an interesting direction for future work.
>
> **Q4**: Theorem 4.2 achieves a variance-dependent regret bound that matches the known-variance case (Theorem 3.4) up to a slightly worse dependence on the eluder dimension. Could the authors provide more insight into whether this additional dependence is intrinsic to the peeling-based approach, or whether it might be improved with a different algorithmic or analytical technique?
>
> **A4**:  This additional dependence comes from the peeling approach in general function approximation and also appears in [1,2]. The linear analysis for peeling in Zhao. et. al is restricted to the linear vector space structure and special form of uncertainty and thus can not be extended to nonlinear space. Hence, in our analysis Lines 332-337, we can only bound $\frac{(f(x_i)-f'(x_i))^2}{w_i^2}$ by $2^{-2l}\cdot\beta_{t-1}^2$ thus leading to the worse order, while $\|\hat\theta-\theta_*\|_{\Sigma_t}$ is upper bounded by $2^{-2l}\sum_i\sigma_i^2/w_i^2$ without the additional order on d.
>
>
> [1] Pacchiano, A. Second order bounds for contextual
> bandits with function approximation, 2024.
>
> [2] Jia, Zeyu, et al. How Does Variance Shape the Regret in Contextual Bandits?. NeurIPS, 2024.

---

### Decision · Program_Chairs · 2025-05-01

**Decision:**

Accept (spotlight poster)

**Comment:**

This paper provides a novel use of Catoni estimator to adapt to variances in contextual bandits. While using Catoni estimator is not new in bandits, using it for nonlinear function approximation settings is new and it was done in nontrivial and interesting ways. This paper is well-written as well.

For the final version, please address comments from the reviewers including typos.